# Combined inhibition of BADSer99 phosphorylation and PARP ablates models of recurrent ovarian carcinoma

Xi Zhang[1,2], Liqiong Wang[3], Shu Chen[2], Peng Huang[2], Lan Ma[1,4], Hui Ding[5], Basappa Basappa [6], Tao Zhu [7,8], Peter E. Lobie [1,2,3✉] & Vijay Pandey [2,3✉]

## Abstract

**Background** Poly (ADP-ribose) polymerase inhibitors (PARPis) have been approved for the treatment of recurrent epithelial ovarian cancer (EOC), regardless of *BRCA* status or homologous recombination repair deficiency. However, the low response of platinum-resistant EOC, the emergence of resistance in *BRCA*-deficient cancer, and therapy-associated toxicities in patients limit the clinical utility of PARPis in recurrent EOC.

**Methods** The association of phosphorylated (p) BADS99 with clinicopathological parameters and survival outcomes in an EOC cohort was assessed by immunohistochemistry. The therapeutic synergy, and mechanisms thereof, between a pBADS99 inhibitor and PARPis in EOC was determined in vitro and in vivo using cell line and patient-derived models.

**Results** A positive correlation between pBADS99 in EOC with higher disease stage and poorer survival is observed. Increased pBADS99 in EOC cells is significantly associated with *BRCA*-deficiency and decreased Cisplatin or Olaparib sensitivity. Pharmacological inhibition of pBADS99 synergizes with PARPis to enhance PARPi $IC_{50}$ and decreases survival, foci formation, and growth in ex vivo culture of EOC cells and patient-derived organoids (PDOs). Combined inhibition of pBADS99 and PARP in EOC cells or PDOs enhances DNA damage but impairs PARPi stimulated DNA repair with a consequent increase in apoptosis. Inhibition of BADS99 phosphorylation synergizes with Olaparib to suppress the xenograft growth of platinum-sensitive and resistant EOC. Combined pBADS99-PARP inhibition produces a complete response in a PDX derived from a patient with metastatic and chemoresistant EOC.

**Conclusions** A rational and efficacious combination strategy involving combined inhibition of pBADS99 and PARP for the treatment of recurrent EOC is presented.

## Plain Language Summary

Ovarian cancer is difficult to successfully treat because it often recurs as the cancer becomes resistant to drugs used to treat it. As such, new drugs or combinations of drugs are needed to treat patients with recurrent ovarian cancer. Here, a drug combination is reported that is effective in experimental models of ovarian cancer, including those derived from patients. The combination approach uses drugs that have previously been approved for use in patients, known as PARP inhibitors, and another drug to inhibit cancer cell survival by targeting activation of a specific protein involved in cancer cell survival. The net effect of this drug combination in ovarian cancer models is greater than the sum of the drugs used individually. With further testing, this combination may offer a potential strategy to treat patients with recurrent ovarian cancer.

[1] Shenzhen Bay Laboratory, Shenzhen, 518055 Guangdong, China. [2] Tsinghua Berkeley Shenzhen Institute, Tsinghua Shenzhen International Graduate School, Tsinghua University, Shenzhen 518055, PR China. [3] Department of Gynecology and Obstetrics, the University of Hong Kong-Shenzhen Hospital, Shenzhen, 518053 Guangdong, China. [4] Institute of Biopharmaceutical and Health Engineering, Tsinghua Shenzhen International Graduate School, Tsinghua University, Shenzhen 518055, PR China. [5] Department of Gynecology and Obstetrics, The Second Xiangya Hospital of Central South University, Changsha 410000 Hunan, China. [6] Department of Studies in Organic Chemistry, University of Mysore, Manasagangotri 570006 Mysore, India. [7] Department of Oncology of the First Affiliated Hospital, Division of Life Sciences and Medicine, University of Science and Technology of China, Hefei, Anhui 230027, China. [8] Hefei National Laboratory for Physical Sciences, the CAS Key Laboratory of Innate Immunity and Chronic Disease, Division of Life Sciences and Medicine, University of Science and Technology of China, Hefei, Anhui 230027, China. ✉email: pelobie@sz.tsinghua.edu.cn; vijay.pandey@sz.tsinghua.edu.cn

Recurrent epithelial ovarian cancer (EOC) remains the most lethal among all gynecological cancers[1]. Despite advances in early detection and treatment, the majority of EOC patients experience disease recurrence[2]. The median survival of patients with recurrent EOC ranges from 12–24 months[3]. Targeted therapies such as poly (ADP-ribose) polymerase (PARP) inhibitors have emerged as a treatment option for recurrent EOC, particularly in patients with deficient BRCA1/2 gene or a non-functional homologous recombination (HR) repair pathway[4,5]. Cancer cells with an impaired HR repair system rely on PARP-mediated DNA repair for survival[6]. Inhibition of the base excision repair (BER) pathway by PARP inhibitors (PARPis) induces double-strand breaks (DSB) from DNA single-strand (DSS) lesions, thus leading to "synthetic lethality"[7]. PARPis are also reported to be involved in "PARP trapping" on damaged DNA, defective recruitment of DNA repair proteins, and activation of error-prone non-homologous end-joining (NHEJ) in cancer cells[8]. Such features of PARPis advocate their clinical utility in patients with BRCA-proficient cancer. However, frequent transient responses to PARPis in BRCA competent or platinum-resistant EOC[9,10], low complete response (CR) to PARPi therapy (complete response (CR): 3%)[11–13], the emergence of resistance to therapy, and therapy-associated toxicities[14–17] (tabulated in Supplementary Table 5) limit the utility of PARPis in the clinic. To efficaciously improve PARPi-based treatment, novel targeted combination approaches that produce a more favorable prognosis are required.

BCL2–associated death promoter (BAD) protein is a BH3-only member of the BCL-2 family of proteins. BAD was first identified in the rat ovary being an ovarian BCL2-interacting pro-apoptotic protein and with a functional contribution to follicular atresia[18]. Unphosphorylated BAD protein heterodimerizes with and sequesters BCL2, BCL2-XL, and BCL-W with consequent promotion of intrinsic apoptosis through disruption of mitochondrial membrane potential[19]. BAD acts as a common and core downstream effector protein of both the RAS/MEK/ERK and PI3K/AKT/mTOR pathways[20]. Human (h) BAD is primarily phosphorylated at Serine (S)75 through p44/42 MAP kinase pathway activation[21], and at S99 through activation of AKT/p70S6K[22,23]. The RAS/MEK/ERK and PI3K/AKT/mTOR signaling cascades converging at BAD participate in maintaining cell survival following both cisplatin and paclitaxel treatment in ovarian cancer. Consequent phosphorylation of BAD protein promotes cell survival and resistance to therapy[24–26]. Therefore, it has been suggested that inhibition of either the RAS/MEK/ERK or PI3K/AKT/mTOR cascades may sensitize ovarian cancer cells to paclitaxel or cisplatin[27]. Interestingly, inhibition of PARP activity by Olaparib in EOC cells has also been reported to stimulate the RAS/MEK/ERK and PI3K/AKT/mTOR signaling pathways[28–31]. A number of studies have demonstrated the potential for combination therapy of PARPis with PI3K/AKT/mTOR-[32–34] or MEK-inhibitors[31] in BRCA-deficient and BRCA-proficient preclinical models[35]. Indeed, clinical responses of Olaparib and PI3K/AKT/mTOR inhibitor combinations in BRCA wild-type or platinum-resistant EOC were superior to monotherapies (summarized in Supplementary Table 7). In an ongoing phase Ib trial (NCT02208375), the mTOR inhibitor Vistusertib (AZD2014) (37) and AKT inhibitor Capivasertib (AZD5363) (38) combined with Olaparib demonstrated a 20% objective response rate (ORR) in ROC and a 24% ORR in a cohort of recurrent breast, endometrial, and OC patients respectively. Of those with OC treated with the combination of Ola and AZD5363, several patients achieved durable responses (4/6 stable disease (SD) and 1/7 partial response (PR)). Similarly, clinical trials of Olaparib in combination with different PI3K inhibitors (NCT02338622 and NCT01623349)[36–38] or MEK inhibitor Selumetinib

(NCT03162627)[39] also demonstrated clinical benefit in patients with advanced OC. Hence, combinations of PI3K/AKT/MEK inhibitors with PARPis may be efficacious beyond BRCA-associated or platinum-sensitive OC[40]. Despite effective responses of the combinations of PARPis and PI3K/AKT/MEK-inhibitors, the ORR of <30% indicates that possible feedback or compensatory bypass mechanisms were activated. To date, no combination with PARP inhibition has received regulatory approval.

Elevated BAD phosphorylation has been reported to be positively correlated with acquired cisplatin/paclitaxel resistance and poorer prognosis of EOC patients[26,41–43]. Based on bioinformatics, physio-chemical, and cell-based analyses previously described[44], a small molecule termed NPB interacts directly with human BAD protein and inhibits the phosphorylation of Serine99. NPB does not affect the phosphorylation of other signaling molecules (including AKT) nor the predominantly p44/42 MAP kinase phosphorylated Ser75 residue on BAD, demonstrated using multiple in vitro biochemical assays[44]. Thus, NPB has been proposed to hinder the phosphorylation of BADS99 independent of the upstream AKT-kinase or other phosphorylation activities[44]. Herein, it was demonstrated that the level of pBADS99 predicted a worse survival outcome for patients with EOC and that NPB was equally efficacious as PARPis in models of EOC. The combination of pBADS99 inhibition with PARPis (Olaparib, Rucaparib, or Talazoparib) synergized to enhance apoptosis of EOC cells by enhancing DNA damage by limiting PARPi-induced DNA repair. The use of EOC cell line and patient-derived organoids and xenografts further demonstrated the efficacy of the NPB-PARPi combination. Hence, a novel and efficacious regimen for the treatment of recurrent EOC is provided.

## Methods

**EOC patient specimens and histopathological analysis.** The human tissue samples used herein consisted of 80 EOC and 20 non-cancer ovarian (NCO) tissues from patients that underwent surgery at the University of Hong Kong-Shenzhen Hospital (HKU-SZH, Shenzhen, Guangdong, China) between 2016 and 2018. The use of patient specimens in this study was approved by the Institutional Ethical Committee of the HKU-SZH (certificate No. hkuszh2019105, approval No.伦 [2019] 096, Date: 2019.03.26). Patient consent forms were obtained from all patients in accordance with the declaration of Helsinki. Clinical information and follow-up data were obtained from the hospital medical records (Supplementary Data 1). The NCO tissue was collected from ovaries of patients with uterine fibroids (non-ovarian pathology) who elected to undergo concomitant salpingo-oophorectomy. All cancer patients were staged according to the International Federation of Gynecology and Obstetrics (FIGO) classification. Ki67 and TP53 status was determined after surgery.

Formalin-fixed, paraffin-embedded (FFPE) tissues were accessed from HKU-SZH. For the immunohistochemistry procedure, dewaxing was executed in xylene at room temperature for 15 min twice, then rehydration was carried out using ethanol with a concentration gradient of 100, 90, 80, 70, 50, and 0%. Tissues were placed in ethanol, 10 min twice for 100% ethanol and 5 min once for the other concentrations of ethanol. Antigen retrieval was achieved with 0.1 M freshly made sodium citrate solution at boiling temperature for 20 min. After cooling, sections were immersed in a solution of 3% hydrogen peroxide for 10 min in order to reduce endogenous peroxidase activity. The sections were washed with phosphate-buffered saline (PBS) for 5 min before blocking with 5% bovine serum albumin (BSA) diluted in PBS. The sections were then incubated with a primary antibody at

4 °C for 16 h. Immunohistochemistry (IHC) analysis was performed using primary antibodies and secondary anti-antibody tabulated in Supplementary Table 9. The sections were washed three times with PBS and incubated with biotinylated secondary antibody for 1 h. Following three washes with PBS, 3,3-diaminobenzidine (ab64238, Abcam, USA) was applied for visualization, and slides were counterstained with hematoxylin. The sections were dehydrated through graded alcohols, immersed in xylene, mounted with coverslips, and analyzed under a light microscope (CX31, Olympus, Japan) with ×4, ×10, or ×20 magnifications. Pictures were obtained with a digital CCD camera system (JVC, Tokyo, Japan). The slides were evaluated by three independent examiners using immunoreactive score (IRS) assessment[45]. IRS consists of a positive cell proportion score (0–4) and staining intensity score (0–3). For the positive cell proportion score, no cell stained was 0, the area of cells stained <10% is 1, the area 10–50% is 2, the area 51–80% is 3, and the area 81–100% was scored 4. For the staining intensity score, no staining was 0, weak staining was 1, moderate staining was 2 and, strong staining was 3. The correlation of pBADS99 IRS with clinicopathological features (Age, FIGO stage, Lymph node metastasis, Ki67, and TP53) of the EOC patient cohort was determined using Spearman's rank correlation coefficient. Survival analyses were performed on 80 EOC patients using SPSS 25 (IBMSPSS Statistics, IBM Corp., Armonk, NY, USA).

**Cell culture and reagents**. Five EOC cell lines from the Singapore ovarian cancer library (SGOCL) that were used in this study are previously described[44,46,47]. Anglne and OVCAR3 were purchased from Procell Life Science & Technology Co. Ltd (Wuhan, China). All cell lines have been tested for the absence of mycoplasma. All experiments were performed with 2% FBS in the respective media. Four fresh patient-derived EOC (~1 g) samples were obtained with written informed consent and approval from the patients and the Ethical Committee of the University of Hong Kong-Shenzhen Hospital (HKU-SZH, research No. hkuszh2019105, approval No.伦 [2019]096, Date: 2019.03.26). All pathologies were diagnosed by a pathologist. The detailed information of the patients is listed in Supplementary Fig. 6A. Tissue was minced with scissors (except for AFC cells), and the tissue fragments or AFC cells were incubated in serum-free Advanced Dulbecco's modified Eagle's medium (DMEM)/F12 (Thermo Fisher Scientific, Inc., Waltham, MA, USA) containing 1.5 mg/ml collagenase IV (Gibco; Thermo Fisher Scientific, Inc.) and 1% penicillin/streptomycin (Thermo Fisher Scientific, Inc.) at 37 °C for 2–4 h with continuous slight shaking. The cells were filtered through a cell strainer after digestion followed by centrifugation for 5 min at a speed of 500 r/min. The supernatant was removed and the pellets were washed with PBS and centrifuged as described above. Single primary cells were resuspended in Advanced Dulbecco's modified Eagle's medium (DMEM)/F12 for use in subsequent 2D or organoid culture. For organoid culture, Matrigel (354262, Corning, US) was added to the plate and polymerized for 15 min at 37 °C, and then primary cells were mixed with growth medium supplemented with 2% FBS, 50% Matrigel (354262, Corning, USA), 1x N2 Supplement (Gibco; Thermo Fisher Scientific, Inc.), 1x B27 (Gibco; Thermo Fisher Scientific, Inc.), 50 ng/ml recombinant human EGF (PEPRO-TECH; USA), and 20 ng/ml basic fibroblast growth factor (Sigma-Aldrich; Merck). 1.25mM N-Acetylcysteine (Sigma-Aldrich; Merck), 50 ng/ml Rock inhibitor (Y-27632) (Sigma-Aldrich; Merck), 10 nM 17-β Estradiol (Sigma-Aldrich; Merck) and 5 nM A83-01 (Sigma-Aldrich; Merck) were added to the wells (200 μl/well in 48-well plates and 500 μl/well in 24-well plates). PDOs usually formed within 3 days of primary culture and were

expanded for 9 days[48]. Treatment was begun on the third day after organoid formation and the medium with treatment was changed every 2 days. Organoids with diameters >100 μm were counted under an inverted microscope to determine organoid expansion. At termination, viability and apoptosis were determined using the ApoTox-Glo Triplex Assay Kit (G6320, Promega, China)[49–52]. The clinical characteristics, molecular profiling, and culture conditions for each cell line are tabulated in Supplementary Tables 3 and 4. The molecular profiling of patient-derived cells was performed at Kingmed Center for Clinical Laboratory, Guangzhou, China. Three PARP inhibitors, Olaparib (A ZD2281), Rucaparib (AG-014699), and Talazoparib (BMN 673) were purchased from Selleckchem (Houston, TX, USA).

**Western blot analysis**. Western blot analysis was performed as previously described[44]. Briefly, cells were lysed in RIPA buffer, and proteins in the cell lysate were resolved using SDS polyacrylamide gel electrophoresis and visualized with Clarity™ and Clarity Max™ Western ECL Blotting Substrates (BIO-RAD, USA). The primary and secondary anti-rabbit and anti-mouse horseradish peroxidase (HRP)-conjugated antibodies used were tabulated in Supplementary Table 9.

**Oncogenic and immunofluorescence (IF) analyses**. Cell viability, apoptosis, and cytotoxicity assays were performed using ApoTox-Glo™ Triplex Assay Kit (G6320, Promega, China) as per the manufacturer's instructions[44]. Fluorescence and luminescence were determined using a Tecan microplate reader. Phosphatidylserine exposure and cell death were assessed by CytoFLEX (Beckman Coulter, Inc. USA) using Annexin-V-FLUOS and PI-stained (Neobioscience, Shenzhen, China) cells as per the manufacturer's instructions. Live/Dead cells visualization was performed as per the manufacturer's instructions[44] using LIVE/DEAD™ Cell Imaging Kit (Thermo Fisher Scientific, USA). Combination index (CI) analysis was performed using the Chou-Talalay CI method[53].

IF analyses were performed as previously described using confocal microscopy (C2+, Nikon, Japan). Briefly, cells plated on chamber slides were fixed with 4% paraformaldehyde (PFA) for 30 min. After three washings with PBS, fixed cells were permeabilized with 0.2% Triton X-100/PBS for 5 min and incubated with 5% BSA /PBS for 15 min. Cells were incubated with primary antibodies overnight. On the second day, after three PBS washings, cells were subsequently incubated for 1 h with secondary antibodies. At least 100 cells were analyzed in three separate fields for each sample with NIS-Elements AR software. Information of primary/secondary antibodies were tabulated in Supplementary Table 9. Nuclei were stained with a mounting medium with DAPI (ab104193, Abcam). All functional assays were performed in medium with 2% FBS.

**In vivo analyses**. Xenograft studies were performed according to the Animal Research: Reporting In Vivo Experiments (ARRIVE) 2.0 guidelines. The care and use of laboratory animals were approved by the Laboratory Animal Ethics Committee (Certificate number: YW) at Peking University Shenzhen as previously described[54], and ethical approval was obtained from Tsinghua Shenzhen International Graduate School (Number:9, Year 2020). Mice were housed in a controlled atmosphere (25 ± 1 °C at 50% relative humidity) under a 12-h light/12-h dark cycle. Animals had free access to food and water at all times. Food cups were replenished with a fresh diet daily. Briefly, A2780, A2780cis, or AFC cells (5 × 10⁶ cells) were injected subcutaneously (s.c.) into the right flanks of five-week-

old female BALB/c nude mice (Charles River, Beijing). Mice ($n = 6$) bearing similar xenograft sizes were randomly assigned to different treatment arms: control, NPB, Olaparib, or NPB-Olaparib as summarized in Supplementary Fig. 8A. The statistical differences between the treatment groups were compared using a one-way ANOVA followed by a Tukey's multiple comparison test. Histological analyses were performed as previously described[44]. Xenograft volume = width × length × length/2. The response was determined by comparing tumor volume change to its baseline at time $t$: % tumor volume change = $\Delta$Volt = $100\% \times ((V_t - V_{Vinitial})/V_{Vinitial})$. The Best Response was the minimum value of $\Delta$Volt for $t \geq 10$ d. For each time $t$, the average of $\Delta$Volt from 0 to $t$ was also calculated. Best average response was defined as the minimum value of this average for $t \geq 10$ d. The criteria for response (mRECIST)[55,56] were defined as followed: complete response (mCR): best response < −95% and best avg response < −40%; partial response (mPR): best response < −50% and best avg response < −20%; stable disease (mSD), best response < 35% and best avg response < 30%; progressive disease (mPD), not otherwise categorized.

**Statistics and reproducibility**. SPSS 25 (IBMSPSS Statistics, IBM Corp., Armonk, NY, USA) and GraphPad Prism 7.0 (GraphPad Software, San Diego, CA, USA) were used to generate graphs and perform statistical analysis as previously described[44]. For in vitro assays, the statistical differences between the treatment groups were compared using a one-way ANOVA followed by a Tukey's multiple comparison test. $p$-values < 0.05(*), $p < 0.01$(**) and $p < 0.001$(***) were considered statistically significant. Quantitative data are expressed as mean ± SD, unless otherwise stated (Supplementary Data 2). See individual "Methods" sections for specific statistical methods. Histopathological assays were replicated at least three times, with each orthogonal method confirming the same result. Western blot assays were replicated at least three times in each tested cell line, showing similar results. Immunoprecipitations followed by silver stains and immunoblots were replicated at least twice. IF assays were performed in triplicate. Different oncogenic assays were performed independently at least three times and showed similar results. Each group in in vivo experiments contained at least six mice.

**Reporting summary**. Further information on research design is available in the Nature Research Reporting Summary linked to this article.

## Results

**Higher BADS99 phosphorylation (pBADS99) levels in EOC are significantly associated with poor survival outcome**. Immunohistochemistry (IHC) was utilized to determine the level of pBADS99 and the expression of BAD in EOC tissue specimens. The potential association between pBADS99 and clinicopathological characteristics of EOC patients was determined. As observed in Fig. 1A, higher levels of pBADS99 positively correlated with higher labeling of Ki67 ($\rho = 0.274$, $p = 0.014$*) and higher International Federation of Gynecology and Obstetrics (FIGO) stage ($\rho = 0.629$, $p < 0.001$***) (Fig. 1A). Kaplan–Meier survival analysis was also performed in the EOC patient cohort, which sub-categorized the level of BADS99 as low or high using a receiver operating characteristic curve (Supplementary Fig. 1A, B). It was observed that EOC patients with higher cancer levels of pBADS99 (IRS = 6–12) ($n = 51$) exhibited significantly poorer progression-free survival (PFS) and overall survival (OS) ($p < 0.05$) compared to those patients whose cancers possess no/low levels of pBADS99 (IRS = 0–4) ($n = 29$) (Fig. 1B). Furthermore, Cox regression analysis demonstrated a significant association of high cancer pBADS99 levels with decreased PFS and OS (Supplementary Fig. 1C). No significant correlation was observed between BAD protein expression and EOC patient PFS or OS (Supplementary Fig. 1D). Thus, phosphorylation of BADS99 is increased in EOC and predicts poor survival outcomes for patients.

Next, the level of pBADS99 and expression of BAD were determined in eight EOC cell lines as described in the methods section. All EOC cell lines exhibited phosphorylation of BADS99 and expression of the BAD protein, as demonstrated using western blot analysis (Fig. 1C). After normalization with the level of BAD protein, BRCA-deficient OVCA433, HEYC2, A2780CisR, and AFC cells exhibited significantly higher levels of pBADS99 compared to BRCA-proficient Anglne, A2780, OVCAR3, and CAOV2 cells. Similarly, platinum-resistant OVCAR433[57], HEYC2[58], A2780CisR, and AFC cells exhibited higher levels of pBADS99 compared to platinum-sensitive A2780, Anglne, OVCAR3, and CAOV2 cells (Fig. 1D). In addition, elevated levels of pBADS99 in EOC cells were observed to be positively correlated with the mutation of PI3KCA and KRAS but negatively associated with TP53 (Supplementary Fig. 1E).

Olaparib monotherapy has been approved as front-line maintenance for patients with either germline or somatic BRCA-mutated ovarian cancer or recurrent EOC regardless of BRCA status 58. Therefore, the correlation of pBADS99 with Olaparib response was determined. The half-maximal inhibitory concentration (IC$_{50}$) of Olaparib (after 6 days) in EOC cells positively correlated with the levels of pBADS99 (Fig. 1F). However, it should be noted that the IC$_{50}$ values of Rucaparib or Talazoparib after 6 days of treatment did not exhibit a positive correlation with the levels of pBADS99 in EOC cells (Supplementary Fig. 1F). However, the IC$_{50}$ of Rucaparib at 48 h and the IC$_{50}$ of Talazoparib at 72 h positively correlated with the levels of pBADS99 in EOC cells, respectively.

Furthermore, the functional contribution of BADS99 in regulating EOC cell survival was assessed by transient-transfection of human BAD and an S99 mutated human BAD (hBADS99A) with a flag-tag (construct described in Supplementary Methods and Supplementary Table 8) into CAOV2 cells. As shown in Supplementary Fig. 1G, forced expression of hBAD-flag in CAOV2 cells increased BAD expression and pBADS99 levels compared to vector-transfected cells. Forced expression of hBADS99A-flag increased hBAD compared to vector-transfected cells and produced equivalently increased hBAD (and flag-tag) as was observed in hBAD-flag transfected cells, compared to vector-transfected cells. Forced expression of hBADS99-flag did not appreciably decrease hBADS99 compared to vector-transfected cells but pBADS99 was significantly less than that observed in hBAD-flag transfected cells. Hence, the densitometric analysis showed that the ratio of pBADS99/BAD decreased after the forced expression of pBADS99A. Functionally, forced expression of hBADS99A in CAOV2 cells resulted in decreased cell viability ($p = 0.002$) and increased CASPASE 3/7 activity ($p = 0.0003$) compared to control vector-transfected cells (Supplementary Fig. 1G). To confirm the Flag-hBADS99A construct was appropriately located within the cell, immunofluorescence (IF) was performed with BAD, pBADS99, and flag antibodies in Flag-tagged wild-type hBAD and hBADS99A transfected CAOV2 cells (Supplementary Fig. 1H). The results are consistent with the western blot data. Note that similar to wild-type BAD protein, Flag-tagged constructs are largely localized within the cytoplasm of the cells (Supplementary Fig. 1H).

**NPB synergizes with PARP inhibition in EOC cells to decrease survival**. It was next examined whether the small molecule

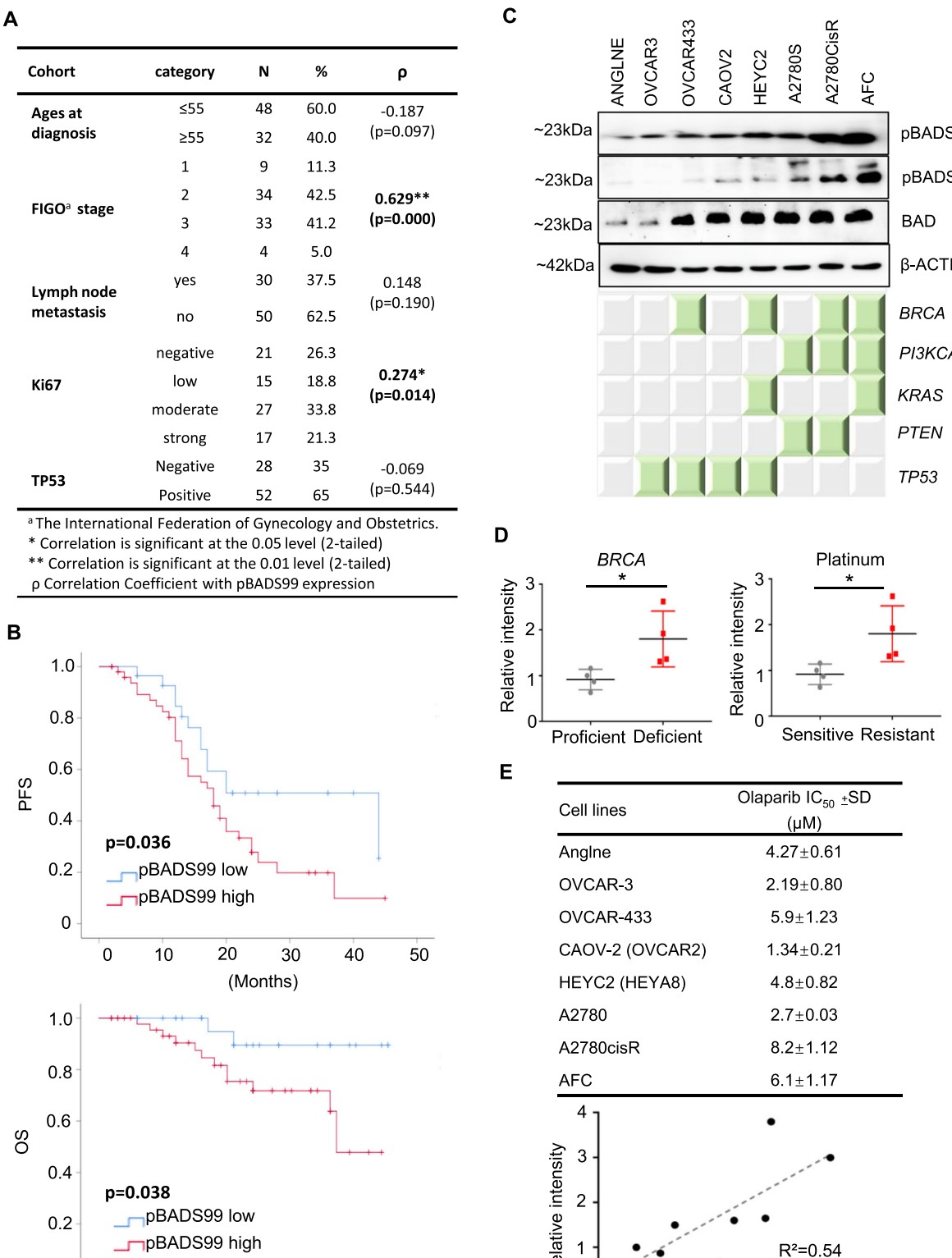

**A**

| Cohort | category | N | % | ρ |
|---|---|---|---|---|
| **Ages at diagnosis** | ≤55 | 48 | 60.0 | -0.187 (p=0.097) |
| | ≥55 | 32 | 40.0 | |
| **FIGO[a] stage** | 1 | 9 | 11.3 | **0.629** (p=0.000) |
| | 2 | 34 | 42.5 | |
| | 3 | 33 | 41.2 | |
| | 4 | 4 | 5.0 | |
| **Lymph node metastasis** | yes | 30 | 37.5 | 0.148 (p=0.190) |
| | no | 50 | 62.5 | |
| **Ki67** | negative | 21 | 26.3 | **0.274*** (p=0.014) |
| | low | 15 | 18.8 | |
| | moderate | 27 | 33.8 | |
| | strong | 17 | 21.3 | |
| **TP53** | Negative | 28 | 35 | -0.069 (p=0.544) |
| | Positive | 52 | 65 | |

[a] The International Federation of Gynecology and Obstetrics.
* Correlation is significant at the 0.05 level (2-tailed)
** Correlation is significant at the 0.01 level (2-tailed)
ρ Correlation Coefficient with pBADS99 expression

**B**

**C**

**D** BRCA   Platinum

**E**

| Cell lines | Olaparib IC$_{50}$ ±SD (μM) |
|---|---|
| Anglne | 4.27±0.61 |
| OVCAR-3 | 2.19±0.80 |
| OVCAR-433 | 5.9±1.23 |
| CAOV-2 (OVCAR2) | 1.34±0.21 |
| HEYC2 (HEYA8) | 4.8±0.82 |
| A2780 | 2.7±0.03 |
| A2780cisR | 8.2±1.12 |
| AFC | 6.1±1.17 |

$R^2 = 0.54$
$P = 0.040$

inhibitor of BADS99 phosphorylation, NPB[44], synergized with PARP inhibition in EOC cell lines (Fig. 2, Supplementary Fig. 2). Based on different PARP-trapping capacities[59–62] (Supplementary Table 6), Olaparib was tested along with Rucaparib and Talazoparib for combination efficacy with NPB in EOC cells. Utilizing combination index (CI) analysis, NPB exhibited a synergistic response with all PARPis in all eight EOC cells (Fig. 2, Supplementary Fig. 2). Next, the effect of NPB on the IC$_{50}$ of the

**Fig. 1 Increased phosphorylation of BAD at Ser99 (pBADS99) in epithelial ovarian cancer (EOC) is significantly associated with reduced survival of patients. A** Clinicopathological association with pBADS99 levels: Spearman correlation analysis between pBADS99 expression and clinicopathological features of the EOC patient cohort. **B** Kaplan–Meier analysis: relapse and survival times were compared by Kaplan–Meier analysis and overall survival (OS) or progression-free survival (PFS) were tested for significance by using the log-rank test. The immunoreactive score (IRS) 0–4 categorized was categorized as low pBADS99 ($n = 29$) and IRS 6–12 as high pBADS99 respectively ($n = 51$) as described in the methods section. **C** pBAD/BAD levels in EOC cell lines: western blot analysis was used to assess the levels of pBADS75 and pBADS99 and BAD expression in EOC cell lines. Soluble whole-cell extracts were run on an SDS-PAGE and immunoblotted as described in the methods section. β-ACTIN was used as input control for cell lysate. The sizes of detected protein bands in kDa are shown on the left side. Densitometries of protein bands were determined using ImageJ software (https://imagej.nih.gov/ij/). Specific gene mutations in the EOC cell lines are indicated by the green-colored square, described in the methods section. **D** pBADS99 levels in EOC cell lines: association between pBADS99 and *BRCA*-deficiency or platinum-resistance in EOC cells. **E** Correlation of pBADS99 to $IC_{50}$ of Olaparib: correlation of pBADS99 with Olaparib half-maximal inhibitory concentration ($IC_{50}$) values in EOC cells. The $IC_{50}$ of Olaparib was calculated by total cell number counting ($n = 3$). Elevated levels of BADS99 phosphorylation exhibited a negative correlation with decreased Olaparib sensitivity in EOC cells. Statistical changes were measured using Pearson correlation analysis between pBADS99 and $IC_{50}$ values of Olaparib in EOC cells. Columns or points are mean of triplicate experiments; bars, ±SD. ***$p < 0.001$, *$p < 0.05$.

PARPis was determined. The addition of 1 µM NPB produced a substantial decrease in PARPi $IC_{50}$ in EOC cell lines. For example, NPB decreased the $IC_{50}$ of Olaparib in *BRCA*-deficient AFC cells by ~350-fold or in OVCAR433 cells by ~400-fold. In addition, NPB decreased the $IC_{50}$ of Olaparib in platinum-resistant A2780CisR cells by ~50-fold. NPB also decreased Olaparib $IC_{50}$ in wild-type Ang1ne cells by ~120-fold (Supplementary Fig. 2). Similar directional fold changes were also observed with NPB in combination with Rucaparib or Talazoparib in EOC cell lines. Moreover, NPB in combination with PARPis exerted synergistic effects with significant cell fraction affected (Fa = 0.50~0.99) in all EOC cell lines (Supplementary Tables 1 and 2). Hence, NPB synergized with PARPis to decrease the survival of EOC cells. To confirm the functional specificity of NPB directed to BAD, CAOV2 cells were treated with NPB after siRNA-mediated depletion of BAD expression (Supplementary Fig. 3A). Transient-transfection of CAOV2 cells with siRNA directed to the BAD transcript decreased BAD expression and decreased levels of pBADS99 compared to control cells (transfected with scrambled oligo) as observed by WB analysis (Supplementary Fig. 3A). No significant changes in CASPASE 3/7 activity nor cell viability were observed upon siRNA-mediated depletion of BAD in CAOV2 cells (Supplementary Fig. 3A) and as previously reported in various carcinoma cells (44). NPB treatment of control transfected cells decreased phosphorylation of BADS99 compared to vehicle-treated CAOV2 cells. Consistently, exposure of CAOV2 control cells to NPB increased CASPASE 3/7 activity and decreased cell viability compared to vehicle-treated cells. In contrast, NPB did not affect CASPASE 3/7 activity nor cell viability in CAOV2 cells with the depleted expression of BAD. Furthermore, no synergistic effect was observed between NPB and PARPis in CAOV2 cells with siRNA-mediated depletion of *BAD* compared to control transfected CAOV2 evaluated by a total cell number assay (Supplementary Fig. 3B).

**NPB synergizes with PARP inhibition to stimulate apoptotic cell death attenuating PARPi-induced HR repair.** Pre-grown colonies of CAOV2 cells in 3D Matrigel were treated with NPB and PARPis either alone or in combination. Live/dead cells analysis of the CAOV2 colonies was evaluated. Markedly increased red staining (indicating loss of plasma membrane integrity) and decreased green staining were observed with combined treatment of NPB and Olaparib compared to NPB or Olaparib alone (Fig. 3A). To determine the survival fraction (SF), ApoTox-Glo Triplex assays were used to evaluate live-cell protease activity (cell viability), and caspase 3/7 activity (apoptosis) of the colonies cultured in 3D Matrigel. NPB in combination with Olaparib significantly decreased cell viability and increased caspase 3/7 activity compared to NPB or Olaparib-treated CAOV2 colonies (Fig. 3B, Supplementary Fig. 4A). Moreover, treatment

of CAOV2 colonies with NPB in combination with Olaparib significantly promoted early (PI negative, FITC-Annexin V positive) and late apoptotic cell death (PI-positive, FITC-Annexin V positive) compared to NPB or Olaparib-treated cells (Fig. 3C, Supplementary Fig. 4B). Concomitantly, CAOV2 cells treated with NPB and Olaparib marginally increased cell populations in the G1-phase along with reduced G2M-phase compared to cells treated with Ola (Supplementary Fig. 4C). In addition, CAOV2 cells treated with NPB and Olaparib exhibited significantly reduced capacity for colony formation on monolayer compared to NPB or Olaparib-treated cells (Fig. 3D, Supplementary Fig. 4D). A similar synergy between NPB in combination with Rucaparib or Talazoparib to stimulate apoptotic cell death was also observed in CAOV2 cells.

The mechanisms underlying the synergistic effects of the combination treatment on apoptotic cell death in CAOV2 cells were analyzed using western blot. NPB treatment of CAOV2 cells produced significantly decreased pBADS99 levels compared to vehicle-treated cells (normalized to BAD expression). In contrast, Olaparib treatment of CAOV2 cells did not alter the pBADS99/BAD ratio compared to vehicle-treated cells. Compared to NPB or Olaparib alone, the cells treated with a combination of NPB and Olaparib exhibited further decreased levels of pBADS99/BAD. Olaparib treatment of cells increased phosphorylation of CHK1 at Ser345 and CHK2 at Thr383 (normalized to CHK1 and CHK2 protein, respectively). In contrast, NPB did not produce a significant change in the phosphorylation of CHK1 or CHK2 protein compared to vehicle-treated cells. In contrast, the NPB-Olaparib combination decreased phosphorylation of CHK1 (Ser345) and CHK2 (Thr383) compared to Olaparib-treated cells. NPB treatment of CAOV2 cells exhibited decreased CCND1 protein levels compared to vehicle-treated cells. Increased levels of γH2AX were observed in CAOV2 cells after Olaparib treatment compared to vehicle or NPB, and a combination of NPB and Olaparib significantly increased the levels of γH2AX. A significantly decreased level of KI67 and increased levels of cleaved CASP7 were observed in CAOV2 cells after treatment with NPB. Similar directional changes were also observed with combined treatment of CAOV2 cells with NPB and Rucaparib or Talazoparib (Fig. 3E, Supplementary Fig. 5A).

Olaparib treatment of OC cells has been reported to increase the expression of the DNA damage signaling marker, γH2AX, along with RAD51, an indicator of HR repair in OC cells[63]. γH2AX expression was enhanced in cells treated with NPB or Olaparib and further increased in combination (Fig. 3F, Supplementary Fig. 5B). Olaparib induced RAD51 foci formation in the nuclei of CAOV2 cells and which was significantly impaired in combination with NPB (Fig. 3F, Supplementary Fig. 5B). Similar directional changes were observed in the expression and foci formation of γH2AX or RAD51 in OVCAR433 cells treated with

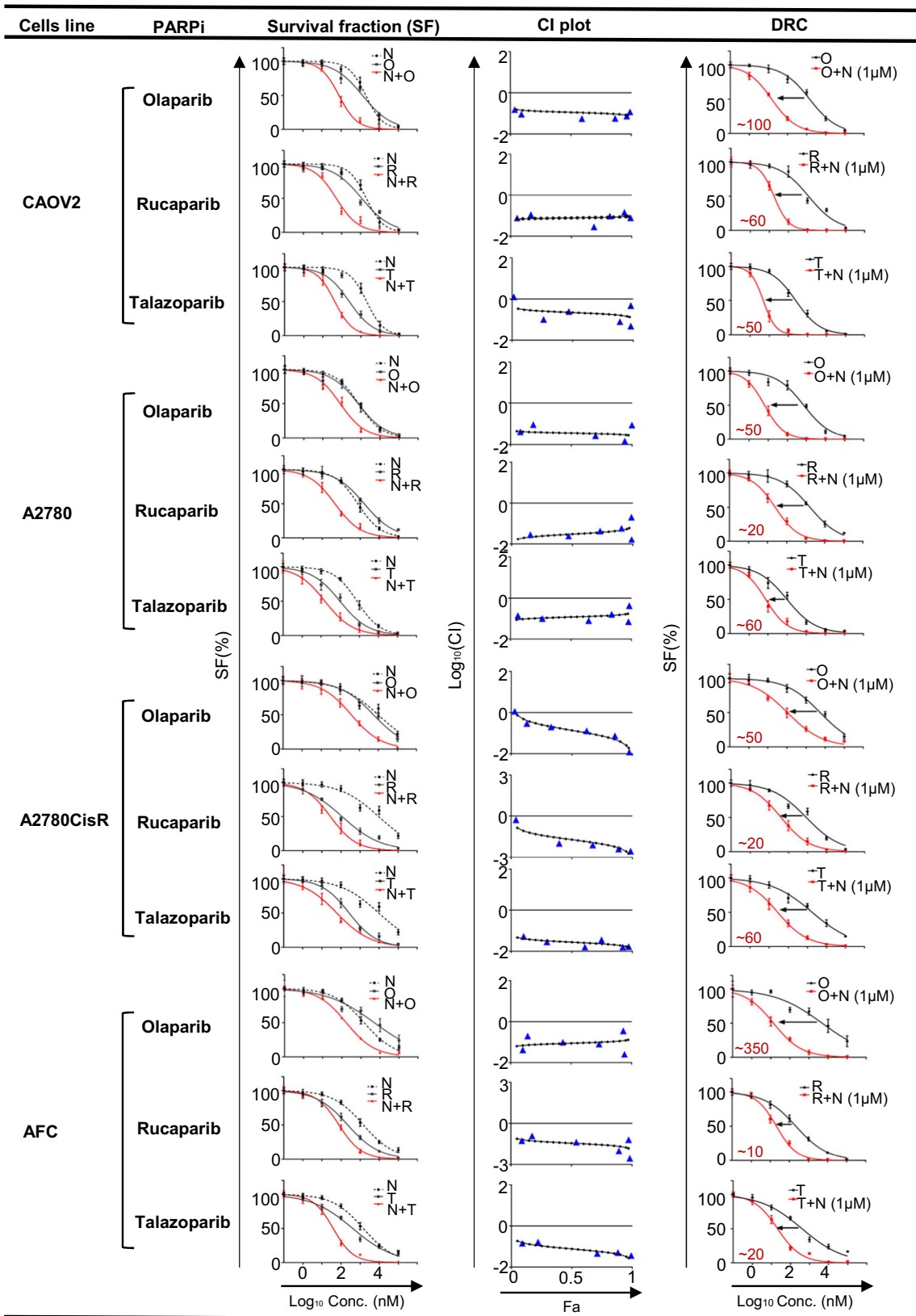

NPB-Olaparib (Supplementary Fig. 5C). It has been reported that dephosphorylated BAD sequesters BCL-XL, BCL-2, and BCL-W, which results in BAK/BAX activation and apoptosis[24]. Treatment of CAOV-2 cells with NPB enhanced the interaction of BAD with BCL-2, BCL-XL, or BCL-W as observed by co-immunoprecipitation (co-IP) assays (Supplementary Fig. 5D). It is recognized that sub-lethal apoptotic stresses trigger a caspase-dependent increase in γ-H2AX expression[64–68]. As DNA

**Fig. 2 NPB synergizes with PARP inhibitors in EOC cells to decrease cell survival.** Chou-Talalay analysis was used for NPB (N) in combination with three PARP inhibitors, namely Olaparib (O), Rucaparib (R), and Talazoparib (T), in a panel of EOC cells including CAOV2 (BRCA-proficient/Platinum sensitive), A2780 (BRCA-proficient/Platinum sensitive), A2780cisR (BRCA- proficient /Platinum resistant) and AFC (BRCA-deficient/Platinum-resistant). Cells were treated with the indicated concentration (Conc., $\log_{10}$ scale) of NPB and mentioned PARP inhibitors for 6 days. The survival fraction (SF) was assessed using a total cell number assay. The logarithmic combination index (CI) value corresponding to cell fraction affected (Fa) was determined using the CompuSyn software (http://www.combosyn.com) as described in the methods section. CI value indicates: <1 synergism; =1 additive synergy; >1 antagonism ($n = 3$). Dose-response curves (DRC) for a panel of cells treated with the indicated concentration of PARP inhibitors with or without 1 μM NPB in total cell number assays. Arrow indicates fold reduction in respective PARP inhibitor IC$_{50}$ in the presence of NPB ($n = 3$). Points are mean of triplicate experiments; bars, ±SD.

fragmentation is a classic apoptotic hallmark mediated by caspase-activated DNase (CAD), DNA laddering assays were performed. After NPB treatment, apoptotic DNA laddering increased accompanied by increased CASPASE-3 and PARP cleavage, and γ-H2AX expression consistent with previous studies reporting that increased γ-H2AX levels may also indicate apoptosis[66–68] (Supplementary Fig. 5E). Transient-transfection of Ser99 phosphorylation deficient human *BAD* (*hBADS99A*) construct into CAOV2 cells also produced increased levels of γ-H2AX protein as evaluated by immunofluorescence (Supplementary Fig. 5F). Given that NPB treatment diminished PARPi-induced CHEK1/CHEK2 phosphorylation and RAD51 foci, which both have an important role in HR[69], it was therefore determined whether NPB modulated HR. A HR reporter assay demonstrated that NPB treatment suppressed HR efficiency both in *BRCA* wild-type (CAOV2) and *BRCA* mutant (AFC) EOC cells (Supplementary Fig. 5G). Hence, NPB attenuated the repair of DSBs induced by PARPi by suppression of HR. Thus, NPB synergizes with PARPi in EOC cells to promote DNA damage and impair DNA repair after PARPi treatment.

**NPB synergizes with PARP inhibitors to decrease cell survival in patient-derived EOC organoids.** Patient-derived EOC organoid (PDO) cultures were established from four patients with recurrent EOC, namely PDO1-4 (Supplementary Fig. 6A–C): PDO1 is derived from post-chemotherapy metastatic cancer cells of ascites; PDO2 and PDO4 both are derived from resected primary cancer and PDO3 is derived from residual cancer after 3-cycles of platinum-based chemotherapy. To determine the effect of NPB in combination with PARPis on apoptotic cell death in EOC PDOs, the effect of NPB, PARPis and the combination was first assessed using PDO1 cells on 2D Matrigel. Both NPB and Olaparib alone decreased PDO1 cell survival. Compared to either Olaparib or NPB alone, PDO1 cells treated with the combination of NPB and Olaparib exhibited significantly decreased cell survival (Supplementary Fig. 6D). Combined NPB-Olaparib treatment of preformed PDO1 cell-derived colonies in 3D Matrigel markedly increased apoptotic cell death (indicated by red staining) compared to NPB or Olaparib alone (Supplementary Fig. 6E). In addition, compared to either NPB or Olaparib alone, PDO1 treated with the combination of NPB and Ola exhibited significantly increased CASPASE 3/7 activities and decreased cell survival (Fig. 4A). Moreover, the combined NPB-Olaparib treatment of PDO1 also significantly reduced organoid growth in 3D Matrigel (Supplementary Fig. 7A, B) and foci formation capacity in monolayer culture (Fig. 4B). Similar directional changes in cell survival, CASPASE 3/7 activities, and organoid growth were observed when PDOs2-4 were treated with PARPis either alone or with the NPB-PARPi (Olaparib, Rucaparib, and Talazoparib) combinations (Fig. 4A, Supplementary Fig. 7A, B).

NPB treatment of PDO1 cells significantly decreased levels of pBADS99 compared to vehicle-treated cells (normalized to BAD expression). In contrast, PARPi treatment of PDO1 cells did not alter the pBADS99/BAD ratio compared to vehicle-treated cells. Compared to PARPis alone, cells treated with a combination of NPB and PARPis exhibited decreased phosphorylation of CHK1 at Ser345 and CHK2 at Thr383 residue (normalized to CHK1 and CHK2 protein, respectively). The combined NPB-Olaparib treatment of PDO1 cells also exhibited decreased levels of CCND1 protein compared to Olaparib-treated PDO1 cells, whereas no significant change was observed compared to NPB-treated PDO1 cells. Markedly, increased levels of γH2AX in PDO1 cells were observed after combined NPB-Olaparib treatment compared to single-agent treatment. Levels of cleaved CASP7 subunit protein also increased after combined NBP-Olaparib treatment of PDO1 cells compared to either NPB or Olaparib-treated cells. Similar directional changes in protein levels were observed with NPB in combination with Rucaparib in PDO1 cells. However, combined NPB-Talazoparib treatment of PDO1 cells did not produce significant changes in γH2AX nor cleaved CASP7 levels compared to either NPB or Talazoparib treated cells (Fig. 4C and Supplementary Fig. 6F). To confirm the functional specificity of NPB to BAD, the effect of NPB exposure after siRNA-mediated depletion of BAD expression was examined in PDO1 cells (Supplementary Fig. 6G), transient-transfection of PDO1 cells with siRNA directed to the *BAD* transcript decreased BAD expression and decreased levels of pBADS99 compared to control PDO1 cells (transfected with scrambled oligo) as observed by WB analysis (Supplementary Fig. 6G). No significant changes in cell viability nor CASPASE 3/7 activity were observed upon *siRNA*-mediated depletion of *BAD* in PDO1, as previously reported for various carcinoma cells (44). As observed in Supplementary Fig. 6G, NPB treatment of the control transfected PDO1 cells decreased phosphorylation of BADS99 compared to control transfected PDO1 cells treated with vehicle. Consistently exposure of control transfected PDO1 to NPB decreased cell viability and increased CASPASE 3/7 activity compared to vehicle-exposed PDO1. In contrast, NPB did not affect CASPASE 3/7 activity nor cell viability in PDO1 with the depleted expression of BAD (Supplementary Fig. 6G).

**NPB and Olaparib synergize to suppress the growth of cisplatin-resistant EOC xenografts.** The effect of NPB in combination with Ola in a xenograft model of recurrent EOC was examined. Xenografts were generated by subcutaneous injection of A2780 and A2780CisR cells into immunocompromised mice. Xenograft-bearing (volume 50~100 mm$^3$ for A2780 cell line/ volume 100~150 mm$^3$ for A2780cisR cell line) mice were randomly grouped ($n = 6$) and were injected i.p. with the vehicle, Olaparib (50 mg/kg), NPB (20 mg/kg), and the combination of NPB and Olaparib daily at the same respective concentrations. The treatment regimen is summarized in Supplementary Fig. 8A. All mice were sacrificed 8 days after commencement of drug treatment. The A2780/A2780CisR generated xenograft volume and mouse body weight was measured daily and are represented

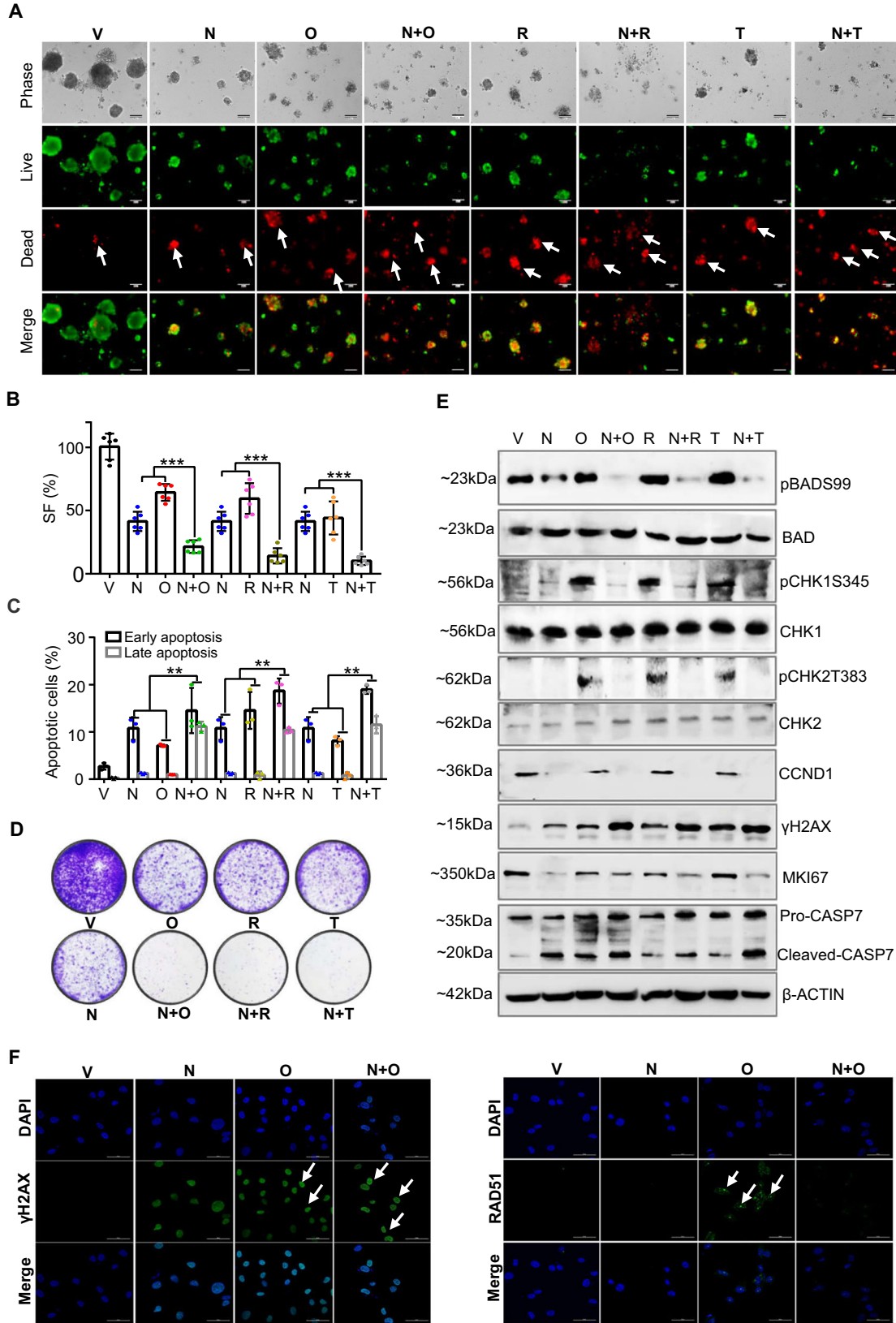

in Fig. 5A. Xenografts derived from A2780 and A2780CisR cells in all treated groups (Olaparib, NPB, or Olaparib-NPB) exhibited a significant reduction in volumes after 3 days of treatment compared to the vehicle group. The NPB-treated group exhibited

significantly decreased A2780 and A2780CisR cell-generated xenograft volume compared to Olaparib-treated animals. In addition, the A2780 and A2780CisR cell-generated xenografts treated with NPB-Olaparib exhibited significantly decreased

**Fig. 3 NPB synergizes with PARP inhibitors to stimulate apoptotic cell death in CAOV2 cells. A** Live/dead cells in 3D matrigel: microscopic visualization of calcein-AM (green) stained colonies (live) and BOBO-3 Iodide (red) stained cell debris (dead) generated by CAOV2 cells cultured in 3D Matrigel after exposure to NPB (N), PARP inhibitors (Olaparib (O), Rucaparib (R), Talazoparib (T)) and combination ($n = 3$). White arrows indicated positive staining. Scale bars, 100 μm. **B** 3D matrigel growth: Survival fraction (SF) was evaluated using the ApoTox-Glo triplex assay Kit in CAOV2 cells treated with NPB (N), PARP inhibitors (OlaparibO, Rucaparib = R, Talazoparib = T) alone or in combination with the indicated concentrations for 14 days in 3D Matrigel culture ($n = 3$). **C** Apoptosis: flow cytometry analysis of Annexin-V and propidium iodide (PI) staining of apoptotic cell death of CAOV2 cells measured after treatment with NPB (N), Olaparib (O), Rucaparib (R), and Talazoparib (T) or combinations using flow cytometry analysis at 72 h as described in methods. Apoptotic cells refer to Annexin-V positive (early apoptosis) + Annexin-V and PI double-positive (late apoptosis) cells ($n = 3$). **D** Foci formation: CAOV2 cells were incubated in the indicated drug concentrations of NPB (N), Olaparib (O), Rucaparib (R), and Talazoparib (T) or combinations for colony formation stained with 0.2% crystal violet ($n = 3$). **E** Western blot: western blot analysis was used to assess the level of various proteins and protein activities in CAOV2 cells after treatment with NPB (N), Olaparib (O), Rucaparib (R), Talazoparib (T), or combinations thereof. Soluble whole-cell extracts were run on an SDS-PAGE and immunoblotted as described in the methods section. β-ACTIN (ACTB) was used as input control for cell lysate. The sizes of detected protein bands in kDa are shown on the left-hand side. **F** IF analysis: representative images of fluorescent DNA damage markers γH2AX and RAD51 in CAOV2 cells were treated with NPB (N), Olaparib (O), or a combination (N + O) for 16 h are taken by confocal microscope. Scale bars, 50 μm. Treatment concentration of NPB, 5 μM; Olaparib, 10 μM; Rucaparib, 5 μM; and Talazoparib, 5 μM. Columns are mean of triplicate experiments; bars, ±SD. *$p < 0.05$, **$p < 0.01$, ***$p < 0.001$.

volume compared to either Olaparib or NPB-treated animals (Fig. 5A). Statistical significance was analyzed based on endpoint xenograft volume. No significant changes in body weight (Fig. 5A) were observed in the treatment groups compared with the vehicle group during the treatment period. Consistently, all treated groups from both A2780 and A2780cisR cell-derived xenografts (NPB, Olaparib, and NPB-Ola) exhibited decreased xenograft weight as compared to the vehicle-treated group. Animals treated with the combination NPB-Olaparib exhibited significantly reduced xenograft weight compared to either NPB or Olaparib alone. Notably, the NPB-treated group also exhibited reduced xenograft weight compared to the Olaparib-treated animal group, although the change for both A2780 and A2780CisR cell-derived xenografts (Fig. 5B and Supplementary Fig. 8B) was not significant.

Histological analyses were performed on the resected xenograft specimens. Stable pooled A2780-Vector and A2780-BAD-knock out (KO) cells using CRISPR-CAS9 (Supplementary Methods) mediated deletion of the BAD gene were generated in A2780 cells to confirm the specificies of the pBADS99 and BAD antibodies (Supplementary Fig. 8C). Xenograft specimens resected from animals treated with NPB exhibited significantly reduced pBADS99-positive cells in both A2780 and A2780cisR cell-derived xenografts compared to vehicle-treated control. No significant change in the percentage of pBADS99-positive cells was observed between Olaparib and vehicle-treated xenograft specimens. The combination of NPB-Olaparib reduced the percentage of pBADS99-positive cells compared to either NPB or Olaparib-treated xenograft specimens. No significant differences were observed in the percentage of BAD-positive cells between the different treatment groups. NPB-Olaparib-treated A2780 and A2780cisR cell-generated xenograft specimens exhibited a decreased percentage of KI67-positive cells and a significantly increased percentage of cleaved-CASP3 positive cells compared to either NPB or Olaparib-treated specimens. No significant change was observed in cleaved-CASP7 positive cells between the groups of xenograft specimens treated with either combined NPB-Olaparib or with single agent (Supplementary Fig. 8D). However, NPB-Olaparib-treated specimens exhibited significantly increased TdT-mediated dUTP nick-end labeling (TUNEL) positive cells compared to either NPB or Olaparib-treated specimens (Fig. 5C).

Next, the potential efficacy of NPB, Ola, or combined NPB-Ola in a patient-derived xenograft (PDX) model of recurrent EOC was examined (obtained from PDO1: *gBRCA2$^{MUT}$* (KingMed, Guangdong, China) and acquired resistance to cisplatin therapy) (Supplementary Table 2). The PDX was generated by subcutaneous injection of AFCs into immunocompromised mice. After 3 days, xenograft-bearing (volume 150~200 mm³) mice were randomly grouped ($n = 6$) for treatment with vehicle, NPB (20 mg/kg), Olaparib (50 mg/kg), or NPB-Olaparib. The mice were euthanized when xenografts reached the humane endpoint (volume ≥ 1100 mm³) as a surrogate for lifespan[70]. Vehicle-treated mice reached the humane endpoint first and were euthanized on days 15 ($n = 2$), 16 ($n = 2$), 17 ($n = 1$) and 18 ($n = 1$). As observed in Fig. 5D, rapid growth in the AFC-generated xenograft volume was observed in animals in the vehicle control group. In contrast, animals treated with either NPB or Olaparib exhibited an initial regression and then delayed growth in xenograft volume. From the 12th treatment day onwards, an increase in the xenograft volumes in mice treated with either NPB or Olaparib was observed. Olaparib-treated mice reached the humane endpoint and were euthanized on days19 ($n = 1$), 20 ($n = 3$), 24 ($n = 1$), and 25 ($n = 1$) days; NPB-treated mice reached the humane endpoint later than animals in the Olaparib group and were euthanized on days 20 ($n = 2$), 24 ($n = 1$), and 25 ($n = 3$) day. The combined NPB-Olaparib treatment exhibited a durable regression in xenograft volume and a significantly more potent effect compared with either NPB or Olaparib-treated AFC-generated xenografts. No xenograft in the combined NPB-Olaparib group reached the humane endpoint during the 21-day treatment period. The mice reached the humane endpoint and were euthanized on the 29th ($n = 1$), 30th ($n = 1$), and 36th ($n = 2$) day. In addition, the xenograft in 2/6 animals treated with the combination exhibited a complete response. The median survival of animals in the control group was 16 days, Olaparib 20 days, NPB 24.5 days, and combined NPB-Olaparib 36 days as demonstrated using Kaplan–Meier survival analysis (Fig. 5E). No significant changes in animal body weight were observed during the treatment period (Fig. 5D).

The best average response to the Olaparib, NPB, and combined NPB-Olaparib was examined utilizing Waterfall plots of response as represented in Fig. 5F. After that, the modified response evaluation criteria in solid cancers (mRECIST) categorized the response into progressive disease (PD), stable disease (SD), partial response (PR), or complete response (CR) according to the best response and the best average response to the treatment[55]. All 6/6 vehicle-treated xenografts were categorized as PD, 2/6 Olaparib and 5/6 NPB-treated xenografts were categorized as PR and 3/6 Olaparib and 1/6 NPB-treated xenografts were categorized as SD. The response in 1/6 Olaparib-treated xenografts remained progressive (PD). In contrast, the response in 4/6 xenografts treated with the combination was sorted as PR and the remaining 2/6 showed a CR.

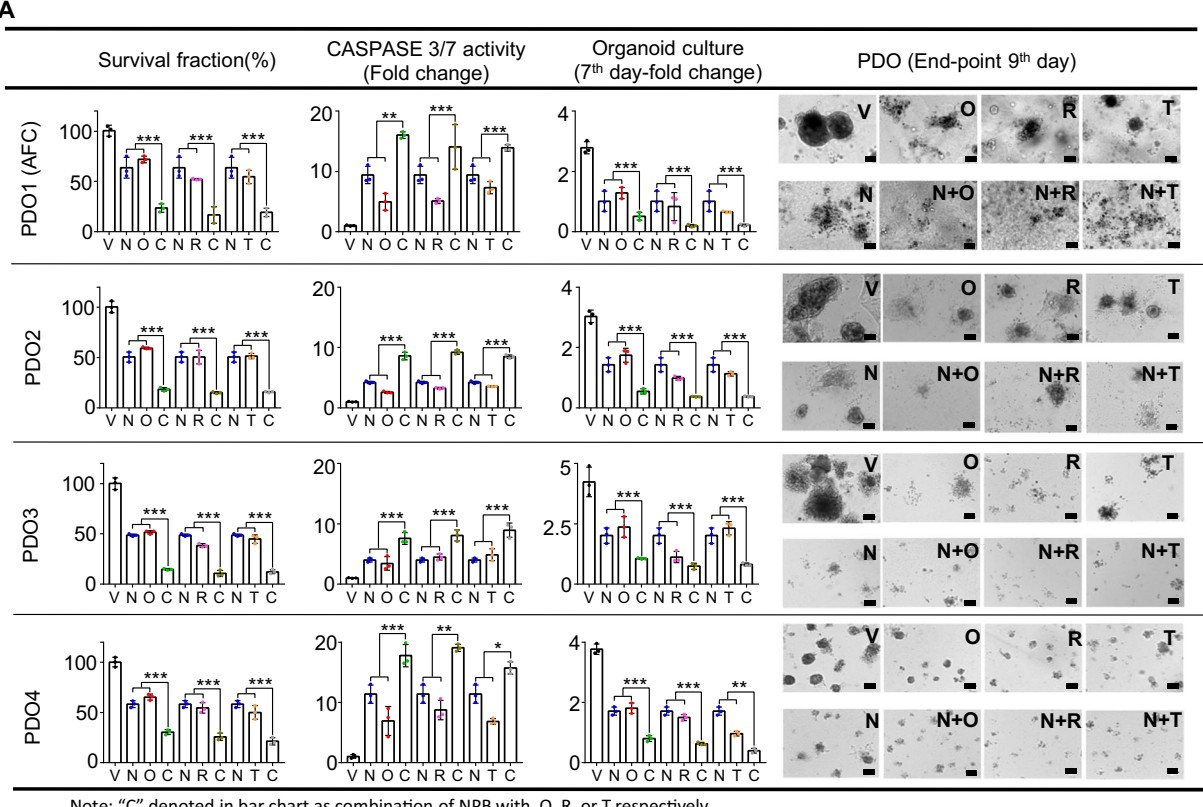

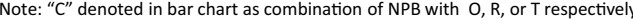

Note: "C" denoted in bar chart as combination of NPB with O, R, or T respectively.

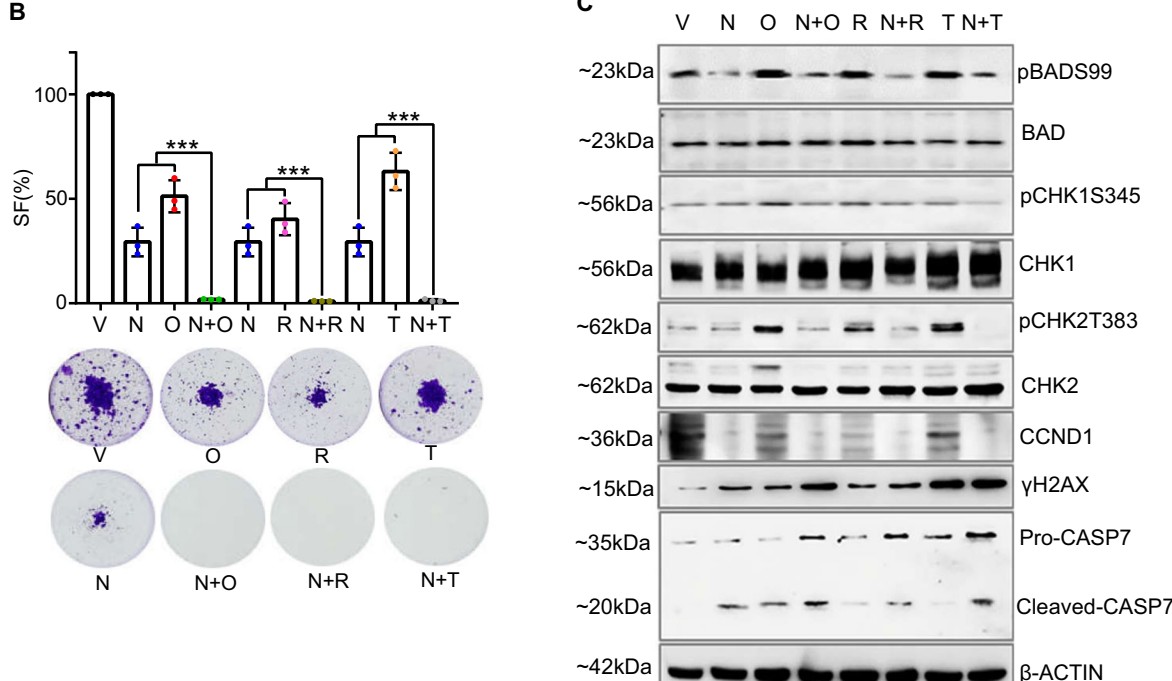

## Discussion

PARPis are approved as a last-line therapy for patients with BRCA-deficient EOC. Numerous studies have also advocated the clinical utility of PARPis in patients with BRCA-proficient and/or platinum-resistant recurrent EOC. However, the suboptimal efficacy of PARPi as a monotherapy in BRCA-proficient, pre-existing platinum-resistant, and acquired PARPi resistance in EOC represents major clinical challenges[71,72]. There is therefore an urgent need to design rational combination therapies that improve the efficacy of monotherapy and expand the potential patient populations that might benefit from PARPis. Aberrant activation of the RAS/MAPK pathway is reported in 25% of high-grade serous ovarian cancer (HGSOC) which accounts for 70% of EOCs and PI3K/AKT/mTOR signaling is increased in ~40–50%

**Fig. 4 NPB synergizes with PARP inhibitors to decrease cell survival in patient-derived EOC organoids. A** NPB and PARP inhibitor combination efficacy in PDOs: effect of NPB (N), PARP inhibitors (Olaparib (O), Rucaparib (R), Talazoparib (T)) or combination treatment for 9 days in 4 EOC patient-derived organoids (PDOs), namely PDO1-4 were evaluated using the ApoTox-Glo Triplex Assay Kit for survival fraction (SF) and caspase3/7 activity. Organoid growth was determined by counting organoids with diameters >100 µm number on the 7th day of culture and images of organoids growth on the 9th day were shown, Scale bars, 100 µm. **B** Foci formation (PDO1): survival fraction (SF) of ascitic fluid cancer (AFC) cells (from PDO1) treated with NPB (N), Olaparib (O), Rucaparib (R), and Talazoparib (T) or combinations for foci formation. Foci were stained with 0.2% crystal violet and quantified using a microplate reader (Tecan Spark®, Switzerland) by detecting the absorbance of soluble crystal violet. **C** Western blot (PDO1): western blot analysis of ascitic fluid cancer (AFC) cell-derived organoids (also designated as PDO-1) in 3D Matrigel after treatment with NPB (N) alone or in combination with Olaparib (O), Rucaparib (R) or Talazoparib (T). Organoids were harvested after depolymerization of Matrigel using Cultrex® organoid harvesting solution (R&D systems, US) and organoid pellets were resuspended in RIPA buffer plus protease inhibitors. Extracts were run on an SDS-PAGE and immunoblotted as described in "Methods". β-ACTIN (ACTB) was used as input control for cell lysate. The sizes of detected protein bands in kDa are shown on the left side. Treatment concentration of NPB, 5 µM; Olaparib, 10 µM; Rucaparib, 5 µM; and Talazoparib, 5 µM. Columns are mean of triplicate experiments; bars, ±SD. *$p < 0.05$, **$p < 0.01$, ***$p < 0.001$.

of EOC[73]. In vitro experimental evidence indicates that PARP-inhibitor-induced activation of the RAS/MEK/ERK and PI3K/AKT/mTOR pathway limit the efficacy of PARPis resulting in the development of PARPi resistance in EOC[31,74]. Hence, increased RAS/MEK/ERK and PI3K/AKT/mTOR activities increase BAD phosphorylation providing an actionable vulnerability, regardless of mutational status. *BRCA* deficiency/mutation activates the AKT pathway with consequent BAD phosphorylation (64, 65). The observation in this study that BRCA-deficient cell lines exhibited significantly higher levels of pBADS99 compared to BRCA-proficient cell lines supports this notion. Furthermore, BCL2 has been reported to regulate DSB repair via its interaction with BRCA1 (70) and this interaction can be disrupted by BH3-only BCL2 family members (71). Previous studies have demonstrated that the proapoptotic BCL-2 family members, Bax and Bid, can promote cell death by inhibiting HR DNA repair independent of apoptosis[75]. Therefore, it is reasonable to suggest that inhibition of BAD phosphorylation abrogates HR and herein NPB treatment and BADS99 mutation was indeed demonstrated to suppress HR. Moreover, mutations in *TP53* are a near-universal characteristic of HGSOC (97%). DNA damage has been reported to enhance both TP53 expression and poly (ADP-ribose) polymerase (PARP) activation[76,77]. In a clinical trial (NCT00753545)[78] a patient population with EOC containing *TP53* mutations attained a significant OS benefit from Olaparib treatment regardless of BRCA mutant status. Indeed, it is reported that TP53 and BAD protein co-operatively promote apoptosis subsequent to DNA damage in cancer cells[79]. Along with Olaparib[80,81], Rucaparib[82] and Talazoparib[83] were also approved for the treatment of ovarian cancer regardless of BRCA-mutational status. Due to the limited clinical genomic annotation, potential correlations between the prognostic value of pBADS99 and BRCA status were not determined in this study. However, synergistic interactions between inhibition of BADS99 phosphorylation and PARPis in EOC including either BRCA-deficient or proficient and/or platinum-resistant models were demonstrated.

Due to the varying pharmacological features of different molecules impacting similar pathways or processes, and the necessary differences in their mechanisms, there exists a fundamental rationale for a diverse response in efficacy. Specifically, Olaparib is reported to trap PARP1 and 2[84], while Rucaparib traps PARP1, PARP2, and PARP3, with higher affinity to PARP1[85,86], and Talazoparib specifically traps only PARP1[87]. In addition, PARP inhibitors differ in their off-target effects. These off-target effects may also contribute to the efficacy attributed to PARP inhibition. Therefore, when the efficacy of different PARP inhibitors was examined, a consistent concentration and temporal endpoint of the treatment to the drug response was utilized. As shown in Supplementary Fig. 1G, Rucaparib and Talazoparib exhibited greater efficacy in EOC cell lines; however, no

significant correlation with the levels of pBADS99 in EOC cells was observed compared to Olaparib treatment after 6-days. However, a significant correlation between pBADS99 and IC$_{50}$ for Rucaparib or Talazoparib was observed at 48 and 72 h respectively. Olaparib was the first FDA-approved PARPi and therefore the experimental approaches in this manuscript were largely performed using Olaparib. NPB as a monotherapy exhibited superior efficacy to Olaparib in xenograft models of platinum-sensitive and resistant EOC. Furthermore, DNA damage consequent to PARP inhibition was facilitated by inhibition of BAD phosphorylation. Importantly, the synergism between NPB and PARP inhibition was independent of *BRCA, P53, PI3K, PTEN, or RAS* mutational status. Hence, concomitant inhibition of BAD phosphorylation may be effective to improve sensitivity to PARP inhibition in EOC including in the presence of secondary BRCA1/2 mutations that restore BRCA1/2 function[71,88] (Supplementary Fig. 10).

Based on data from clinical trials, grade 3 or greater toxicities occurred in approximately 35–56% of patients treated with the approved PARPis. DNA damaging agents such as platinum, topoisomerase inhibitors, and Gemcitabine have all been investigated in combination with PARPis in EOC. The combination enhances the effects of cytotoxic chemotherapies via the impairment of DNA repair processes. A major concern for a combination of PARPis and cytotoxic chemotherapy is its narrow therapeutic window: PARPi enhancement of cytotoxicity in normal cells led to lowered doses of combinational agents at the maximally tolerated dose (MTD)[6,89–91]. Therefore, some trials of the combination of PARP inhibitors with cytotoxic chemotherapy have failed to demonstrate efficacy[79,92,93]. Combinations of PARPis with RAS/MEK/ERK and PI3K/AKT/mTOR inhibitors also resulted in significant toxicities (tabulated in Supplementary Table 7). Preliminary data have indicated that inhibition of BADS99 phosphorylation by NPB is well tolerated[44]. NPB is not predicted to be genotoxic by stringent in silico analyses[94] (Supplementary Fig. 9). Increased expression of γ-H2AX following NPB treatment herein was therefore presumably due to caspase-dependent DNA damage as reported for other agents targeting BCL-2 family members such as Venetoclax (ABT-199)[64,65]. Furthermore, the combination of BADS99 and PARP inhibition appeared to be well tolerated herein with no significant differences in animal weight nor behavior observed[54,95]. Hence, the combined inhibition of PARP and BADS99 phosphorylation may provide effective PARPi dose reduction with less toxicity yet with improved efficacy.

The majority of EOC are initially platinum-sensitive but eventually develop resistance[96]. Outcomes in platinum-resistant EOC are extremely poor, with a median survival of only 12 months[97]. PARPis have also recently been approved for patients with platinum-refractory EOC or recurrent EOC as the first- or second-line maintenance settings[3]. Unfortunately, almost

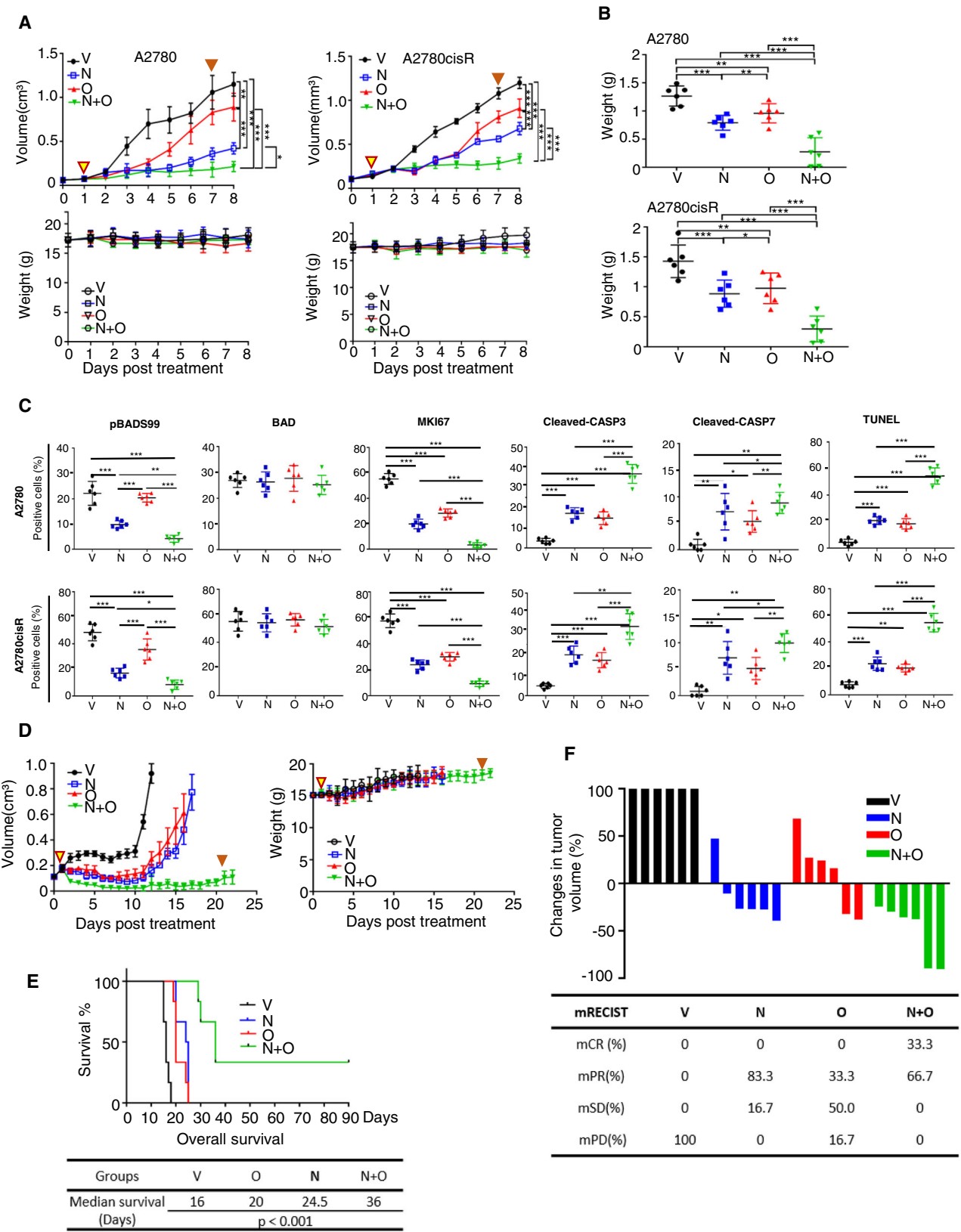

all patients will also eventually acquire resistance to PARPi treatment. It was recently reported that NPB suppressed cisplatin-induced BADS99 phosphorylation, as well as the elevated level of BADS99 phosphorylation in cisplatin-resistant ovarian cancer cells[54]. Inhibition of BADS99 phosphorylation partially overcame acquired cisplatin resistance through the promotion of apoptosis and inhibition of CSC-like behavior[54]. Herein, the combined NPB-Olaparib regimen was demonstrated to be efficacious in a platinum-resistant PDX model, in which one-third of xenograft-bearing mice achieved a durable CR without relapse despite

**Fig. 5 NPB and Olaparib synergize to suppress the growth of cisplatin-resistant EOC xenografts. A** Xenograft: A2780 or A2780 cisplatin-resistant $(5 \times 10^6)$ cells were injected subcutaneously into the flank of 5-week-old BALB/c athymic mice, respectively. When the xenograft reached 100-150 m$^3$, in each cell line group, the mice were randomized into the four indicated treatment groups ($n = 6$). Mice were treated daily with vehicle (V), 20 mg/kg NPB (N), 50 mg/kg Olaparib (O), or a combined NPB-Olaparib by i.p. injection. Xenograft growth was monitored daily by the measurement of the xenograft volume. The statistical change between treatment groups was analyzed using a one-way ANOVA followed by a Tukey's multiple comparison test on endpoint xenograft volume. Mean animal weights of each treatment group are indicated. Animal weight was monitored daily. The yellow triangle points to the start of the treatment. The brown triangle points to the end of the treatment. Results represent the mean ± SEM of six animals. **B** Mice were treated daily with vehicle (V), 20 mg/kg NPB (N), 50 mg/kg Olaparib (O), or a combined NPB-Olaparib by i.p. injection. Xenograft weight: mean xenograft weight of each treatment group after sacrifice on the 8th day. Results represent the mean ± SEM of six animals. **C** Immunohistochemistry evaluation: pBADS99, BAD, KI67, cleaved-CASP3/7, and TdT-mediated dUTP Nick-End Labeling (TUNEL) positivity was assessed in resected xenografts treated with vehicle (V), 20 mg/kg NPB (N), 50 mg/kg Olaparib (O) or a combined NPB-Olaparib by i.p. injection using immunohistochemistry (IHC) as described in the methods. Quantification of positive staining cells from the indicated treatment group for pBADS99, BAD, KI67, cleaved-CASP3/7, and TUNEL are shown in individual value plots ($n = 6$). **D** Patient-derived xenograft: confluent cultured cancer $(5 \times 10^6)$ cells from the patient's ascites were injected subcutaneously into the flank of 5-week-old BALB/c athymic mice. When xenografts reached 150–200 m$^3$, the mice were randomized into the four indicated treatment groups ($n = 6$). Mice were treated daily with vehicle (V), 20 mg/kg NPB (N), 50 mg/kg Olaparib (O) or combined NPB-Olaparib by i.p. injection for 21 days. Xenograft growth was monitored daily by measurement of the xenograft volume until the humane endpoint. Below: the mean animal weights of each treatment group are indicated. Animal weight was monitored daily. Results represent the mean ± SEM of six animals. **E** Kaplan–Meier curve: Kaplan–Meier survival curves showed median survival to the humane endpoint and surrogate survival rate in the vehicle (V), NPB (N), Olaparib (O), or a combined NPB-Olaparib-treated group. Six animals in each group. Log-rank test $\chi2 = 75.0054$. $p < 0.001$. **F** The drug response evaluation: waterfall plot of responses to the vehicle (V), NPB (N), Olaparib (O), or a combined NPB-Olaparib are described by xenograft volume changes (%) from Best Average Response (The minimum value of the average of ΔVolt for $t \geq 10$ d); each bar represents an individual PDX. The criteria for response (mRECIST) was used to make response calls (as described in "Methods"). mRECIST were defined as followed: complete response (mCR): best response < −95% and best avg response < −40%; partial response (mPR): best response < −50% and best avg response < −20%; stable disease (mSD), best response < 35% and best avg response <30%; progressive disease (mPD), not otherwise categorized. A total of four treatment groups were tested in 6 PDXs respectively. Xenograft studies were performed according to the animal research: reporting in vivo experiments (ARRIVE) 2.0 guidelines. Columns or points are the mean of triplicate experiments; bars, ±SEM. *symbols above the bar of different groups represent the statistical significance compared to vehicle. *$p < 0.05$, **$p < 0.01$, ***$p < 0.001$ .

cessation of treatment. Hence, a rational and efficacious combination strategy involving combined inhibition of PARP and BAD phosphorylation may ameliorate the prognosis of recurrent EOC patients.

## Data availability

The data that support the findings of this study are available from the corresponding authors upon reasonable request.

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

## Acknowledgements

This work was also supported by the Shenzhen Key Laboratory of Innovative Onco-therapeutics (ZDSYS20200820165400003) (Shenzhen Science and Technology Innovation Commission), China; Shenzhen Development and Reform Commission Subject Construction Project ([2017]1434), China; Overseas Research Cooperation Project (HW2020008) (Tsinghua Shenzhen International Graduate School), China; Universities Stable Funding Key Projects (WDZC20200821150704001); The Shenzhen Bay Laboratory, Oncotherapeutics (21310031), China; TBSI Faculty Start-up Funds, China; the Guangdong Basic and Applied Basic Research Foundation (2019A1515110970), China; the National Natural Science Foundation of China (Grant No. 82172618); DBT-NER, and Vision Group on Science and Technology (CESEM), Government of Karnataka.

## Author contributions

X.Z., P.E.L., and V.P. designed research; X.Z., P.H., and V.P. performed in vitro assays; X.Z. and S.C. performed in vivo assays; B. synthesized NPB; L.W., H.D., X.Z., and P.H. collected specimens and analyzed patient data; X.Z., P.H., T.Z. L.M., P.E.L., and V.P. analyzed the data; X.Z., P.E.L., and V.P. wrote the manuscript. All authors have read and approved the manuscript for publication.

## Competing interests

The authors declare the following competing interests: V.P., B., and P.E.L. are listed as inventors on a patent application and derivatives thereof for NPB which is used in this work (WO/2019/194520). P.E.L. is an equity holder in Sinotar Pharmaceuticals Ltd which currently holds the license for this patent. All other authors have no competing interests to declare.
