## [Peer Review File · Communications Medicine]

Reviewers' comments:

Reviewer #1 (Remarks to the Author):

In this manuscript by Zhang et al, authors reported phospho-BAD (pBAD-S99) as a potential therapeutic target to enhance therapeutic effects of PARP inhibitors (PARPis). They provided evidence that higher immunoreactivity scores (IRS) for pBAD-S99 are associated with poor outcome in ovarian cancer patients. pBAD-S99 expression in cancer cell lines also positively correlated with IC50 of olaparib in these cells. A small molecular inhibitor of BAD-S99 phosphorylation NPB attenuated BAD-S99 phosphorylation enhances sensitivity to PARPis by enhancing cytotoxic effects of PARPis in cell lines, 3D culture, patient-derived organoids, and patient-derived xenografts.

Overall, authors have presented results of well-controlled studies with a few exceptions outlined below. Additional mechanistic studies will strengthen the impact of this manuscript.

Comments:

1. Given ref#40, to improve the novelty, it would be important to state how current studies are different from studies reported in ref#40 for ovarian cancer.
2. Although specificity of NPB against S99 phosphorylation has been shown in several cancer cell lines in previously published studies, it would be important to show that these results are reproducible in patient-derived organoids.
3. Figure 3A. Overlap of green and red signals in 3D organoids suggest insufficient resolution to separate live (green) and dead (red) cells in the image analysis. This insufficient signal separation may affect determination of surviving fraction. This analysis should be performed at higher resolution and preferably using confocal microscopy to separate potential crossover signals from Z-axis (ie, dead cells on top of live cells in 3D organoid).
4. Figure 3C, it would be nice to present the results by separately grouping early and late apoptotic groups. In current format, it is not apparent that there is a difference between N vs N+O for early apoptosis cell numbers although text claims that it is.
5. Figure 3D, it should be quantified and presented with statistics.
6. Figure 3E, it is interesting that NPB treatment attenuated PARPi-induced CHEK1 and CHEK2 phosphorylation. This inhibition likely affect DNA damage response signaling. What is the potential mechanism? Could this be the mechanism contributing the drug synergies between NPB and PARPis?
7. Figure 3F and SI 5B, it is interesting to see that NPB treatment alone increases gamma-H2AX IF. Is NPB treatment inducing genotoxicity?
8. What is a potential explanation for good correlation between pBAD-S99 phosphorylation and IC50 for olaparib but not with rucaparib or talazoparib?

Formatting errors:

Ref 55

Figure 2 Y-axis

Figure 4B Y-axis

SI 4D: ELISA reader should be replaced with plate reader

Reviewer #2 (Remarks to the Author): see attachment

Reviewer #3 (Remarks to the Author):

Rationale:

Poly (ADP-ribose) polymerase inhibitors (PARPi) are approved as maintenance therapy for the treatment of primary and recurrent epithelial ovarian cancer (EOC). New combination treatments are needed to improve responses to PARPi. Here, a combination of PARPi and a small molecule (NPB), which inhibits phosphorylation of BAD, is investigated.

The rationale for targeting BAD is that it is a key factor involved in apoptosis and it is downstream of RAS/MEK/ERK and PI3K/AKT/mTOR pathways that are implicated in sensitivity to PARPi.

Strengths

1. The main strengths of this manuscript are the significant amount of data to support synergy between NPB and the PARPi (olaparib, rucaparib, and talazoparib) in vitro and in vivo.
2. Confirmation of an association of pBADS99 with poor PFS and OS, consistent with previous reports.
3. Association of pBADS99 with relative resistance to the PARPi olaparib.
4. The use of patient-derived cancer cells and ascetic fluid cancer cells.

Weaknesses

1. The non-cancer ovarian (NCO) tissue may not be the appropriate normal comparison for EOC, which is thought to arise from fallopian tube epithelium.
2. There is concern about the EOC cell lines. Of the EOC cell lines, only 3 of the 8 represented had TP53 mutations. These EOC cell lines do not represent the most common type of EOC, high grade serous histology (Domcke, Nat Comm 2013). In addition, A2780 and A2780cp do not have BRCA mutations in the CCLE.
 - a. How were the BRCA-deficient commercial cell lines validated?
3. PARPi have different PARP trapping capabilities that are associated with potency. It is difficult to detect these differences based on how the data was presented in logarithmic form (Fig 2).
4. Line 24: The statement, "PARP inhibitors are the last line of therapy for recurrent EOC" is misleading. PARP inhibitors are used in primary/frontline and recurrent treatment for EOC. There are many other options for last line of therapy.
5. Line 85 and Lines 336-337: Although 1/3 of tumor-bearing mice in a single platinum-resistant PDX model achieved a durable CR, it is overstated to claim that the combination of NPB with PARPi would be "may be curative."
6. The results are primarily descriptive and are not mechanistic.

Zhang et al.. Pandey. Combined inhibition of PARP and BAD-Ser99 phosphorylation ablates recurrent ovarian carcinoma

Zhang et al. explore targeting of PARP and BAD-Ser99 as a potential therapeutic for ovarian cancer (EOC). This manuscript addresses important clinical problems in ovarian cancer therapy—those of resistance to platinum first-line therapies, and challenges of toxicity and resistance to PARPi last-line therapies. The authors state that mechanisms of therapeutic resistance are RAS/MAPK and PI3K/AKT/mTOR pathway activation that interfere with platinum- and PARPi-mediated tumor cell death. The authors propose to stimulate downstream modulators of apoptosis to circumvent upstream resistance mechanisms. They suggest that inhibition of BAD-S99 phosphorylation will increase the therapeutic effect of PARPi's.

The authors provide convincing rationale for these studies. They demonstrate that BAD-Ser99 is associated with poor survival of EOC patients, BRCA-deficiency and decreased platinum and PARPi sensitivity. They developed a BAD-Ser99 inhibitor (NPB) that forms the basis of these studies. They extensively test NPB in combination with 3 different PARPi's to assess effects on DNA damage and apoptosis. They see concordant results with a number of different models (EOC cell line panel, patient-derived-organoids—PDO's, and patient-derived-xenografts—PDX's) that support their conclusions that combining inhibitors to PARP and BADS99 inhibit EOC growth.

This is a very good study and I have a few suggestions for improvements. Points 1, 3, 4, 6 and 10 are the most critical points that need to be addressed:

1. P-BAD-S99 levels are associated with poor survival and also with BRCA-deficiency. Can the authors do a multivariate analysis to ascertain if the prognostic value of P-BAD-S99 is dependent on BRCA-deficiency and/or all other significant markers of outcome? This will give insight into the genetic interaction between P-BAD-S99 and BRCA and other markers.
2. Along the same lines as point #1, what do the authors speculate is the relationship between BRCA-deficiency and phosphorylation of BAD? Do the authors think that P-BAD is upstream, downstream or independent of BRCA? For example, as a downstream model, does BRCA-deficiency lead to activation of BAD kinases? Some discussion on this may be interesting.
3. How specific is the NPB-effect on BAD? The authors show that NPB decreases P-BAD-S99 levels, which is associated with increased apoptosis. Given that BAD phosphorylation also modulates non-apoptotic pathways, the authors should demonstrate the specificity of NPB for BAD in the presented assays here. For example, can the authors test whether loss-of-function (BAD knock-out) prevents NPB-mediated apoptosis? Does loss-of-function BAD knock-out prevent the synergy between NPB and PARPi with respect to growth effects/markers of proliferation apoptosis, etc.?
4. Along the same lines as point #3, what is the mechanism by which NPB inhibits phosphorylation of BAD? Is this via blocking accessibility of BAD-kinases to S99, or via inhibiting BAD-kinases directly, or stimulating BAD-phosphatase activity? Some explanation of this is needed.

5. Again, along the same lines as point #3, does NPB increase the BAD-specific apoptotic pathway? One would expect increased BAD:Bcl-2-like interactions. Similarly, the authors demonstrate that mutant BAD-S99A decreases cell survival. The authors should show that this tagged construct is appropriately located within the cell.
6. Controls are needed for some of the immunohistochemistry data. Importantly, the specificities of the anti-P-BAD-S99 and anti-BAD antibodies are not clear. This is critically important. Under these IHC conditions, the authors should demonstrate a loss of signal in BAD knock-out cells/tissue, or loss of signal in the presence of competing BAD-peptides. Also, in Fig. SI 1, are the NCO samples under lower magnification than the EOC?
7. It is not clear how the authors concluded that P-BAD is associated with increased FIGO scores (Fig. 1)
8. Can the authors speculate on the underlying mechanisms causing differential effects with the different PARPi's? For example (a) the IC50 for Olaparib was associated with P-BAD-S99 status, but the other PARPi's were not; and (b) N+O increases apoptosis and does not affect cell cycle, whereas N+T and N+R have clear G2/M arrest. Some discussion around this would be interesting.
9. The authors suggest that combining NPB with PARPi's would be less toxic than combining PARPi's with RAS or PI3K inhibitors. It would have been great to compare these different treatments directly for in vivo toxicities relative to tumor response. This would have enabled direct "benchmarking" to address the authors' comments on lines 322 regarding already documented toxicities. Understandably, this would be considerable additional experimentation and is not critical.
10. All representative western blots should have accompanying quantitation of biological replicates with appropriate statistics. This is critical as the authors make quantitative conclusions from this data. (For example, Fig. 3E and Fig. 4C). In Fig. 4C, the authors conclude that caspase 7 cleavage is elevated in response to N/O vs N or O alone. This is not convincing on the representative western blot.
11. Figure 2, the Y-axis for SF and DRC is "off".
12. In Figure 5A xenograft experiment, how was the statistical analysis done? Are the differences based on the endpoint tumor volume?

Reviewer #1 (Remarks to the Author):

The constructive comments of the reviewer are appreciated. We have revised the manuscript accordingly.

1. Given ref#40, to improve the novelty, it would be important to state how current studies are different from studies reported in ref#40 for ovarian cancer.

Response: We acknowledge the reviewer's concern. However, the experimental approach, conclusions and impact of this manuscript is significantly different to the previously published article Ref 40.

In Ref#40 (Ref 41 in the revised manuscript) entitled "BAD phosphorylation determines ovarian cancer chemosensitivity and patient survival", the authors aimed to explore and characterize the BAD mediated apoptosis pathway as a determinant of chemo-sensitivity and patient survival using clinical specimens and a panel of cancer cell lines based on expression studies. The outcome of the findings suggested that BAD phosphorylation modulates ovarian cancer chemosensitivity and overall survival of patients. In agreement, this study also concluded that the BAD pathway exhibits clinical relevance as a biomarker of therapeutic response, patient survival, and as a promising therapeutic target.

However, the focus of this study was a functional and novel combination strategy using drugs (PARPis) which were not FDA approved for use in ovarian cancer in 2011 when the previous paper was published (Olaparib was approved by the FDA for use in ovarian cancer in 2017) and a novel BAD phosphorylation inhibitor only reported in 2018. Hence, neither of these drugs were even alluded to in the previously published paper. The major focus of this study was the demonstration that combined inhibition of BADSer99 and PARP in recurrent EOC synergistically increased apoptotic cell death and achieved a complete response in a percentage patient-derived xenografts. Thus, the emphasis of this manuscript is that a novel drug combination may provide a potential curative strategy for recurrent EOC.

In accordance with the reviewer's suggestion, to emphasize the novelty of the current study, we have incorporated and highlighted new details in the introduction section of the revised manuscript as described below:

"Elevated BAD phosphorylation has been reported to be positively correlated with acquired cisplatin/paclitaxel resistance and poorer prognosis of EOC patients (26, 41-43). A small molecule (NPB) which specifically inhibits phosphorylation at hBADS99 and promotes apoptotic cell death in cancer cells has been recently reported (44). Herein, the synergistic combination of NPB with PARPis (Olaparib, Rucaparib, and Talazoparib) is exemplified as a possible efficacious regimen for EOC."

2. Although specificity of NPB against S99 phosphorylation has been shown in several

cancer cell lines in previously published studies, it would be important to show that these results are reproducible in patient-derived organoids.

Response: As per reviewer's request, we have confirmed the functional specificity of NPB for BAD by assessing the effect of NPB exposure after *siRNA*-mediated depletion of BAD expression in CAOV2 cell and PDO1 (SI 3C&D, SI 6G). We have also now incorporated new data in the results section of the revised manuscript demonstrating the specificity and efficacy of NPB against BADS99 phosphorylation in carcinoma cell line (SI 3C&D) and patient derived organoids (SI 6G). Additionally, we have also revised the western blots in Figure 4C and the quantification analysis summarized in the supplementary data - S6F, as described below:

Line 145-157:

“To confirm the functional specificity of NPB directed to BAD, CAOV2 cells were treated with NPB after *siRNA* mediated depletion of BAD expression (SI 3C). Transient-transfection of CAOV2 cells with *siRNA* directed to the BAD transcript decreased BAD expression and decreased levels of pBADS99 compared to control cells (transfected with scrambled oligo) as observed by WB analysis (SI 3C). No significant changes in caspase3/7 activity nor cell viability were observed upon *siRNA*- mediated depletion of BAD in CAOV2 cells (SI 3C) and as previously reported in various carcinoma cells (44). NPB treatment of control transfected cells decreased phosphorylation of BADS99 compared to vehicle-treated CAOV2 cells. Consistently, exposure of CAOV2 control cells to NPB increased caspase3/7 activity and decreased cell viability compared to vehicle-treated cells. In contrast, NPB did not affect caspase 3/7 activity nor cell viability in PDO1 with depleted expression of BAD. Furthermore, no synergistic effect was observed between NPB and PARPi in CAOV2 cells with *siRNA* mediated depletion of *BAD* compared to control transfected CAOV2 evaluated by a total cell number assay (SI 3D).”

Line 244-254:

“To confirm the functional specificity of NPB to BAD, the effect of NPB exposure after *siRNA*-mediated depletion of BAD expression was examined in PDO1 (SI 6G). Transient-transfection of PDO1 with *siRNA* directed to the BAD transcript decreased BAD expression and decreased levels of pBADS99 compared to control PDO1 (transfected with scrambled oligo) as observed by WB analysis (SI 6G). No significant changes in cell viability nor caspase 3/7 activity were observed upon *siRNA*-mediated depletion of BAD in PDO1, and as previously reported for various carcinoma cells (44). As observed in SI 6G, NPB treatment of the control transfected PDO1 decreased phosphorylation of BADS99 compared to control transfected PDO1 treated with vehicle. Consistently, exposure of control transfected PDO1 to NPB decreased cell viability and increased caspase 3/7 activity compared to vehicle-exposed PDO1. In contrast, NPB did not affect caspase 3/7 activity nor cell viability in PDO1 with depleted expression of BAD (SI 6G).”

Supplementary information 3.

C. CAOV2/ CAOV2-siRNA BAD

D. Combination efficacy of NPB and PARP inhibitors in CAOV2/CAOV2-siRNA BAD

SI 3C: *siRNA*-mediated depletion of BAD expression abrogated the effect of NPB in CAOV2 cells. WB analysis was used to assess the levels of pBADS99 and BAD protein in CAOV2 cells after treatment with 10μM NPB. Depletion of BAD expression was achieved using transient transfection of *siRNA* directed to the BAD transcript as previously described [1]. Soluble whole-cell extracts were run on an SDS/PAGE gel and were immunoblotted as described in Materials and Methods. β-ACTIN was used as input control. Caspase 3/7 activity and cell viability were evaluated using the ApoTox-Glo Triplex Assay Kit.

SI 3D: Dose-response curves for CAOV2 and CAOV2 cells with *siRNA* mediated depletion of BAD treated with the indicated concentration of PARPis with or without 1uM NPB in a total cell number assay. Arrow indicates fold reduction in respective PARP inhibitor IC₅₀ in the presence of NPB (n=3).

SI 6G. PDO1/PDO1-siRNA BAD

SI 6G: *siRNA*-mediated depletion of BAD expression abrogates the effect of NPB in PDO1: WB analysis was used to assess the levels of pBAD599 and BAD protein in PDO1 after treatment with NPB. Depletion of BAD expression was achieved using transient transfection of *siRNA* directed to the BAD transcript as previously described [2] (Figure 6A&B). Organoids were harvested after depolymerization of Matrigel using Cultrex® organoid harvesting solution (R&D systems, US) and organoid-pellets were resuspended in RIPA buffer plus protease inhibitors. Extracts were run on an SDS-PAGE and immunoblotted as described in materials and methods. β-ACTIN was used as input control. Cell viability and Caspase 3/7 activities were evaluated using the ApoTox-Glo Triplex Assay Kit.

Supplementary text (F): **siRNAs and transfection:** The target-directed *siRNAs* (*shRNA-BAD1*, 5'-GCUCCGCACCAUGAGUGACGAGUUU-3' and *shRNA-BAD2*, 5'-AAACUCGUCACUCAUCCUCCGGAGC-3') and the negative control *siRNA*, which was used for normalization, were synthesized by GENEWIZ (Suzhou, China). After seeding and adherence for 24h, cells were washed with PBS and transfected with the *siRNAs* for 24h using FuGENE HD (Promega, US) according to the manufacturer's instructions. The *siRNAs* were transfected either separately with constant-ratio diluted PARP inhibitors (single treatments) or with combinations of constant-ratio diluted PARP inhibitors and NPB(1μM).

Figure 4C. Western Blot (PDO1)

Figure 4C: Western blot analysis of ascitic fluid cancer (AFC) cell derived organoids (also designated as PDO-1) in 3D Matrigel after treatment with NPB (N) alone or in combination with Olaparib (O), Rucaparib (R) or Talazoparib (T). Organoids were harvested after depolymerization of Matrigel using Cultrex® organoid harvesting solution (R&D systems, US) and organoid-pellets were resuspended in RIPA buffer plus protease inhibitors. Extracts were run on an SDS-PAGE and immunoblotted as described in materials and methods. β-ACTIN (ACTB) was used as input control for cell lysate. The sizes of detected protein bands in kDa are shown on the left side.

SI 6F. Western Blot quantification (PDO1 Patient derived cells)

SI 6F: Densitometric analysis of western blots in Figure 4C. Densitometric analysis of protein

blots was determined using ImageJ software (<https://imagej.nih.gov/ij/>). Statistical changes were assessed by using an unpaired *two-tailed Student's t test* ($P < 0.05$ was considered as significant), GraphPad Prism. Columns are mean of triplicate experiments; bars, \pm SD. * $p < 0.05$, ** $p < 0.01$, *** $p < 0.001$

3. Figure 3A. Overlap of green and red signals in 3D organoids suggest insufficient resolution to separate live (green) and dead (red) cells in the image analysis. This insufficient signal separation may affect determination of surviving fraction. This analysis should be performed at higher resolution and preferably using confocal microscopy to separate potential crossover signals from Z-axis (ie, dead cells on top of live cells in 3D organoid).

Response: We agree with the reviewer's observation that live and dead cell imaging herein is not an optimal quantification approach due to the inadequate signal separation. As per the reviewer's suggestion, we therefore performed multiple biological repeats and also performed live and dead imaging analysis using confocal microscopy. However, Z-axis based confocal microscopy of 3D colonies also did not significantly improve the results as shown by the representative images from both microscopy techniques below.

Microscopic visualization of Calcein-AM (green) stained colonies (live) and BOBO-3 Iodide (red) stained cell debris (dead) generated by CAOV2 cells cultured in 3D Matrigel after exposure to NPB (N), Olaparib(O), and combination (n=3). Scale bars, 100 μ m.

To address this issue, we compensated the findings with multiple biological experimental repeats using an additional biochemical assay. The quantification of the survival fraction was additionally determined by the ApoTox-Glo Triplex Assay Kit to compare with the live and dead imaging-based analysis. The results of the ApoTox-Glo Triplex Assay are summarized in Figure 3B&SI 4A and details are provided in the figure legend. To be in accordance with the reviewer

concerns, we have modified the representation of the Live/Dead analysis images by enhancing the resolution and size of the images for better illustration in Figure 3A and have incorporated the methodology and results interpretation in the revised manuscript as described below:

Line 164-167:

“To determine the survival fraction (SF), ApoTox-Glo Triplex assays were used to evaluate live-cell protease activity (cell viability), and caspase 3/7 activity (apoptosis) of the colonies cultured in 3D Matrigel. NPB in combination with Olaparib significantly decreased cell viability and increased caspase 3/7 activity compared to NPB or Olaparib treated CAOV2 colonies (Figure 3B, SI 4A).”

Figure 3B. 3D Matrigel growth

Figure 3B: Survival fraction (SF) was evaluated using the ApoTox-Glo Triplex Assay Kit in CAOV2 cells treated with NPB (N (1 μ M)), PARP inhibitors (Olaparib=O (1 μ M), Rucaparib=R (0.5 μ M), Talazoparib=T (0.5 μ M)) alone or in combination at the indicated concentrations for 14 days in 3D Matrigel culture (n=3).

SI 4A. 3D Matrigel

SI 4A: Caspase3/7 activity was evaluated using the ApoTox-Glo Triplex Assay Kit in CAOV2 cells treated with NPB (N (1 μ M)), PARP inhibitors (Olaparib=O (1 μ M), Rucaparib=R (0.5 μ M), Talazoparib=T (0.5 μ M)) alone or in combination at the indicated concentrations for 14 days in 3D Matrigel culture (n=3).

- Figure 3C, it would be nice to present the results by separately grouping early and late apoptotic groups. In current format, it is not apparent that there was a difference between N vs N+O for early apoptosis cell numbers although text claims that it is.

Response: As per the reviewer's suggestion, we have now modified Figure 3C format in the revised manuscript as described below:

Figure 3C. Apoptosis

Figure 3C: Flow cytometry analysis of Annexin-V and propidium iodide (PI) staining of apoptotic cell death of CAOV2 cells measured after treatment with NPB (N), Olaparib (O), Rucaparib (R) and Talazoparib (T) or combinations using flow cytometry analysis at 72 hours as described in materials and methods. Apoptotic cells refer to Annexin-V positive (early apoptosis) + Annexin-V and PI double positive (late apoptosis) cells (n=3).

5. Figure 3D, it should be quantified and presented with statistics.

Response: As per the reviewer's request, we have now quantified data represented in Figure 3D and incorporated results in supplementary information (SI) 4D in the revised manuscript as below.

SI 4D. Foci formation

SI 4D: CAOV2 cells in foci formation assays were treated with NPB (N), Olaparib (O), Rucaparib (R) and Talazoparib (T) or combinations and at assay termination were stained with 0.2% crystal violet for 20 mins. To quantify, crystal violet stained colonies were resuspended in Isopropanol. Absorbance of soluble crystal violet (n=3) was detected at 570 nm using microplate reader (Tecan Spark®, Switzerland). Percentage change in absorbance plotted as survival fraction (%) of CAOV2 cells after treatment.

6. Figure 3E, it is interesting that NPB treatment attenuated PARPi-induced CHEK1 and

CHEK2 phosphorylation. This inhibition likely affects DNA damage response signaling. What is the potential mechanism? Could this be the mechanism contributing the drug synergies between NPB and PARPis?

Response: We thank the reviewer for the constructive suggestion.

Both the ATR-CHEK1 and ATM-CHEK2 signaling pathway can respond to DNA double strand breaks (DSBs) through activation of homologous recombination repair (HRR) [3]. Herein we observed that the treatment of OC cells with NPB alone promoted intrinsic mitochondria-based apoptosis and did not produce a significant increase in the levels of phosphorylated CHEK1/2 protein (Figure 3E). However, PARP inhibition in OC cells triggered a DNA damage response and increased levels of phosphorylated CHEK1/2 protein (Figure 3E) to facilitate DNA repair through activation of HR (Figure 3F) and NPB treatment attenuated PARPi-induced CHEK1 and CHEK2 phosphorylation. In addition, it was observed that Olaparib induced RAD51 foci formation in the nuclei of CAOV2 cells was significantly impaired in combination with NPB. Therefore, to determine the role of NPB in the homologous recombination repair process, a HR reporter assay was established (SI 5F). The HR reporter assay demonstrated that NPB did indeed impair the HR process (SI 5F). Therefore, it may be concluded that one mechanism of drug synergy between NPB and PARPis is that NPB also promotes DNA damage, and impairs DNA repair after PARPi treatment by attenuation of HR.

As per the reviewer's suggestion, we have now incorporated new information in the results section of the revised manuscript as described below:

Line 158-159 (subtitle):

“NPB synergizes with PARP inhibition to stimulate apoptotic cell death attenuating PARPi induced HR repair.”

Line 205-211:

“Given that NPB treatment diminished PARPi-induced CHEK1/CHEK2 phosphorylation and RAD51 foci, which both have an important role in HR [3], it was therefore determined whether NPB modulated HR. A HR reporter assay demonstrated that NPB treatment suppressed HR efficiency both in BRCA wild type (CAOV2) and BRCA mutant (AFC) EOC cells (SI 5F). Hence, NPB attenuated the repair of DSBs induced by PARPis by suppression of HR. Thus, NPB synergizes with PARPis in EOC cells to promote DNA damage and impair DNA repair after PARPi treatment.”

SI 5F. Homologous Recombination (HR) reporter assay

SI 5F: The *HR-GFP reporter* consists of two copies of mutant *GFP* genes. Cells were transfected with I-SceI expressing vector to induce DSB and which may be repaired by HR using iGFP as a template and restore a functional *GFP* gene. The percentage of GFP-positive (GFP+) cells was measured using immunofluorescence and flow cytometry as an indicator of HR efficiency. Cells with DSBs introduced by I-SceI endonuclease HR exhibit increased levels of pBAD599 in EOC cells as determined by western blot analysis. A reduction is observed in GFP positive cells after NPB treatment in BRCA wild type and mutant EOC cells (n = 3), Scale bar, 50 μ m.

Supplementary text (D): **HR reporter assay and sensitivity assays:** To establish CAOV2-*DR-GFP* and AFC-*DR-GFP* cells, the *pDR-GFP* plasmid was stably integrated into CAOV2 and AFC cell lines as previously described [4]. After 48 hours of pretreatment with NPB, CAOV2-*DR-GFP* and AFC-*DR-GFP* (1×10^6) cells were transfected by electroporation using FuGENE HD (Promega, US) with 2 μ g pCMV-I-SceI, respectively. The cells were assessed for green fluorescence emission using flow cytometry. Results were corrected according to the transfection efficiency of the cell line.

7. Figure 3F and SI 5B, it is interesting to see that NPB treatment alone increases gamma-H2AX IF. Is NPB treatment inducing genotoxicity?

Response: To address the reviewer's concern, we first performed *in silico* prediction analysis to evaluate potential NPB genotoxicity as per the ICH M7 guidelines and according to the methods described in the previously published literature [5]. Both the genotoxicity predictions with the statistical model GT1_BMUT and the expert rules model GT_EXPERT in CASE Ultra (version: v1.8.0.2) QSAR platforms [6] predicted a NEGATIVE outcome for potential genotoxicity associated

with NPB. The chemical structure of NPB was identified as a potential ICH M7 Class 5 compound. A copy of the *in-silico* analysis is provided in SI12 of the revised manuscript and also described below;

Secondly, based on the previously published reports, sub-lethal apoptotic stresses trigger caspase-dependent DNA damage as demonstrated by a caspase-dependent increase in γ -H2AX both *in vitro* and *in vivo* [7, 8]. Such DNA damage is observed with the related BCL-2 inhibitors (ABT-737 and ABT-199) and with other targeted agents (Mcl-1-selective inhibitor: A-1210477). Hence, an increase in γ -H2AX would also be expected with NPB. Fragmentation of DNA represents a classical apoptotic hallmark mediated by caspase-activated DNase (CAD). To further elucidate the extent of DNA damage, we performed DNA laddering assays (SI 5D). NPB treatment of OC cells increased DNA laddering compared to control cells, accompanied by elevated levels of cleaved-CASP3, PARP cleavage, and γ -H2AX protein, which is consistent with the previous studies in which elevated levels of γ -H2AX protein serve as an indicator of apoptosis [9-11]. We also demonstrated that transient-transfection of a Ser99 phosphorylation deficient human BAD (*hBADS99A*) construct into CAOV2 cells produced increased levels of γ -H2AX protein as evaluated by immunofluorescence (SI 5E).

We have added a more detailed interpretation and incorporated corresponding new data as a supplementary information in SI12 and SI 5D&E of the revised manuscript as described below:

Line 198-205:

“It is recognized that sub-lethal apoptotic stresses trigger a caspase-dependent increase in γ -H2AX expression (53-57). As DNA fragmentation is a classic apoptotic hallmark mediated by caspase-activated DNase (CAD), DNA laddering assays were performed. After NPB treatment, apoptotic DNA laddering increased accompanied by increased caspase-3 and PARP cleavage and γ -H2AX expression consistent with previous studies reporting that increased γ -H2AX levels may also indicate apoptosis (55-57) (SI 5D). Transient-transfection of a Ser99 phosphorylation deficient human BAD (*hBADS99A*) construct into CAOV2 cells also produced increased levels of γ -H2AX protein as evaluated by immunofluorescence (SI 5E).”

Line 388-394:

“Preliminary data indicated that inhibition of BADS99 phosphorylation by NPB is well tolerated (44). NPB is not predicted to be genotoxic by stringent *in-silico* analyses (83) (SI 12). Increased expression of γ -H2AX following NPB treatment herein was therefore due to caspase-dependent DNA damage as reported for other agents targeting BCL-2 family members such as Venetoclax (ABT-199) (53, 54). Furthermore, the combination of BADS99 and PARP inhibition appeared to be well tolerated herein with no significant differences in animal weight nor behavior observed (84, 85).”

Supplementary information 12.

In Silico Genotoxicity Prediction for NPB

1. Outcome of GT1_BMUT Model: Negative

Positive Alerts	Deactivating Features	Unknown Fragments
2	4	0
Alert ID 47: cH:c(c)-N3 Alert ID 97: c-N3(-C3H2)-C3H2	Fragment ID 320: c:(c)-Cl Fragment ID 498: Cl-c1:cH:cH:cH:c1-Cl Fragment ID 1144: c:cH:c(cH:cH)-C2(-N3H=O) Fragment ID 1906: cH:cH:c(-OH):c(-C3H):cH	

There are 2 positive alerts and 4 deactivating features with the model. The QSAR calculated probability is 4.7%, which is lower than the model's current classification threshold (50.0%) and within the gray zone (40.0% ~ 60.0%).

2. Outcome of Expert Rule Model: Negative

Positive Alerts	Deactivating Features	Unknown Fragments
0	0	0

There is no positive alert with the model. The QSAR calculated probability is 0.0%, which is lower than the model's current classification threshold (50.0%) and not within the gray zone (40.0% ~ 60.0%). The outcome of GT1_BMUT model is negative, and the outcome of Expert Rule model is negative. To sum up, the prediction result of the compound is negative and belongs to a potential ICH M7 Class 5 compound.

Expert Review

The outcome of GT1_BMUT model is negative, and the outcome of Expert Rule model is negative. To sum up, the prediction result of the compound is negative and belongs to a potential ICH M7 Class 5 compound.

SI 12: *In-Silico* Genotoxicity Prediction for NPB

The genotoxicity predictions were evaluated by the statistical model GT1_BMUT and the expert rules model GT_EXPERT in CASE Ultra (version: v1.8.0.2) software, respectively. Meanwhile, the konsolidator database BMUT_KONSOLIDATOR_DB (ver 4.0) with experimental data have been used to facilitate expert analysis (ICAS, China).

SI 5D. Caspase-dependent DNA damage

SI 5D: Western blot analysis was used to assess the level of various protein markers and protein activities in CAOV2 cells after treatment with NPB (N) at the indicated concentrations. Soluble whole cell extracts were run on SDS-PAGE and immunoblotted as described in materials and methods. β-ACTIN (ACTB) was used as input control for cell lysate. Genomic DNA and apoptotic DNA fragments were isolated from NPB-treated CAOV2 cells as described in supplementary text and were run on a 2% TAE agarose gel.

Supplementary text (E): **Apoptotic DNA Laddering:** Cells were collected by centrifugation, and the cell pellets were lysed by gentle, repeated pipetting in a 2-mL microcentrifuge tube with 0.5 mL cell lysing buffer (50 mM Tris-HCl, pH 7.5, 0.5% Triton X-100, 20 mM EDTA). After centrifugation at 12000 rpm for 10 min to remove cell debris, as well as intact nuclei of non-apoptotic cells, the supernatant was subject to one round of phenol:chloroform:isoamyl alcohol (25:24:1; pH 7.4; 0.5mL) extraction. Apoptotic DNA fragments and contaminating RNA present in the liquid phase were precipitated by adding 50μL 3M sodium acetate and 50μL phenol:chloroform:isoamyl alcohol (25:24:1; pH 7.4; 0.5 mL). After incubation on ice for 5 min, the precipitated nucleic acids were pelleted by centrifugation at 12000rpm for 10 min. The pellet was resuspended in RNaseA buffer (Promega, Madison, WI, USA) by incubation at 37°C for 30 min or longer. After centrifugation at 12000rpm for 10 min, the pellet was resuspended in phenol:chloroform:isoamyl alcohol (25:24:1; pH 7.4; 0.5mL) and this step was repeated. Apoptotic DNA fragments were resolved by 2 % TAE agarose gel electrophoresis (17, 18). Genomic DNA was extracted followed Wizard® Genomic DNA Purification Kit instructions (Promega, Madison, WI, USA).

SI 5E. IF analysis

SI 5E: Forced expression of *Flag-hBADS99A* in CAOV2 cells exhibited no pBADS99 immunoreactive signal and induced γ H2AX expression as confirmed by immunofluorescent analysis, Scale bar, 20 μ m.

8. What is a potential explanation for good correlation between pBAD-S99 phosphorylation and IC50 for olaparib but not with rucaparib or talazoparib?

Response: Due to the varying pharmacological features of different chemical inhibitors impacting on similar pathways or processes and the necessary differences in their mechanisms there exists a fundamental rationale for a diverse response in efficacy [12]. The PARP family comprises 17 members [13]. All PARP family proteins vary greatly in structure and in their cellular functions. Among PARP family proteins, PARP1, PARP2, PARP4, and PARP-5a/5b have been reported to exhibit PARP activity [14], which are reported for the DNA damage repair (DDR) [15]. In addition to catalytic inhibition, PARP inhibitors exert their cytotoxicity by trapping PARP on SSB sites [16]. Specifically, Olaparib is reported to trap PARP1 and 2 [17], while Rucaparib traps PARP1, PARP2 and PARP3, with higher affinity to PARP1 [18, 19], and Talazoparib specifically traps only PARP1 [20]. Therefore, PARP inhibitors should be evaluated based both on catalytic PARP inhibition and PARP–DNA trapping [21]. Western blotting analyses of drug-treated cells showed that all three PARP inhibitors, Olaparib, Rucaparib or Talazoparib, reduced total cellular PARP levels in a concentration-dependent manner indicating that the 3 inhibitors are comparable at inhibiting PARP protein catalytic activity [21]. Talazoparib is ~100-fold more potent at trapping PARP–DNA complexes and more cytotoxic as a single agent than Olaparib, whereas both Olaparib and Rucaparib exhibit similar potencies in trapping PARP–DNA complexes [21]. In addition, PARP inhibitors differ in their off-target effects: For instance, Olaparib, mitigates DNA damage repair activity via a G2 cell-cycle arrest-like effect in a p53-dependent manner [22]. These off-target effects may also add to the efficacy attributed to PARPi.

When studying the efficacy of different PARP inhibitors, we utilized a consistent

concentration and consistent temporal end point of the treatments to determine the drug response. As shown in SI 1G, Rucaparib and Talazoparib exhibited greater efficacy in EOC cell lines compared to Olaparib after 6-days of treatment. We also tested different time-points for IC₅₀ of the three PARPis (24h/48h/72h) (not previously shown in the manuscript), which exhibited a good correlation between pBAD-S99 phosphorylation and IC₅₀ for Rucaparib and Talazoparib at 48h and 72h respectively as described below.

To answer the reviewer, we have now incorporated new data and related information in the revised manuscript as described below:

line 115-120:

“The half maximal inhibitory concentration (IC₅₀) of Olaparib (after 6 days) in EOC cells positively correlated with the levels of pBADS99 protein (Figure 1F). However, it should be noted that the IC₅₀ values of Rucaparib or Talazoparib after 6 days of treatment did not exhibit a positive correlation with the levels of pBADS99 in EOC cells (SI 1F). However, the IC₅₀ of Rucaparib at 48h and the IC₅₀ of Talazoparib at 72h positively correlated with the levels of pBADS99 in EOC cells, respectively.”

SI 1F. Correlation of pBADS99 to IC₅₀ of Rucaparib or Talazoparib

SI 1F: Correlation between pBADS99 levels and Rucaparib or Talazoparib half maximal inhibitory concentration (IC₅₀) values of EOC cell lines. Left: The IC₅₀ of Rucaparib or Talazoparib in different time endpoints was calculated by total cell number counting (n=3). Right: levels of BADS99 protein showed no correlation with IC₅₀ of Rucaparib or Talazoparib sensitivity in EOC cells: the IC₅₀ of Rucaparib for 48h and the IC₅₀ of Talazoparib for 72h exhibited a significant correlation with the levels of pBADS99 protein in EOC cells. Statistical changes were measured using *Pearson correlation* analysis between levels of pBADS99 protein and IC₅₀ values of Rucaparib or Talazoparib in EOC cell lines.

9. Formatting errors:

Ref 55; Figure 2 Y-axis; Figure 4B Y-axis; SI 4D: ELISA reader should be replaced

with plate reader

Response: We thank reviewer for highlighting the errors. We have now rectified the mentioned errors in the revised manuscript as described below.

Ref 55 (Ref 63 in revised manuscript):

56. Sun C, Fang Y, Labrie M, Li X, & Mills GB (2020) Systems approach to rational

combination therapy: PARP inhibitors. *Biochem Soc Trans* 48(3):1101-1108.

Figure 2 Y-axis:

Figure 2. Combination efficacy of NPB and PARP inhibitors in EOC cell lines

Figure 2: NPB synergizes with PARP inhibitors in EOC cells to decrease cell survival: Chou-Talalay analysis was used for NPB (N) in combination with and three PARP inhibitors, namely Olaparib (O), Rucaparib (R) and Talazoparib (T), in a panel of EOC cells including CAOV2 (BRCA-proficient/Platinum sensitive), A2780 (BRCA-proficient/Platinum sensitive), A2780cisR

(BRCA- proficient /Platinum resistant) and AFC (BRCA-deficient/Platinum-resistant). Cells were treated with the indicated concentration (log10 scale) of NPB and mentioned PARP inhibitors for 6 days. The percentage of survival fraction was assessed using total cell number assay. The logarithmic combination index (CI) value corresponding to cell fraction affected (Fa) was determined using the CompuSyn software (<http://www.combosyn.com>) as described in materials and methods. CI value indicates: < 1 synergism; = 1 additive Synergy; > 1 antagonism (n=3). Dose-response curves for a panel of cells treated with the indicated concentration of PARP inhibitors with or without 1µM NPB in total cell number assays. Arrow indicates fold reduction in respective PARP inhibitor IC₅₀ in the presence of NPB (n=3). Points are mean of triplicate experiments; bars, ±SD.

Figure 4B Y-axis:

Figure 4B. Foci formation (PDO1)

Figure 4B: Survival fraction (SF) of ascitic fluid cancer (AFC) cells (from PDO1) treated with NPB (N), Olaparib (O), Rucaparib (R) and Talazoparib (T) or combinations for foci formation. Foci were stained with 0.2% crystal violet and quantified using a microplate reader (Tecan Spark®, Switzerland) by detecting absorbance of soluble crystal violet. Below: Whole-well images of foci formation.

SI 4D: CAOV2 cells treated with NPB (N), Olaparib (O), Rucaparib (R) and Talazoparib (T) or combinations in foci formation were stained with 0.2% crystal violet for 20 mins. To quantify, crystal violet stain colonies were resuspended in Isopropanol and absorbance of soluble crystal violet (n=3) was determined at 570 nm using a microplate reader (Tecan Spark®, Switzerland). Percentage change in absorbance plotted as survival fraction (%) of CAOV2 cells after treatment.

Reviewer #2 (Remarks to the Author):

This is a very good study and I have a few suggestions for improvements. Points 1, 3, 4, 6 and 10 are the most critical points that need to be addressed:

Response: We thank the reviewer for the constructive, and positive overview of the manuscript.

1. P-BAD-S99 levels are associated with poor survival and also with BRCA-deficiency.

Can the authors do a multivariate analysis to ascertain if the prognostic value of P-BAD-S99 is dependent on BRCA-deficiency and/or all other significant markers of outcome? This will give insight into the genetic interaction between P-BAD-S99 and BRCA and other markers.

Response: Whilst we carefully screened and analyzed the patient data in the cohort used, the complete molecular profiling of patient specimens was not available in the hospital due to the burden of cost for such profiling being placed on the patient. Given the time lapse between new specimen collection and accumulation of 5-year survival data, we therefore adopted *in vitro* based approaches to analyze the genetic interaction between pBADS99 and BRCA or other markers summarized in Figure 1C, Figure 1D and SI 1E. Furthermore, along with Olaparib [23, 24], Rucaparib [25] and Talazoparib [26] have also been approved for non-BRCA mutant ovarian cancer. Therefore, herein, we examined the combined inhibition of BAD phosphorylation and PARP in EOC regardless of BRCA mutational status. We have now also incorporated new textual details explaining the current limitation of patient specimens in the revised manuscript as described below:

Line 353-356:

“Along with Olaparib (73, 74), Rucaparib (75) and Talazoparib (76) were also approved by the FDA for the treatment of ovarian cancer regardless of BRCA-mutational status. However, due to the limited clinical genomic annotation, potential correlations between the prognostic value of pBADS99 and BRCA-status were not determined.”

2. Along the same lines as point #1, what do the authors speculate is the relationship between BRCA-deficiency and phosphorylation of BAD? Do the authors think that P-

BAD is upstream, downstream or independent of BRCA? For example, as a downstream model, does BRCA-deficiency lead to activation of BAD kinases? Some discussion on this may be interesting.

Response: It is reported that BRCA1 acts as a negative regulator of the PI3K/AKT pathway and hence the significance of the BRCA pathway (including BRCA1/2) in cancer development and progression as *BRCA* deficiency/mutation activates the AKT pathway upstream of BAD phosphorylation [27-29].

Furthermore, BCL-2 has been reported to regulate DSB repair via its interaction with BRCA1 [30] and relocate to sites on the chromatin, where it can directly interact with and inhibit PARP1. This interaction can be disrupted by BH3-only BCL2 family members (eg. BAD) [31]. Responding to DNA damage, TP53 directly binds to the *BAD* promoter region and upregulates *BAD* transcription [32]. *BRCA1* deficient cancers harbor mutations in *TP53* suggesting that inactivation of p53 is a requirement for cancer progression in *BRCA1* deficiency.

As per the reviewer's suggestion, we have now incorporated new information in the discussion section of the revised manuscript as below:

Line 337-344:

“Increased RAS/MEK/ERK and PI3K/AKT/mTOR activities increase BAD phosphorylation providing an actionable vulnerability, regardless of mutational status. *BRCA* deficiency/mutation activates the AKT pathway with consequent BAD phosphorylation (64, 65). Furthermore, BCL2 has been reported to regulate DSB repair via its interaction with BRCA1 (70) and this interaction can be disrupted by BH3-only BCL2 family members (71). Therefore, BRCA-deficiency promotes PI3K/AKT pathway dependent phosphorylation of BAD protein. The observation in this study that BRCA-deficient cell lines exhibited significantly higher levels of pBADS99 compared to BRCA-proficient cell lines supports this notion.”

3. How specific is the NPB-effect on BAD? The authors show that NPB decreases P-BADS99 levels, which is associated with increased apoptosis. Given that BAD phosphorylation also modulates non-apoptotic pathways, the authors should demonstrate the specificity of NPB for BAD in the presented assays here. For example, can the authors test whether loss-of-function (BAD knock-out) prevents NPB-mediated apoptosis? Does loss-of-function BAD knock-out prevent the synergy between NPB and PARPi with respect to growth effects/markers of proliferation

apoptosis, etc.?

Response: In accordance with the reviewer's suggestion, we have now assessed the specificity of NPB in the models (cell lines and PDO) utilized in this manuscript and also the NPB-PARPi synergy after *siRNA*-mediated depletion of BAD expression in CAOV2 cells and PDO1. We have now incorporated new data as supplementary information (SI 3C&D, SI 6G) and description in the result section of the revised manuscript as described below:

Line 145-157:

“To confirm the functional specificity of NPB directed to BAD, CAOV2 cells were treated with NPB after *siRNA* mediated depletion of BAD expression (SI 3C). Transient-transfection of CAOV2 cells with *siRNA* directed to the BAD transcript decreased BAD expression and decreased levels of pBADS99 compared to control cells (transfected with scrambled oligo) as observed by WB analysis (SI 3C). No significant changes in caspase3/7 activity nor cell viability were observed upon *siRNA*- mediated depletion of BAD in CAOV2 cells (SI 3C) and as previously reported in various carcinoma cells (44). NPB treatment of control transfected cells decreased phosphorylation of BADA99 compared to vehicle-treated CAOV2 cells. Consistently, exposure of CAOV2 control cells to NPB increased caspase3/7 activity and decreased cell viability compared to vehicle-treated cells. In contrast, NPB did not affect caspase 3/7 activity nor cell viability in PDO1 with depleted expression of BAD. Furthermore, no synergistic effect was observed between NPB and PARPi in CAOV2 cells with *siRNA* mediated depletion of *BAD* compared to control transfected CAOV2 evaluated by a total cell number assay (SI 3D).”

Line 244-254:

“To confirm the functional specificity of NPB to BAD, the effect of NPB exposure after *siRNA*-mediated depletion of BAD expression was examined in PDO1 (SI 6G). Transient-transfection of PDO1 with *siRNA* directed to the BAD transcript decreased BAD expression and decreased levels of pBADS99 compared to control PDO1 (transfected with scrambled oligo) as observed by WB analysis (SI 6G). No significant changes in cell viability nor caspase 3/7 activity were observed upon *siRNA*-mediated depletion of BAD in PDO1, and as previously reported for various carcinoma cells (44). As observed in SI 6G, NPB treatment of the control transfected PDO1 decreased phosphorylation of BADA99 compared to control transfected PDO1 treated with vehicle. Consistently, exposure of control transfected PDO1 to NPB decreased cell viability and increased caspase 3/7 activity compared to vehicle-exposed PDO1. In contrast, NPB did not affect caspase 3/7 activity nor cell viability in PDO1 with depleted expression of BAD (SI 6G).”

Supplementary information 3.

C. CAOV2/ CAOV2-siRNA BAD

D. Combination efficacy of NPB and PARP inhibitors in CAOV2/CAOV2-siRNA BAD

SI 3C: *siRNA*-mediated depletion of BAD expression abrogated the effect of NPB in CAOV2 cells. WB analysis was used to assess the levels of pBADS99 and BAD protein in CAOV2 cells after treatment with 10μM NPB. Depletion of BAD expression was achieved using transient transfection of *siRNA* directed to the BAD transcript as previously described [1]. Soluble whole-cell extracts were run on an SDS/PAGE gel and were immunoblotted as described in Materials and Methods. β-ACTIN was used as input control. Caspase 3/7 activity and cell viability were evaluated using the ApoTox-Glo Triplex Assay Kit.

SI 3D: Dose-response curves for CAOV2 and CAOV2 cells with *siRNA* mediated depletion of BAD treated with the indicated concentration of PARPis with or without 1uM NPB in a total cell number assay. Arrow indicates fold reduction in respective PARP inhibitor IC₅₀ in the presence of NPB (n=3).

SI 6G. PDO1/PDO1-siRNA BAD

SI 6G: *siRNA*-mediated depletion of BAD expression abrogates the effect of NPB in PDO1: WB analysis was used to assess the levels of pBADs99 and BAD protein in PDO1 after treatment with NPB. Depletion of BAD expression was achieved using transient transfection of *siRNA* directed to the BAD transcript as previously described [2] (Figure 6A&B). Organoids were harvested after depolymerization of Matrigel using Cultrex® organoid harvesting solution (R&D systems, US) and organoid-pellets were resuspended in RIPA buffer plus protease inhibitors. Extracts were run on an SDS-PAGE and immunoblotted as described in materials and methods. β-ACTIN was used as input control. Cell viability and Caspase 3/7 activities were evaluated using the ApoTox-Glo Triplex Assay Kit.

Supplementary text (F): **siRNAs and transfection:** The target-directed *siRNAs* (*shRNA-BAD1*, 5'-GCUCCGCACCAUGAGUGACGAGUUU-3' and *shRNA-BAD2*, 5'-AAACUCGUCACUCAUCCUCCGGAGC-3') and the negative control *siRNA*, which was used for normalization, were synthesized by GENEWIZ (Suzhou, China). After seeding and adherence for 24h, cells were washed with PBS and transfected with the *siRNAs* for 24h using FuGENE HD (Promega, US) according to the manufacturer's instructions. The *siRNAs* were transfected either separately with constant-ratio diluted PARP inhibitors (single treatments) or with combinations of constant-ratio diluted PARP inhibitors and NPB(1μM).

- Along the same lines as point #3, what is the mechanism by which NPB inhibits phosphorylation of BAD? Is this via blocking accessibility of BAD-kinases to S99, or via inhibiting BAD-kinases directly, or stimulating BAD-phosphatase activity? Some explanation of this is needed.

Response: The proposed mechanism of NPB inhibition of phosphorylation of BAD has been previously described [2]. Briefly, NPB does not affect phosphorylation levels of other signaling molecules nor the predominantly p44/42 MAP kinase phosphorylated Ser75 residue on human BAD [1]. Although NPB is also predicted to interact with BCL-2, NPB interacts directly with BAD in the absence of BCL-2 as predicted by Laplacian-modified naive Bayesian classifier algorithm analysis and as previously demonstrated using surface plasmon resonance (SPR) [2]. Based on *in silico* docking analyses, the dichlorophenyl moiety of NPB was predicted to occupy an additional hydrophobic side pocket within the human BCL-2/BAD interface formed by the side-chains of Leu97, Trp144, and Phe198 residues, in comparison to BCL-2 inhibitors which do not. Additionally, no effect of NPB on AKT kinase activity was observed after cellular treatment with NPB nor in *in vitro* kinase assays [2]. Thus, NPB has been proposed to inhibit phosphorylation of BADS99 by steric hindrance independent of the upstream AKT-kinase or other phosphorylation activities [2].

5. Again, along the same lines as point #3, does NPB increase the BAD-specific

apoptotic pathway? One would expect increased BAD: Bcl-2-like interactions.

Similarly, the authors demonstrate that mutant BAD-S99A decreases cell survival.

The authors should show that this tagged construct is appropriately located within the cell.

Response: We acknowledge the rationality of the reviewer's comment regarding the NPB treatment dependent increase in the BAD:BCL2 interactions, which would allow activation of BAK/BAX with enhanced release of cytochrome C to the cytoplasm with the subsequent promotion of the intrinsic apoptotic pathway. Herein, we therefore used WB and apoptotic DNA laddering assays to demonstrate the NPB-dependent increase of BAD-specific apoptosis (SI 5D), in addition to the further experimentation in point 3 from this reviewer to demonstrate specificity of NPB to BAD.

SI 5D. Caspase-dependent DNA damage

SI 5D: Western blot analysis was used to assess the level of various protein markers and protein activities in CAOV2 cells after treatment with NPB (N) at the indicated concentrations. Soluble whole cell extracts were run on SDS-PAGE and immunoblotted as described in materials and methods. β-ACTIN (ACTB) was used as input control for cell lysate. Genomic DNA and apoptotic DNA fragments were isolated from NPB-treated CAOV2 cells as described in supplementary text and were run on a 2% TAE agarose gel.

Supplementary text (E): Apoptotic DNA Laddering: Cells were collected by centrifugation, and the cell pellets were lysed by gentle, repeated pipetting in a 2-mL microcentrifuge tube with 0.5 mL cell lysing buffer (50 mM Tris-HCl, pH 7.5, 0.5% Triton X-100, 20 mM EDTA). After centrifugation at 12000 rpm for 10 min to remove cell debris, as well as intact nuclei of non-apoptotic cells, the supernatant was subject to one round of phenol:chloroform:isoamyl alcohol (25:24:1; pH 7.4; 0.5mL) extraction. Apoptotic DNA fragments and contaminating RNA present in the liquid phase were precipitated by adding 50μL 3M sodium acetate and 50μL phenol:chloroform:isoamyl alcohol (25:24:1; pH 7.4; 0.5 mL). After incubation on ice for 5 min, the precipitated nucleic acids were pelleted by centrifugation at 12000rpm for 10 min. The pellet was resuspended in RNaseA buffer (Promega, Madison, WI, USA) by incubation at 37°C for 30 min or longer. After centrifugation at 12000rpm for 10 min, the pellet was resuspended in phenol:chloroform:isoamyl alcohol (25:24:1; pH 7.4; 0.5mL) and this step was repeated. Apoptotic DNA fragments were resolved by 2 % TAE agarose gel electrophoresis (17, 18). Genomic DNA was extracted following Wizard® Genomic DNA Purification Kit instructions (Promega, Madison, WI, USA).

To demonstrate that the tagged BADS99A was appropriately located within the cell, we transiently transfected *BADS99A-flag* construct and localized the protein produced with flag antibody by immunofluorescence (IF). We also utilized *flag tagged wild type BAD* as control (SI 1H). We have now incorporated new data as supplementary information (SI 1H) and description in the result section of the revised manuscript as described below:

Line 127-131:

“To confirm the *Flag-hBADS99A* construct was appropriately located within the cell, we performed immunofluorescence (IF) with BAD/pBADS99 and flag antibodies in flag-tagged wild type *hBAD* and *hBADS99A* transfected CAOV2 cells (SI 1H). Note that similar to wild-type BAD protein, *flag-tagged BADS99A* localized within the cytoplasm of the cells (SI 1H).”

SI 1H. Localization of Tagged *hBAD-Flag* and *hBADS99A-Flag* construct in CAOV2

SI 1H: Immunofluorescence staining was performed with a rabbit anti-BAD or pBADS99 and mouse monoclonal anti-Flag antibodies in Flag tagged wild type *hBAD* and *hBADS99A* transfected CAOV2 cells. Scale bar, 20 μm .

6. Controls are needed for some of the immunohistochemistry data. Importantly, the specificities of the anti-P-BAD-S99 and anti-BAD antibodies are not clear. This is critically important. Under these IHC conditions, the authors should demonstrate a loss of signal in BAD knock-out cells/tissue, or loss of signal in the presence of competing BAD peptides. Also, in Fig. SI 1, are the NCO samples under lower magnification than the EOC?

Response: We acknowledge the reviewer's concern regarding antibody specificity. In accordance with the reviewer's suggestion, we depleted BAD protein in CAOV2 cells using *siRNA* and examined using IHC. Depletion of BAD resulted in loss of immunoreactive signals for both pBADS99 and BAD. Representative images of BAD and pBADSer99 immunoreactivities in CAOV2 cells transfected with vector control or BAD-siRNA are shown below:

SI 8C. Immunocytochemistry

SI 8C: Representative images of BAD and pBADS99 immunoreactivities in siRNA-mediated depletion of BAD in CAOV2 cells, Scale bar, 20 μm .

To further exemplify the specificity of the BAD and pBADS99 antibody, we also transfected CAOV2 cells with the flag-tagged wild type *hBAD* and *hBADS99A* as part of Figure (SI 1H). There was no p-BADS99 immunoreactive signal observed in cells transfected with the phosphorylation deficient BADS99A mutant whereas control and hBAD transfected cells exhibited pBADSer99

immunoreactivity.

SI 1H. Localization of Tagged *hBAD-Flag* and *hBADS99A-Flag* construct in CAOV2

SI 1H: Immunofluorescence staining was performed with a rabbit anti-BAD or pBADS99 and a mouse monoclonal anti-Flag antibodies in Flag tagged wild type hBAD and hBADS99A transfected CAOV2 cells. Scale bar, 20 μ m.

As per the reviewer's suggestion, we have now incorporated new data (above) in the supplementary information (SI 8C and SI 1H) of the revised manuscript.

7. It is not clear how the authors concluded that p-BAD is associated with increased

FIGO scores (Fig. 1)

Response: We thank the reviewer for highlighting this oversight. pBADS99 IRS was correlated with FIGO stages (S1-4) using *Spearman's rank correlation coefficient*. We have now incorporated methodology and statistical information in the "Material and Methods" in the revised manuscript as described below:

Line 419-426:

"All cancer patients were staged according to the International Federation of Gynecology and Obstetrics (FIGO) classification.

Immunohistochemistry (IHC) analysis was performed as previously described [2] using primary antibodies tabulated in Supplementary text. The details of the cohort and histoscore methodology have been previously described [33] and briefly in Supplementary text. The correlation of pBADS99 IRS with clinicopathological features (Age, FIGO stage, Lymph node metastasis, Ki67 and TP53) of the EOC patient cohort was determined using Spearman's rank correlation coefficient.

Figure 1A: Spearman's correlation analysis between pBADS99 and clinicopathological features of the EOC patient cohort."

8. Can the authors speculate on the underlying mechanisms causing differential effects with the different PARPi's? For example (a) the IC50 for Olaparib was associated with PBAD-S99 status, but the other PARPi's were not; and (b) N+O increases apoptosis and does not affect cell cycle, whereas N+T and N+R have clear G2/M arrest. Some discussion around this would be interesting.

Response: we thank reviewer for highlighting this important point.

Due to the varying pharmacological features of inhibitors with different chemical structures impacting on similar pathways or processes and the necessary differences in their mechanisms there exists a fundamental rationale for a diverse response in efficacy [12]. The PARP family comprises 17 members [13]. All PARP family proteins vary greatly in structure and in their cellular functions. Among PARP family proteins, PARP1, PARP2, PARP4, and PARP-5a/5b have been reported to exhibit PARP activity [14]. Specifically, Olaparib is reported to trap PARP1 and 2 [17], while Rucaparib traps PARP1, PARP2 and PARP3, with higher affinity to PARP1 [18, 19], and Talazoparib specifically traps only PARP1 [20], which are reported for the DNA damage repair DDR [15]. In addition to catalytic inhibition, PARP inhibitors exert their cytotoxicity by trapping PARP on SSB sites[16]. Specifically, Olaparib is reported to trap PARP1 and 2 [17], while Rucaparib traps PARP1, PARP2 and PARP3, with higher affinity to PARP1 [18, 19], and Talazoparib specifically

traps only PARP1 [20]. Therefore, PARP inhibitors should be evaluated based both on catalytic PARP inhibition and PARP–DNA trapping [21]. Talazoparib is ~100-fold more potent at trapping PARP–DNA complexes and more cytotoxic as a single agent than Olaparib, whereas both Olaparib and Rucaparib exhibit similar potencies in trapping PARP–DNA complexes [21]. In addition, PARP inhibitors differ in their off-target effects: For instance, Olaparib, mitigates DNA damage repair activity via a G2 cell-cycle arrest-like effect in a p53-dependent manner [22]. These off-target effects may also add to the efficacy attributed to PARPi. When studying the efficacy of different PARP inhibitors, we utilized a consistent concentration and temporal end point of the treatment to determine the drug response. As shown in SI 1G, Rucaparib and Talazoparib exhibited greater efficacy in EOC cell lines compared to Olaparib after 6-days of treatment. We also tested different time-point IC₅₀ of the three PARPis (24h/48h/72h) (not shown in previous manuscript), which exhibited a good correlation between pBAD-S99 phosphorylation and IC₅₀ for Rucaparib and Talazoparib at 48h and 72h respectively as described below. Compared to Rucaparib and Talazoparib, Olaparib is the most widely clinically utilized and the best studied PARP inhibitor, so we performed the assays with fixed-time treatment in which Olaparib demonstrated the effective response.

We have now added a more detailed interpretation and also incorporated corresponding new data as a supplementary information in SI 1G of the revised manuscript as described below:

line 115-120:

“The half maximal inhibitory concentration (IC₅₀) of Olaparib (after 6 days) in EOC cells positively correlated with the levels of pBADS99 protein (Figure 1E). It should be noted that the IC₅₀ values of Rucaparib or Talazoparib after 6 days of treatment did not exhibit a positive correlation with the levels of pBADS99 in EOC cells (SI 1F). However, the IC₅₀ of Rucaparib at 48h and the IC₅₀ of Talazoparib at 72h positively correlated with the levels of pBADS99 in EOC cells, respectively.”

SI 1F. Correlation of pBADS99 to IC₅₀ of Rucaparib or Talazoparib

SI 1F: Correlation between pBADS99 levels and Rucaparib or Talazoparib half maximal inhibitory concentration (IC₅₀) values of EOC cell lines. Left: The IC₅₀ of Rucaparib or Talazoparib in different time endpoints was calculated by total cell number counting (n=3). Right:

levels of BADS99 protein showed no correlation with IC₅₀ of Rucaparib or Talazoparib sensitivity in EOC cells: the IC₅₀ of Rucaparib for 48h and the IC₅₀ of Talazoparib for 72h exhibited a significant correlation with the levels of pBADS99 protein in EOC cells. Statistical changes were measured using *Pearson correlation* analysis between levels of pBADS99 protein and IC₅₀ values of Rucaparib or Talazoparib in EOC cell lines.

Similarly for cell cycle: We have kept the same treatment times for all PARPis: hence, cells were undergoing NPB induced G1 arrest at the same time as Olaparib induced G2 arrest, so the population of cell subphases in N+O seems not changed; compared to N+O, N+T and N+R have clear G2/M arrest as N+T and N+R both already exhibit catastrophic effects on cell death at this time point. As Olaparib is currently the most widely clinically utilized PARPi, and hence we preferred to keep the *in vitro* assays consistent with an effect of Olaparib.

We have now incorporated new information in the discussion section of the revised manuscript as described below:

Line358-371:

“Of note, due to the varying pharmacological features of different inhibitors impacting on similar pathways or processes and the necessary differences in their mechanisms, there exists a fundamental rationale for a diverse response in efficacy. Specifically, Olaparib is reported to trap PARP1 and 2 [17], while Rucaparib traps PARP1, PARP2 and PARP3, with higher affinity to PARP1 [18, 19], and Talazoparib specifically traps only PARP1 [20]. In addition, PARP inhibitors differ in their off-target effects. These off-target effects may also contribute to the efficacy attributed to PARP inhibition. Therefore, when the efficacy of different PARP inhibitors was examined, a consistent concentration and temporal end point of the treatment to the drug response was utilized. As shown in SI 1G, Rucaparib and Talazoparib exhibited greater efficacy in EOC cell lines; however no significant correlation with the levels of pBADS99 in EOC cells was observed compared to Olaparib treatment after 6-days. However, a significant correlation between pBADS99 and IC₅₀ for Rucaparib or Talazoparib was observed at 48h and 72h respectively. Olaparib was the first FDA approved PARPi and therefore the experimental approaches were tailored using Olaparib.”

9. The authors suggest that combining NPB with PARPi's would be less toxic than combining PARPi's with RAS or PI3K inhibitors. It would have been great to compare these different treatments directly for *in vivo* toxicities relative to tumor response. This would have enabled direct “benchmarking” to address the authors' comments on lines 322 regarding already documented toxicities. Understandably, this would be considerable additional experimentation and is not critical.

Response: We acknowledge the rationality of the reviewer's comment regarding the evaluation of *in vivo* toxicities associated with combining NPB-PARPis compared to RAS or PI3K inhibitors. Indeed, such considerations are rational and important for any future clinical development of combination strategies. However, we believe the assessment of comparisons of *in vivo* toxicities associated with NPB combinations is beyond scope of the current study. We would like to assure the reviewer that we are indeed currently examining a RAS-BAD phosphorylation inhibitor combination in KRAS mutated cancer and the results would be better presented therein.

10. All representative western blots should have accompanied quantitation of biological replicates with appropriate statistics. This is critical as the authors make quantitative conclusions from this data. (For example, Fig. 3E and Fig. 4C). In Fig. 4C, the authors conclude that caspase 7 cleavage is elevated in response to N/O vs N or O alone. This is not convincing on the representative western blot.

Response: We acknowledge the reviewer's comment and agree on the need for improved clarity. We have performed the densitometric analysis of all the western blots represented in Figure 1D, 3E, and 4C using J Image online platform (<https://imagej.nih.gov>). The densitometric analysis of Figure 1D is summarized in SI 1F, the densitometric of Figure 3E is summarized in SI 5A, and the densitometric of Figure 4C summarized in SI 6F are now included in the revised manuscript as described below:

Figure 1D. pBADS99/BAD levels in EOC cells lines

Figure 1D: Western blot analysis was used to assess the levels of pBADS75 and pBADS99 and BAD expression in EOC cell lines. Soluble whole cell extracts were run on an SDS-PAGE and immunoblotted as described in materials and methods. β -ACTIN was used as input control for cell lysate. The sizes of detected protein bands in kDa are shown on the left side. Densitometries of protein bands were subsequently determined using ImageJ software (<https://imagej.nih.gov/ij/>). Below: Specific gene mutations in the EOC cell lines are indicated by the green colored square, described in materials and methods.

SI 1F. pBADS99 and EOC molecular status correlation

SI 1F: **Left: Relative intensity of pBADS99/BAD in 8 EOC cell lines.** Right: No statistically significant association of pBADS99 with mutations of KRAS, PTEN, PI3KCA or TP53) were observed in the panel of EOC cells.

Figure 3E. Western Blot

Figure 3E: Western blot analysis was used to assess the level of various proteins and protein activities in CAOV2 cells after treatment with NPB (N), Olaparib (O), Rucaparib (R), Talazoparib (T) or combinations thereof. Soluble whole cell extracts were run on an SDS-PAGE and immunoblotted as described in materials and methods. β-ACTIN (ACTB) was used as input control for cell lysate. The sizes of detected protein bands in kDa are shown on the left hand side.

Supplementary information 5.

A. Western Blot analysis and quantification

SI 5A: Densitometry of western blot analysis represented in the Figure 3E. Densitometric analysis of protein blots was determined using ImageJ software (<https://imagej.nih.gov/ij/>). Statistical significance is indicated by asterisks.

changes were assessed by using an unpaired two-tailed Student's t test ($P < 0.05$ was considered as significant), GraphPad Prism.

In accordance with the reviewer's comment, we have now incorporated a new cleaved-CASPASE7 blot in the figure 4C of the revised manuscript, as described below:

Figure 4C. Western Blot (PDO1)

Figure 4C: Western blot analysis of ascitic fluid cancer (AFC) cell derived organoids (also designated as PDO-1) in 3D Matrigel after treatment with NPB (N) alone or in combination with Olaparib (O), Rucaparib (R) or Talazoparib (T). Organoids were harvested after depolymerization of Matrigel using Cultrex® organoid harvesting solution (R&D systems, US) and organoid-pellet was resuspended in RIPA buffer plus protease inhibitors. Extracts were run on an SDS-PAGE and immunoblotted as described in materials and methods. β-ACTIN (ACTB) was used as input control for cell lysate. The sizes of detected protein bands in kDa are shown on the left side.

Supplementary information 6.

F. Western Blot quantification (PDO1 Patient derived cells)

SI 6F: Western blot analysis was used to assess the level of various protein markers and protein activity in ascitic fluid cancer (AFC) cells (from PDO1) after treatment with NPB (N), Olaparib (O), Rucaparib (R) and Talazoparib (T) or combinations. Soluble whole cell extracts were run on an SDS-PAGE and immunoblotted as described in materials and methods. β -ACTIN (ACTB) was used as input control for cell lysate. Densitometric analysis of protein bands was subsequently performed using ImageJ software (<https://imagej.nih.gov/ij/>). Statistical significance was assessed by using an unpaired two-tailed Student's t test ($P < 0.05$ was considered as significant) using GraphPad Prism.

11. Figure 2, the Y-axis for SF and DRC is "off".

Response: We thank the reviewer for highlighting this error. We have now rectified the error in the revised manuscript as described below:

Figure 2. Combination efficacy of NPB and PARP inhibitors in EOC cell lines

Figure 2: NPB synergizes with PARP inhibitors in EOC cells to decrease cell survival: Chou-Talalay synergy analysis for NPB (N) in combination with and three PARP inhibitors, namely Olaparib (O), Rucaparib (R) and Talazoparib (T), in a panel of EOC cells including CAOV2 (BRCA-proficient/Platinum sensitive), A2780 (BRCA-proficient/Platinum sensitive), A2780cisR (BRCA- proficient /Platinum resistant) and AFC (BRCA-deficient/Platinum-resistant). Cells were treated with the indicated concentration (log10 scale) of NPB and mentioned PARP inhibitors for

6 days. The percentage of survival fraction was assessed using total cell number. The logarithmic combination index (CI) value corresponding to cell fraction affected (Fa) was determined using the CompuSyn software (<http://www.combosyn.com>) as described in materials and methods. CI value indicates: < 1 synergism; = 1 additive Synergy; > 1 antagonism (n=3). Dose-response curves for a panel of cells treated with the indicated concentration of PARP inhibitors with or without 1 μ M NPB in total cell number assays. Arrow indicates fold reduction in respective PARP inhibitor IC₅₀ in the presence of NPB (n=3). Points are mean of triplicate experiments; bars, \pm SD.

12. In Figure 5A xenograft experiment, how was the statistical analysis done? Are the differences based on the endpoint tumor volume?

Response: The statistical differences between the treatment groups were compared using an *unpaired two-tailed Student's t test* based on endpoint tumor volume. To clarify and in accordance with the reviewer concerns, we have added a brief description in the revised manuscript and Figure 5A legend as described below.

Line 268-272:

“In addition, the A2780 and A2780CisR cell generated tumors treated with NPB-Olaparib exhibited significantly decreased tumor volume compared to either Olaparib or NPB treated animals (Figure 5A). Statistical significance was analyzed based on endpoint tumor volume. No significant changes in body weight (Figure 5A) were observed in treatment groups compared with the vehicle group during the treatment period.”

Figure 5.

A. Xenograft

Figure 5A: A2780 or A2780 cisplatin resistant (5×10^6) cells were injected subcutaneously into the flank of 5-week-old BALB/c athymic mice, respectively. When the tumor volume reached 100-150mm³, in each cell line group, the mice were randomized into the 4 indicated treatment groups (n

= 6). Mice were treated daily with vehicle (V), 20 mg/kg NPB (N), 50 mg/kg Olaparib (O) or combined NPB-Olaparib by i.p. injection. Tumor growth was monitored daily by measurement of the tumor volume. The statistical changes between treatment groups were analyzed using an *unpaired two-tailed Student's t test* based on endpoint tumor volume. Below: mean animal weights of each treatment group are indicated. Animal weight was monitored daily. Yellow triangle points the start of the treatment. Brown triangle points the end of the treatment. Results represent the mean \pm SEM of six animals.

Reviewer #3 (Remarks to the Author):

We thank the reviewer for the meticulous, constructive, and positive review of the manuscript.

1. The non-cancer ovarian (NCO) tissue may not be the appropriate normal comparison for EOC, which is thought to arise from fallopian tube epithelium.

Response: To avoid confusion and to improve clarity of this manuscript, we have now removed the data comparing pBADSer99 in normal ovarian tissue to ovarian carcinoma. The removal of this data does not impact on the conclusions of the manuscript as the associations of pBADSer99 with clinicopathologic and survival data of the OC cohort remains.

We have now incorporated the changes in Figure 1, Figure legends and Results section of the revised manuscript as below:

Line 89-102:

“Immunohistochemistry (IHC) was utilized to determine the level of pBADS99 and expression of BAD in EOC tissue specimens. The potential association between pBADS99 and clinicopathological characteristics of EOC patients was determined. As observed in Figure 1A, higher levels of pBADS99 positively correlated with higher labeling of Ki67 ($\rho = 0.274$, $p = 0.014^*$) and higher International Federation of Gynecology and Obstetrics (FIGO) stage ($\rho = 0.629$, $p < 0.001^{***}$) (Figure 1A). Kaplan-Meier survival analysis was also performed in the EOC patient cohort, which sub-categorized the level of BADS99 as low or high using a receiver operating characteristic curve (SI 1A&B). It was observed that EOC patients with higher tumor levels of pBADS99 (IRS=6-12) (n=51) exhibited significantly poorer progression-free survival (PFS) and overall survival (OS) ($p < 0.05$) compared to those patients whose tumors possess no/low levels of pBADS99 (IRS=0-4) (n=29) (Figure 1B). Furthermore, Cox regression analysis demonstrated a significant association of high tumor pBADS99 levels with decreased PFS and OS (SI 1C). No significant correlation was observed between BAD protein expression and EOC patient PFS or OS (SI 1D). Thus, phosphorylation of BADS99 is increased in EOC and predicts poor survival outcome of patients.”

Figure 1.

A. Clinicopathological association with pBADS99

Cohort	category	N	%	ρ
Ages at diagnosis	≤55	48	60.0	-0.187 (p=0.097)
	≥55	32	40.0	
	1	9	11.3	
FIGO ^a stage	2	34	42.5	0.629** (p=0.000)
	3	33	41.2	
	4	4	5.0	
	negative	21	26.3	
Lymph node metastasis	yes	30	37.5	0.148 (p=0.190)
	no	50	62.5	
Ki67	low	15	18.8	0.274* (p=0.014)
	moderate	27	33.8	
	strong	17	21.3	
TP53	Negative	35	35	-0.069 (p=0.544)
	Positive	65	65	

^a The International Federation of Gynecology and Obstetrics.
* Correlation is significant at the 0.05 level (2-tailed)
** Correlation is significant at the 0.01 level (2-tailed)
 ρ Correlation Coefficient with pBADS99 expression

B. Kaplan-Meier analysis

Figure 1: Increased phosphorylation of BAD at S99 residue (pBADS99) in epithelial ovarian cancer (EOC) is significantly associated with reduced survival of patients.

(A) Spearman correlation analysis between pBADS99 expression and clinicopathological features of the EOC patient cohort. (B) Relapse and survival times were compared by Kaplan–Meier analysis and overall survival (OS) or progression free survival (PFS) were tested for significance by using the log rank test. The immunoreactive score (IRS) 0 to 4 categorized was categorized as low pBADS99 (n=29) and IRS 6 to 12 as high pBADS99 respectively (n=51) as described in materials and methods section.

2. There is concern about the EOC cell lines. Of the EOC cell lines, only 3 of the 8 represented had TP53 mutations. These EOC cell lines do not represent the most common type of EOC, high grade serous histology (Domcke, Nat Comm 2013). In addition, A2780 and A2780cp do not have BRCA mutations in the CCLE.

Response: We acknowledge the reviewer’s concern. In this study, there are 4/8 (CAOV2/OVCAR-3/OVCAR433/HEY2) cell lines that possess TP53 mutation (associated with loss of TP53 function), 3/8 (A2780/A2780cisR/AFC) cell lines possess wild-type TP53 protein and in 1/8 (Anglne) cell lines the TP53 status is not reported [34, 35]. 4/8 (CAOV2/OVCAR-3/HEY2/Anglne) cell lines are high grade serous ovarian cancer. Moreover, consistent with the most common type of EOC, all 4 PDOs were diagnosed as high grade serous ovarian cancer. The detailed information of the cell lines was summarized in SI 6A and SI 9 as described below.

In this study, we examined if combined inhibition of BAD phosphorylation and PARP activity would be efficacious in EOC regardless of mutational status or OC subtype. Therefore, diverse

EOC *in vitro* models were included. Previous studies reported that A2780 cells possess a BRCA2 mutation (loss of function) using at least 2 different methodologies [36, 37] as illustrated below. Since the A2780 cell line used herein is obtained from Singapore Ovarian Cancer Library (SGOCL) [38], which has been established as per European Collection of Cell Cultures (ECACC) guidelines, we agree that A2780 cells from ECACC do not contain BRCA (1/2) mutations. However, A2780cisR cells possess BRCA1 mutation (loss of function), as described in the article Beaufort, C.M., et al., PLoS One, 2014 (also see below). We also reconfirmed the BRCA status of A2780 and A2780cisR cell lines with SGOCL.

Moreover, we have re-analyzed the data excluding the BRCA status of A2780 and A2780cisR. The significant association between BRCA status and pBADS99 still exists after excluding the BRCA- status data associated with A2780/A2780cisR cells. We have now incorporated the changes in Figure 1D&E and SI 9A of the revised manuscript as below:

Beaufort, C.M., et al., Ovarian cancer cell line panel (OCCP): clinical importance of *in vitro* morphological subtypes. PLoS One, 2014.

Figure 1D. pBADS99/BAD levels in EOC cells lines

Figure 1D: Western blot analysis was used to assess the levels of pBADS75 and pBADS99 and BAD expression in EOC cell lines. Soluble whole cell extracts were run on an SDS-PAGE and immunoblotted as described in materials and methods. β-ACTIN was used as input control for cell lysate. The sizes of detected protein bands in kDa are shown on the left side. Densitometries of protein bands were subsequently determined using ImageJ software (<https://imagej.nih.gov/ij/>). Below: Specific gene mutations in the EOC cell lines are indicated by the green colored square, described in materials and methods.

Figure 1E. pBADS99 in EOC cell lines

Figure 1E: Association between pBADS99 and BRCA-deficiency or Platinum-resistance in EOC cells.

Supplementary information 6.

A. Clinical information of PDO Patients

Patient ID	Tissue type for PDO	Pathological subtype	Age at diagnosis (years)	FIGO stage	Surgery	Chemotherapy	metastasis	Family history
PDO1	Metastatic ascites	HGSOC	54	FIGO IV	yes	TP (8 cycles)	peritoneal liver	N
PDO2	Primary debulking	HGSOC	43	FIGO III	yes	NY	peritoneal	N
PDO3	Interval debulking	HGSOC	65	FIGO III	yes	Adjuvant chemotherapy TP (3 cycles)	peritoneal	N
PDO4	Primary debulking	HGSOC	56	FIGO III	yes	NY	peritoneal	N

HGSOC: high grade serous ovarian cancer; FIGO: The International Federation of Gynecology and Obstetrics.
 NY: Not yet, N:No

SI 6A: Clinical information of PDO1-4 patients.

Supplementary information 9.

A. Cancer Cell lines Profiling

Cell	Clinical pathology			Molecular profiling					
	Site	Age at diagnosis (years)	Ethnicity	PI3KCA	PTEN	P53	BRCA	KRAS	
Epithelial Ovarian Cancers (EOC) cell lines	CAOV2 [#] (OVCAR2)	Metastatic ascites	N.D.	Caucasian	N.D.	N.D.	√	N.D.	-
	OVCAR-3 [#]	Metastatic ascites	60	Caucasian	-	-	√	-	-
	OVCAR-433 ^{#*}	Metastatic ascites	N.D.	Caucasian	-	-	√	BRCA2	-
	HEYC2 [*] (HEYA8)	Human ovarian cancer xenograft	N.D.	Caucasian	N.D.	N.D.	√	BRCA1	√
	Anglne [‡]	primary	N.D.	Caucasian	N.D.	N.D.	N.D.	N.D.	N.D.
	A2780 [‡]	primary	N.D.	African	√	√	-	-	-
	A2780CisR ^{‡§}	primary	N.D.	African	√	√	-	BRCA1	-
Patient derived cancer cells	PDO-1 (AFC ^{‡*})	Metastatic ascites	54	Asian	√	-	-	BRCA2	√
	PDO-2 [‡]	Primary	43	Asian	N.D.	N.D.	N.D.	N.D.	N.D.
	PDO-3 [‡]	primary	65	Asian	N.D.	N.D.	N.D.	N.D.	N.D.
	PDO-4 [‡]	primary	56	Asian	N.D.	N.D.	N.D.	N.D.	N.D.

N.D. not determined; * platinum resistant; [#] Metastatic ascites; [‡] primary cell (PC); ascitic fluid cancer cells (AFC); Patient derived organoid (PDO). Cell lines gene profiling information was collected from website: https://cancer.sanger.ac.uk/cell_lines

SI 9A: Clinicopathologic and molecular profiling information of the cell lines used in this study.

B. Cancer Cell lines Source and 2D culture condition

Cells used in this study	Original Histological subtype	Source	Culture medium	
Epithelial Ovarian Cancers (EOC) cell lines	CAOV2 [#] (OVCAR2)	HGS EOC	Kyoto U.	RPMI 1640, 10% FBS, 10 μ g/ml insulin
	OVCAR-3 [#]	HGS EOC	ATCC	RPMI 1640, 20% FBS
	OVCAR-433 ^{#*}	Papillary serous cystadenocarcinoma	Kyoto U.	DMEM, 10% FBS
	HEYC2 (HEYA8)	HGS EOC	Kyoto U.	RPMI 1640, 10% FBS
	Angline [#]	HGS EOC	DSMZ	RPMI 1640, 10% FBS
	A2780 [#]	Endometrioid EOC	ECACC	RPMI 1640, 10% FBS
	A2780CisR [#]	Endometrioid EOC	ECACC	RPMI 1640, 10% FBS, 1 μ M cisplatin
Patient derived cancer cells	PDO-1 (AFC ^{#*})	HGS EOC	HKUSZH	Advanced DMEM(F12), 10% FBS, growth factors
	PDO-2 [#]	HGS EOC	HKUSZH	Advanced DMEM(F12), 10% FBS, growth factors
	PDO-3 [#]	HGS EOC	HKUSZH	Advanced DMEM(F12), 10% FBS, growth factors
	PDO-4 [#]	HGS EOC	HKUSZH	Advanced DMEM(F12), 10% FBS, growth factors

platinum resistant; [#] ascitic fluid cancer cells (AFC); ^{*} primary cell (PC); epithelial ovarian cancer (EOC); HGS: high-grade serous; Kyoto U.: Dr. Noriomi Matsumura Kyoto University

SI 9B: Cancer cell line source and Culture condition of cancer cell lines used in this study.

3. How were the BRCA-deficient commercial cell lines validated?

Response: All EOC cell lines we used in this study were obtained from the SGOCL, an OC cell line library [39]. The SGOCL performed molecular profiling of cell lines and maintain their database [36, 37, 40, 41] as per the guidelines of Sanger Cancer institute (https://cancer.sanger.ac.uk/cell_lines) as mentioned in SI 9. The AFC (PDO1) cell line was established in this laboratory from a patient derived specimen. The pathologist from HKU-SZH performed the histological and molecular profile analysis of AFC cells. The BRCA mutational status of AFC cells were confirmed by DNA sequencing analysis (KingMed, Guangdong, China), as described in SI 9. To clarify the presentation of the information, we have now modified the information in SI 9 of the revised manuscript as described below:

Supplementary information 9.

A. Cancer Cell lines Profiling

Cell	Clinical pathology			Molecular profiling					
	Site	Age at diagnosis (years)	Ethnicity	PI3KCA	PTEN	P53	BRCA	KRAS	
Epithelial Ovarian Cancers (EOC) cell lines	CAOV2 [#] (OVCAR2)	Metastatic ascites	N.D.	Caucasian	N.D.	N.D.	√	N.D.	-
	OVCAR-3 [#]	Metastatic ascites	60	Caucasian	-	-	√	-	-
	OVCAR-433 ^{*#}	Metastatic ascites	N.D.	Caucasian	-	-	√	BRCA2	-
	HEYC2* (HEYA8)	Human ovarian cancer xenograft	N.D.	Caucasian	N.D.	N.D.	√	BRCA1	√
	Angline [§]	primary	N.D.	Caucasian	N.D.	N.D.	N.D.	N.D.	N.D.
	A2780 [§]	primary	N.D.	African	√	√	-	-	-
	A2780CisR ^{*§}	primary	N.D.	African	√	√	-	BRCA1	-
Patient derived cancer cells	PDO-1 (AFC ^{*#})	Metastatic ascites	54	Asian	√	-	-	BRCA2	√
	PDO-2 [§]	Primary	43	Asian	N.D.	N.D.	N.D.	N.D.	N.D.
	PDO-3 [§]	primary	65	Asian	N.D.	N.D.	N.D.	N.D.	N.D.
	PDO-4 [§]	primary	56	Asian	N.D.	N.D.	N.D.	N.D.	N.D.

N.D. not determined; * platinum resistant; # Metastatic ascites; § primary cell (PC); ascitic fluid cancer cells (AFC); Patient derived organoid (PDO). Cell lines gene profiling information was collected from website: https://cancer.sanger.ac.uk/cell_lines

B. Cancer Cell lines Source and 2D culture condition

Cells used in this study	Original Histological subtype	Source	Culture medium	
Epithelial Ovarian Cancers (EOC) cell lines	CAOV2 [#] (OVCAR2)	Kyoto U.	RPMI 1640, 10% FBS, 10 μ g/ml insulin	
	OVCAR-3 [#]	ATCC	RPMI 1640, 20% FBS	
	OVCAR-433 ^{*#}	Papillary serous cystadenocarcinoma	Kyoto U.	DMEM, 10% FBS
	HEYC2 (HEYA8)	HGS EOC	Kyoto U.	RPMI 1640, 10% FBS
	Angline [§]	HGS EOC	DSMZ	RPMI 1640, 10% FBS
	A2780 [§]	Endometrioid EOC	ECACC	RPMI 1640, 10% FBS
	A2780CisR ^{*§}	Endometrioid EOC	ECACC	RPMI 1640, 10% FBS, 1 μ M cisplatin
Patient derived cancer cells	PDO-1 (AFC ^{*#})	HKUSZH	Advanced DMEM(F12), 10% FBS, growth factors	
	PDO-2 [§]	HKUSZH	Advanced DMEM(F12), 10% FBS, growth factors	
	PDO-3 [§]	HKUSZH	Advanced DMEM(F12), 10% FBS, growth factors	
	PDO-4 [§]	HKUSZH	Advanced DMEM(F12), 10% FBS, growth factors	

platinum resistant; # ascitic fluid cancer cells (AFC); § primary cell (PC); epithelial ovarian cancer (EOC); HGS: high-grade serous; Kyoto U.: Dr. Noriomi Matsumura Kyoto University

(A) SI 9A: Clinicopathologic and molecular profiling details of the cell lines used in this study.

(B) SI 9B: Cancer cell line source and 2D Culture condition.

4. PARPi have different PARP trapping capabilities that are associated with potency. It

is difficult to detect these differences based on how the data was presented in

logarithmic form (Fig 2).

Response:

The data is represented in logarithmic scale to focus on the combinational efficacy of NPB with PARPis rather than single agent PARPi efficacy.

The IC₅₀ values of the different PARPis in the different cell lines are presented in Figure2 and SI 3B below and the reviewer would be able to compare the relative potencies of the PARPis in different cell lines therein.

Figure 2. Combination efficacy of NPB and PARP inhibitors in EOC cell lines

Figure 2: NPB synergizes with PARP inhibitors in EOC cells to decrease cell survival: Chou-Talalay synergy analysis using for NPB (N) in combination with and three PARP inhibitors, namely Olaparib (O), Rucaparib (R) and Talazoparib (T), in a panel of EOC cells including CAOV2 (BRCA-proficient/Platinum sensitive), A2780 (BRCA-proficient/Platinum sensitive), A2780cisR (BRCA- proficient /Platinum resistant) and AFC (BRCA-deficient/Platinum-resistant). Cells were treated with the indicated concentration (log10 scale) of NPB and mentioned PARP

inhibitors for 6 days. The percentage of survival fraction was assessed using total cell number. The logarithmic combination index (CI) value corresponding to cell fraction affected (Fa) was determined using the CompuSyn software (<http://www.combosyn.com>) as described in materials and methods. CI value indicates: < 1 synergism; = 1 additive Synergy; > 1 antagonism (n=3). Dose-response curves for a panel of cells treated with the indicated concentration of PARP inhibitors with or without 1 μ M NPB in total cell number assays. Arrow indicates fold reduction in respective PARP inhibitor IC₅₀ in the presence of NPB (n=3). Points are mean of triplicate experiments; bars, \pm SD.

Supplementary information 3.

B. IC₅₀ of NPB, PARPis and Combination

Cell lines	IC ₅₀ \pm SD (μ M)						
	NPB	OLA	OLA+NPB(1 μ M)	RUC	RUC+NPB(1 μ M)	TAL	TAL+NPB(1 μ M)
CAOV2	2.0 \pm 0.35	1.34 \pm 0.21	0.013 \pm 0.01	1.14 \pm 0.33	0.018 \pm 0.002	0.24 \pm 0.05	0.005 \pm 0.0007
OVCAR3	0.66 \pm 0.2	2.19 \pm 0.80	0.022 \pm 0.02	1.48 \pm 0.36	0.054 \pm 0.012	2.16 \pm 0.48	0.24 \pm 0.056
OVCAR433	2.26 \pm 0.3	5.9 \pm 1.23	0.015 \pm 0.03	6.03 \pm 2.04	0.027 \pm 0.003	0.90 \pm 0.37	0.003 \pm 0.001
HEYC2	2.26 \pm 0.3	4.8 \pm 0.82	0.027 \pm 0.03	1.2 \pm 0.39	0.074 \pm 0.016	0.79 \pm 0.22	0.006 \pm 0.0008
Angline	2.04 \pm 0.6	4.27 \pm 0.61	0.036 \pm 0.00	0.25 \pm 0.08	0.012 \pm 0.003	0.14 \pm 0.003	0.007 \pm 0.001
A2780	0.75 \pm 0.2	2.7 \pm 0.03	0.0264 \pm 0.00	1.63 \pm 0.42	0.025 \pm 0.005	0.96 \pm 0.003	0.007 \pm 0.001
A2789CisR	9.58 \pm 1.3	8.2 \pm 1.12	0.11 \pm 0.003	0.87 \pm 0.28	0.042 \pm 0.001	1.77 \pm 0.53	0.028 \pm 0.007
AFC	1.3 \pm 0.2	6.1 \pm 1.17	0.015 \pm 0.00	0.22 \pm 0.47	0.021 \pm 0.004	0.41 \pm 0.12	0.021 \pm 0.05

SI 3B: IC₅₀ was calculated in 8 cell lines treated with the indicated concentration of NPB and PARP inhibitor for 6 days. The presence of 1 μ M NPB led to a significant 10~400-fold decrease in PARP inhibitor IC₅₀. Data are represented as mean \pm SD.

5. Line 24: The statement, “PARP inhibitors are the last line of therapy for recurrent EOC” is misleading. PARP inhibitors are used in primary/frontline and recurrent treatment for EOC. There are many other options for last line of therapy.

Response: In the revised version of the manuscript, we have now modified the statement as described below:

Line 23:

PARPis are one of a last line of therapies for recurrent EOC.

6. Line 85 and Lines 336-337: Although 1/3 of tumor-bearing mice in a single platinum-

resistant PDX model achieved a durable CR, it is overstated to claim that the combination of NPB with PARPi would be “may be curative.”

Response: We have now modified the text in accordance with the reviewer’s suggestion in the revised manuscript as described below:

Line 83-84:

“Herein, the synergistic combination of NPB with PARPis (Olaparib, Rucaparib, and Talazoparib) is exemplified as a possible efficacious regimen for EOC.”

Line 404-408:

“Herein, the combined NPB-Olaparib regimen was demonstrated to be somewhat efficacious in a platinum-resistant PDX model, in which one third of tumor-bearing mice achieved a durable CR without relapse despite cessation of treatment. Hence, a rational and efficacious combination strategy involving combined inhibition of PARP and BAD phosphorylation may ameliorate the prognosis of recurrent EOC patients.”

7. The results are primarily descriptive and are not mechanistic.

Response: In accordance with the reviewer’s comment, we have now incorporated new mechanistic data (SI 5D, 5E and 5F) and a graphical abstract (SI 13) in the revised manuscript which clarifies the mechanism of synergy between NPB and PARPis, as described below:

Line 158-159:

subtitle:

“NPB synergizes with PARP inhibition to stimulate apoptotic cell death attenuating PARPi induced HR repair.”

Line 198-211:

“It is recognized that sub-lethal apoptotic stresses trigger a caspase-dependent increase in γ -H2AX expression [7-11]. As DNA fragmentation is a classic apoptotic hallmark mediated by caspase-activated DNase (CAD), DNA laddering assays were performed. After NPB treatment, apoptotic DNA laddering increased accompanied by caspase-3, PARP cleavage and γ -H2AX expression consistent with previous studies reporting that increased γ -H2AX levels may also indicate apoptosis [9-11] (SI 5D). Transient-transfection of a mutated human *BAD* (*hBADS99A*) into CAOV2 cells also exhibited increased levels of γ -H2AX protein as evaluated by immunofluorescence (SI 5E). Given that NPB treatment diminished PARPi-induced CHEK1/CHEK2 phosphorylation and RAD51, both of which possess important roles in HR, it was therefore determined whether NPB modulated HR. A HR reporter assay demonstrated that NPB treatment suppressed HR efficiency both in BRCA wild type (CAOV2) and BRCA mutant

(AFC) EOC cells (SI 5F). Hence, NPB attenuated the repair of DSBs induced by PARPis by suppression of HR. Thus, NPB synergizes with PARPis in EOC cells to promote DNA damage and impair DNA repair after PARPi treatment.”

Line 344-348:

“Previous studies demonstrated that the proapoptotic BCL-2 family members, Bax and Bid, can enhance the efficiency of cell death by inhibiting HR DNA repair independent of the apoptosis [42]. Therefore, it may be speculated that inhibition of BAD phosphorylation abrogates HR. Indeed, the HR reporter assay confirmed that NPB treatment suppressed HR.”

SI 5D. Caspase-dependent DNA damage

SI 5D: Western blot analysis was used to assess the level of various protein markers and protein activities in CAOV2 cells after treatment with NPB (N) at the indicated concentrations. Soluble whole cell extracts were run on SDS-PAGE and immunoblotted as described in materials and methods. β-ACTIN (ACTB) was used as input control for cell lysate. Genomic DNA and apoptotic DNA fragments were isolated from NPB-treated CAOV2 cells as described in supplementary text and were run on a 2% TAE agarose gel.

Supplementary text (E): Cells were collected by centrifugation, and the cell pellets were lysed by gentle, repeated pipetting in a 2-mL microcentrifuge tube with 0.5 mL cell lysing buffer (50 mM Tris-HCl, pH 7.5, 0.5% Triton X-100, 20 mM EDTA). After centrifugation at 12000 rpm for 10 min to remove cell debris, as well as intact nuclei of non-apoptotic cells, the supernatant was subject to one round of phenol:chloroform:isoamyl alcohol (25:24:1; pH 7.4; 0.5mL) extraction. Apoptotic DNA fragments and contaminating RNA present in the liquid phase were precipitated by adding 50μL 3M sodium acetate and 50μL phenol:chloroform:isoamyl alcohol (25:24:1; pH 7.4; 0.5 mL). After incubation on ice for 5 min, the precipitated nucleic acids were pelleted by centrifugation at 12000rpm for 10 min. The pellet was resuspended in RNaseA buffer (Promega, Madison, WI, USA) by incubation at 37°C for 30 min or longer. After centrifugation at 12000rpm

for 10 min, the pellet was resuspended in phenol:chloroform:isoamyl alcohol (25:24:1; pH 7.4; 0.5mL) and this step was repeated. Apoptotic DNA fragments were resolved by 2 % TAE agarose gel electrophoresis (17, 18). Genomic DNA was extracted following Wizard® Genomic DNA Purification Kit instructions (Promega, Madison, WI, USA).

SI 5E. IF analysis

SI 5E: Forced expression of *Flag-hBAD599A* in CAOV2 cells exhibited no p-BADS99 immunoreactive signal and induced γ H2AX expression as confirmed by immunofluorescent analysis, Scale bar, 20 μ m.

SI 5F. Homologous Recombination (HR) reporter assay

SI 5F: The *HR-GFP reporter* consists of two copies of mutated *GFP genes*. Cells were transfected with the I-SceI expressing vector to induce DSBs which may be repaired by HR using iGFP as a template and restore a functional *GFP gene*. The percentage of GFP-positive (GFP+) cells was measured using immunofluorescence and flow cytometry as an indicator of HR efficiency. DSBs introduced by I-SceI endonuclease HR exhibited increased levels of BAD599 in EOC cells as

determined by western blot analysis. A 50-70% reduction is observed in GFP positive cells after NPB treatment in BRCA wild type and mutant EOC cells (n = 3), Scale bar, 50µm.

Supplementary text (D): **HR reporter assay and sensitivity assays:** To establish CAOV2-DR-GFP and AFC-DR-GFP cells, the *pDR-GFP* plasmid was stably integrated into CAOV2 and AFC cell lines as previously described [4]. After 48 hours of pretreatment with NPB, CAOV2-DR-GFP and AFC-DR-GFP (1×10^6) cells were transfected by electroporation using FuGENE HD (Promega, US) with 2µg pCMV-I-SceI, respectively. The cells were assessed for green fluorescence emission using flow cytometry. Results were corrected according to the transfection efficiency of the cell line.

SI 13

SI 13: Interactions between the intrinsic apoptosis and DNA damage/repair processes in EOC provide a rational combination strategy for inhibition of PARP and BADSer99 phosphorylation

Poly (ADP-Ribose) polymerase (PARP), as a major component of the DNA damage response (DDR), is recruited to initiate cellular responses to single-strand DNA breaks (SSB). PARP inhibition, therefore, results in cell death through synthetic lethality in EOC cells with homologous recombination (HR) deficiency (eg. loss of BRCA1/2 function). PARP inhibition in EOC cells has been reported to stimulate the activation of RAS/MEK/ERK and PI3K/AKT/mTOR signaling pathways [43-46]. The RAS/MEK/ERK and PI3K/AKT/mTOR signaling cascades converge at BCL-2 associated death promoter (BAD) and participate in BAD phosphorylation at

Serine (S)75 residue through the p44/42 MAP kinase pathway [47] and at S99 residue through activation of PI3K/AKT/mTOR [48, 49] to promote cancer cell survival, which is one of the mechanisms of drug resistance and cancer relapse. Treatment of HR-proficient EOC with BAD phosphorylation inhibitor NPB induced intrinsic apoptosis and attenuated HR efficiency. In response to DNA damage, TP53 upregulates *BAD* transcription. Dephosphorylated BAD also heterodimerizes with TP53 [50] or with and sequesters BCL2, BCL2-XL, and BCL-W with consequent intrinsic apoptosis through disruption of mitochondrial membrane potential [51]. The functional interaction between the intrinsic apoptosis and DNA damage processes in EOC imply that a combination of inhibition of both BAD phosphorylation and PARP may ameliorate outcomes in recurrent EOC.

Reference

1. Pandey, V., et al., *Discovery of a small-molecule inhibitor of specific serine residue BAD phosphorylation*. Proc. Natl. Acad. Sci. U. S. A., 2018. **115**(44): p. E10505.
2. Pandey, V., et al., *Discovery of a small-molecule inhibitor of specific serine residue BAD phosphorylation*. Proc Natl Acad Sci U S A, 2018. **115**(44): p. E10505-E10514.
3. Konstantinopoulos, P.A., S. Lheureux, and K.N. Moore, *PARP Inhibitors for Ovarian Cancer: Current Indications, Future Combinations, and Novel Assets in Development to Target DNA Damage Repair*. Am Soc Clin Oncol Educ Book, 2020. **40**: p. 1-16.
4. Pierce, A.J., et al., *XRCC3 promotes homology-directed repair of DNA damage in mammalian cells*. Genes Dev, 1999. **13**(20): p. 2633-8.
5. Wilson, A., et al., *Transforming early pharmaceutical assessment of genotoxicity: applying statistical learning to a high throughput, multi end point in vitro micronucleus assay*. Sci Rep, 2021. **11**(1): p. 2535.
6. Benigni, R. and C. Bossa, *Data-based review of QSARs for predicting genotoxicity: the state of the art*. Mutagenesis, 2019. **34**(1): p. 17-23.
7. Ichim, G., et al., *Limited mitochondrial permeabilization causes DNA damage and genomic instability in the absence of cell death*. Mol Cell, 2015. **57**(5): p. 860-872.

8. Luedtke, D.A., et al., *Inhibition of Mcl-1 enhances cell death induced by the Bcl-2-selective inhibitor ABT-199 in acute myeloid leukemia cells*. *Signal Transduct Target Ther*, 2017. **2**: p. 17012.
9. Rogakou, E.P., et al., *Initiation of DNA fragmentation during apoptosis induces phosphorylation of H2AX histone at serine 139*. *J Biol Chem*, 2000. **275**(13): p. 9390-5.
10. Plesca, D., S. Mazumder, and A. Almasan, *DNA damage response and apoptosis*. *Methods Enzymol*, 2008. **446**: p. 107-22.
11. Tanaka, T., et al., *Cytometry of ATM activation and histone H2AX phosphorylation to estimate extent of DNA damage induced by exogenous agents*. *Cytometry A*, 2007. **71**(9): p. 648-61.
12. Murai, J., et al., *Trapping of PARP1 and PARP2 by Clinical PARP Inhibitors*. *Cancer Res*, 2012. **72**(21): p. 5588-99.
13. Morales, J., et al., *Review of poly (ADP-ribose) polymerase (PARP) mechanisms of action and rationale for targeting in cancer and other diseases*. *Crit Rev Eukaryot Gene Expr*, 2014. **24**(1): p. 15-28.
14. Min, A. and S.A. Im, *PARP Inhibitors as Therapeutics: Beyond Modulation of PARylation*. *Cancers (Basel)*, 2020. **12**(2).
15. Min, A. and S.-A. Im, *PARP Inhibitors as Therapeutics: Beyond Modulation of PARylation*. *Cancers*, 2020. **12**(2): p. 394.
16. Satoh, M.S. and T. Lindahl, *Role of poly(ADP-ribose) formation in DNA repair*. *Nature*, 1992. **356**(6367): p. 356-8.

17. Menear, K.A., et al., *4-[3-(4-cyclopropanecarbonylpiperazine-1-carbonyl)-4-fluorobenzyl]-2H-phthalazin-1-one: a novel bioavailable inhibitor of poly(ADP-ribose) polymerase-1*. J Med Chem, 2008. **51**(20): p. 6581-91.
18. Thomas, H.D., et al., *Preclinical selection of a novel poly(ADP-ribose) polymerase inhibitor for clinical trial*. Mol Cancer Ther, 2007. **6**(3): p. 945-56.
19. Murthy, P. and F. Muggia, *PARP inhibitors: clinical development, emerging differences, and the current therapeutic issues*. Cancer Drug Resistance, 2019. **2**(3): p. 665-679.
20. Shen, Y., et al., *BMN 673, a novel and highly potent PARP1/2 inhibitor for the treatment of human cancers with DNA repair deficiency*. Clin Cancer Res, 2013. **19**(18): p. 5003-15.
21. Murai, J., et al., *Stereospecific PARP trapping by BMN 673 and comparison with olaparib and rucaparib*. Mol Cancer Ther, 2014. **13**(2): p. 433-43.
22. Jelinic, P. and D.A. Levine, *New insights into PARP inhibitors' effect on cell cycle and homology-directed DNA damage repair*. Mol Cancer Ther, 2014. **13**(6): p. 1645-54.
23. Ledermann, J., et al., *Olaparib maintenance therapy in platinum-sensitive relapsed ovarian cancer*. N Engl J Med, 2012. **366**(15): p. 1382-92.
24. Poveda, A., et al., *Olaparib tablets as maintenance therapy in patients with platinum-sensitive relapsed ovarian cancer and a BRCA1/2 mutation (SOLO2/ENGOT-Ov21): a final analysis of a double-blind, randomised, placebo-controlled, phase 3 trial*. Lancet Oncol, 2021. **22**(5): p. 620-631.
25. Ledermann, J.A., et al., *Rucaparib for patients with platinum-sensitive, recurrent*

- ovarian carcinoma (ARIEL3): post-progression outcomes and updated safety results from a randomised, placebo-controlled, phase 3 trial.* The Lancet. Oncology, 2020. **21**(5): p. 710-722.
26. Dhawan, M.S., et al., *Differential Toxicity in Patients with and without DNA Repair Mutations: Phase I Study of Carboplatin and Talazoparib in Advanced Solid Tumors.* Clinical Cancer Research, 2017. **23**(21): p. 6400.
 27. Xiang, T., et al., *Targeting the Akt/mTOR pathway in Brca1-deficient cancers.* Oncogene, 2011. **30**(21): p. 2443-50.
 28. Xiang, T., et al., *Negative Regulation of AKT Activation by BRCA1.* Cancer Res, 2008. **68**(24): p. 10040-4.
 29. Alexandre, M., Z.P. Lin, and E.S. Ratner, *Abstract 850: The Pi3K/Akt pathway mediates epithelial-mesenchymal transition (EMT) and malignant progression in BRCA-defective epithelial ovarian cancer.* Cancer Research, 2017. **77**(13 Supplement): p. 850-850.
 30. Laulier, C., et al., *Bcl-2 inhibits nuclear homologous recombination by localizing BRCA1 to the endomembranes.* Cancer Res, 2011. **71**(10): p. 3590-602.
 31. Bharatham, N., S.W. Chi, and H.S. Yoon, *Molecular basis of Bcl-X(L)-p53 interaction: insights from molecular dynamics simulations.* PLoS One, 2011. **6**(10): p. e26014.
 32. Jiang, P., et al., *The Bad guy cooperates with good cop p53: Bad is transcriptionally up-regulated by p53 and forms a Bad/p53 complex at the mitochondria to induce apoptosis.* Molecular and cellular biology, 2006. **26**(23): p. 9071-9082.
 33. Remmele, W. and H.E. Stegner, *[Recommendation for uniform definition of an*

immunoreactive score (IRS) for immunohistochemical estrogen receptor detection (ER-ICA) in breast cancer tissue. Pathologie, 1987. **8**(3): p. 138-40.

34. Mullany, L.K., et al., *Specific TP53 Mutants Overrepresented in Ovarian Cancer Impact CNV, TP53 Activity, Responses to Nutlin-3a, and Cell Survival*. Neoplasia, 2015. **17**(10): p. 789-803.
35. Bhatt, M., et al., *Drug-dependent functionalization of wild-type and mutant p53 in cisplatin-resistant human ovarian tumor cells*. Oncotarget, 2017. **8**(7): p. 10905-10918.
36. Beaufort, C.M., et al., *Ovarian cancer cell line panel (OCCP): clinical importance of in vitro morphological subtypes*. PLoS One, 2014. **9**(9): p. e103988.
37. Tudrej, P., et al., *Establishment and Characterization of the Novel High-Grade Serous Ovarian Cancer Cell Line OVPA8*. Int J Mol Sci, 2018. **19**(7).
38. Huang, R.Y.J., et al., *An EMT spectrum defines an anoikis-resistant and spheroidogenic intermediate mesenchymal state that is sensitive to e-cadherin restoration by a src-kinase inhibitor, saracatinib (AZD0530)*. Cell Death & Disease, 2013. **4**(11): p. e915-e915.
39. Huang, R.Y., et al., *An EMT spectrum defines an anoikis-resistant and spheroidogenic intermediate mesenchymal state that is sensitive to e-cadherin restoration by a src-kinase inhibitor, saracatinib (AZD0530)*. Cell Death Dis, 2013. **4**: p. e915.
40. Hamilton, T.C., et al., *Characterization of a human ovarian carcinoma cell line (NIH:OVCA8-3) with androgen and estrogen receptors*. Cancer Res, 1983. **43**(11): p. 5379-89.

41. Stordal, B., et al., *BRCA1/2 mutation analysis in 41 ovarian cell lines reveals only one functionally deleterious BRCA1 mutation*. Mol Oncol, 2013. **7**(3): p. 567-79.
42. Dumay, A., et al., *Bax and Bid, two proapoptotic Bcl-2 family members, inhibit homologous recombination, independently of apoptosis regulation*. Oncogene, 2006. **25**(22): p. 3196-205.
43. Tapodi, A., et al., *Pivotal role of Akt activation in mitochondrial protection and cell survival by poly(ADP-ribose) polymerase-1 inhibition in oxidative stress*. Journal of Biological Chemistry, 2005. **280**(42): p. 35767-35775.
44. Veres, B., et al., *Decrease of the inflammatory response and induction of the Akt/protein kinase B pathway by poly-(ADP-ribose) polymerase 1 inhibitor in endotoxin-induced septic shock*. Biochemical Pharmacology, 2003. **65**(8): p. 1373-1382.
45. Veres, B., et al., *Regulation of kinase cascades and transcription factors by a poly(ADP-ribose) polymerase-1 inhibitor, 4-hydroxyquinazoline, in lipopolysaccharide-induced inflammation in mice*. Journal of Pharmacology and Experimental Therapeutics, 2004. **310**(1): p. 247-255.
46. Sun, C., et al., *Rational combination therapy with PARP and MEK inhibitors capitalizes on therapeutic liabilities in RAS mutant cancers*. Sci Transl Med, 2017. **9**(392).
47. Tan, Y., et al., *p90(RSK) blocks bad-mediated cell death via a protein kinase C-dependent pathway*. J Biol Chem, 1999. **274**(49): p. 34859-67.
48. Manning, B.D. and L.C. Cantley, *AKT/PKB signaling: navigating downstream*. Cell,

2007. **129**(7): p. 1261-74.

49. Harada, H., et al., *p70S6 kinase signals cell survival as well as growth, inactivating the pro-apoptotic molecule BAD*. Proc Natl Acad Sci U S A, 2001. **98**(17): p. 9666-70.
50. Jiang, P., W. Du, and M. Wu, *p53 and Bad: remote strangers become close friends*. Cell Res, 2007. **17**(4): p. 283-5.
51. Adachi, M. and K. Imai, *The proapoptotic BH3-only protein BAD transduces cell death signals independently of its interaction with Bcl-2*. Cell Death Differ, 2002. **9**(11): p. 1240-7.

Reviewers' comments:

Reviewer #1 (Remarks to the Author):

Authors have sufficiently addressed my initial concerns. This revised version contains substantial improvements.

However, one of the new data on DR-GFP assay (SI 5F) presented in the rebuttal raises a different concern related to the technical aspect of this experiment. Original papers by Pierce et al (Genes Dev, 1999) and Bennardo et al (PLOS Genetics, 2008) and many subsequent studies reported low HR proficiency (up to 10% of GFP+ cells) using this assay. In SI 5F, authors reported more than 50% of GFP+ cells. Authors need to check if there is a technical issues with the experiment.

Reviewer #2 (Remarks to the Author):

The authors responded adequately to many but not all comments. Concerns that remain unaddressed are:

Reviewer's comment 4:

"What is the mechanism by which NPB inhibits phosphorylation of BAD? Is this via blocking accessibility of BAD-kinases to S99, or via inhibiting BAD-kinases directly, or stimulating BAD-phosphatase activity?" The authors' rebuttal (pasted verbatim below) should be included in the discussion along with clarifying the point they were trying to make with regard to "steric hindrance". Specifically, can they explain, Steric hindrance of what? Do the authors propose that NPB binds to BAD and shields S99 from kinases? Do they have evidence that NPB binds at the S99 region?

Authors' original Response that is not clear on "steric hindrance": The proposed mechanism of NPB inhibition of phosphorylation of BAD has been previously described [2]. Briefly, NPB does not affect phosphorylation levels of other signaling molecules nor the predominantly p44/42 MAP kinase phosphorylated Ser75 residue on human BAD [1]. Although NPB is also predicted to interact with BCL-2, NPB interacts directly with BAD in the absence of BCL-2 as predicted by Laplacian-modified naive Bayesian classifier algorithm analysis and as previously demonstrated using surface plasmon resonance (SPR) [2]. Based on in silico docking analyses, the dichlorophenyl moiety of NPB was predicted to occupy an additional hydrophobic side pocket within the human BCL-2/BAD interface formed by the side-chains of Leu97, Trp144, and Phe198 residues, in comparison to BCL-2 inhibitors which do not. Additionally, no effect of NPB on AKT kinase activity was observed after cellular treatment with NPB nor in in vitro kinase assays [2]. Thus, NPB has been proposed to inhibit phosphorylation of BADS99 by steric hindrance independent of the upstream AKT-kinase or other phosphorylation activities [2].

Reviewer's comment 5: "Does NPB increase the BAD-specific apoptotic pathway? One would expect increased BAD:Bcl-2-like interactions."

The authors did not address this comment. They should show BAD immunoprecipitation and probe for Bcl-2/XL/w interaction.

Reviewer's comment 5: "Similarly, the authors demonstrate that mutant BAD-S99A decreases cell survival. The authors should show that this tagged construct is appropriately located within the cell."

The authors provide new data in SI 1H. However, there are numerous inconsistencies with both the data and the authors' conclusion.

1. CAOV2 cells have endogenous BAD (strong signal with anti-BAD immunofluorescence in SI 1H upper panels). That endogenous BAD is phosphorylated (pBADS99 signal in WB SI 1G and SI 1H lower panel). Why then is there no pBADS99 signal in the immunofluorescence images when cells are transfected with hBADS99AFLAG (H lowest panel)? Clearly, there is pBADS99 signal in the corresponding samples analyzed by western blot shown in G. Why does transfection of hBADS99A-Flag eliminate endogenous pBADS99 in the immunofluorescence?
2. The localization of hBADS99A-Flag is not consistent (compare SI 1H Flag-tag signal in upper versus lower panels). The upper panel Flag-tag shows BAD dispersed in the cytosol. The lower shows BAD polarized to one side of the membrane. The upper and lower samples should be the same in that channel. What is the reason for this difference?
3. The western blot (SI 1G) attempts to discriminate transfected BADS99A-Flag from endogenous BAD. In the Flag-BAD probed blot, why is endogenous BAD, (lower band) increased with transfection?

Reviewer's original comment 6: "Controls are needed for some of the immunohistochemistry data. Importantly, the specificities of the anti-P-BAD-S99 and anti-BAD antibodies are not clear."

(a) The authors provide new data in SI 8C however there are some questions regarding this data. In SI 8C, the control scrambled oligo samples appear to be stained differently than siRNA-BAD. There is a clear difference in hematoxylin (blue) staining between the upper and lower panels. Additionally, the siRNA-BAD alters the cell size and shape. The cells have a different morphology than the control scrambled-oligo treatment. Clarification is needed.

(b) The authors add: "To further exemplify the specificity of the BAD and pBADS99 antibody, we also transfected CAOV2 cells with the flag-tagged wild type hBAD and hBADS99A as part of Figure (SI 1H). There was no p-BADS99 immunoreactive signal observed in cells transfected with the phosphorylation deficient BADS99A mutant whereas control and hBAD transfected cells exhibited pBADSer99 immunoreactivity". However, again, this observation is a concern. Why does the pBADS99 antibody not react with the endogenous pBAD upon BADS99A-Flag transfection? The antibody clearly reacts with endogenous BAD in the vector control samples. The authors should explain this.

Reviewer's comment 10: In Fig. 4C, the authors conclude that caspase 7 cleavage is elevated in response to N/O vs N or O alone. This is not convincing on the representative western blot." The authors incorporated a new cleaved-CASPASE7 blot in Figure 4C of the revised manuscript. This new blot must be accompanied with a loading control from that same membrane. The b-actin loading control in the original figure is not relevant for a different membrane.

Reviewer's original comment 12: "In Figure 5A, xenograft experiment, how was the statistical analysis done?" The authors respond, "The statistical differences between the treatment groups were compared using an unpaired two-tailed Student's t test based on endpoint tumor volume." Statistical analysis using Student's t test does not account for error of multiple comparisons. An appropriate multiple comparison test should be used. Possibly a one-way ANOVA followed by a Tukey's multiple comparison test is appropriate, but the authors must first test the distribution of their data.

Reviewer #3 (Remarks to the Author):

Poly (ADP-ribose) polymerase inhibitors (PARPi) are approved as maintenance therapy for the treatment of primary and recurrent epithelial ovarian cancer (EOC). New combination treatments are needed to improve responses to PARPi. Here, a combination of PARPi and a small molecule (NPB), which inhibits phosphorylation of BAD, is investigated.

The rationale for targeting BAD is that it is a key factor involved in apoptosis and it is downstream of RAS/MEK/ERK and PI3K/AKT/mTOR pathways that are implicated in sensitivity to PARPi.

This reviewer's concerns have been appropriately addressed.

Reviewer #1 (Remarks to the Author):

Authors have sufficiently addressed my initial concerns. This revised version contains substantial improvements.

However, one of the new data on DR-GFP assay (SI 5F) presented in the rebuttal raises a different concern related to the technical aspect of this experiment. Original papers by Pierce et al (Genes Dev, 1999) and Bennardo et al (PLOS Genetics, 2008) and many subsequent studies reported low HR proficiency (up to 10% of GFP+ cells) using this assay. In SI 5F, authors reported more than 50% of GFP+ cells. Authors need to check if there is a technical issue with the experiment.

Response: We thank the reviewer for the suggestion. We re-examined technical details of the experiment and indeed observed elevated levels of the GFP-positive cell population due to instrument settings. Based on the previous published literature [1, 2], we readjusted the analysis setting of the flow cytometer. The new and consistent data is represented below and incorporated as SI 5G in the revised manuscript as below;

F. Homologous Recombination (HR) reporter assay

SI 5G: The HR-GFP reporter consists of two copies of mutant GFP genes. Cells were transfected with I-SceI expressing vector to induce DSBs and which may be repaired by HR using iGFP as a template and restore a functional GFP gene. The percentage of GFP-positive (GFP+) cells was measured using flow cytometry as an indicator of HR efficiency. Cells with DSBs introduced by I-SceI endonuclease HR exhibit increased levels of pBAD599 in EOC cells as determined by western blot analysis. A reduction is observed in GFP positive cells after NPB treatment in BRCA wild type and mutant EOC cells.

Reviewer #2 (Remarks to the Author):

The authors responded adequately to many but not all comments. Concerns that remain unaddressed are:

Reviewer's comment 4: "What is the mechanism by which NPB inhibits phosphorylation of BAD? Is this via blocking accessibility of BAD-kinases to S99, or via inhibiting BAD-kinases directly, or stimulating BAD-phosphatase activity?" The authors' rebuttal (pasted verbatim below) should be included in the discussion along with clarifying the point they were trying to make with regard to "steric hindrance". Specifically, can they explain, Steric hindrance of what? Do the authors propose that NPB binds to BAD and shields S99 from kinases? Do they have evidence that NPB binds at the S99 region?

Authors' original Response that is not clear on "steric hindrance": The proposed mechanism of NPB inhibition of phosphorylation of BAD has been previously described [2]. Briefly, NPB does not affect phosphorylation levels of other signaling molecules nor the predominantly p44/42 MAP kinase phosphorylated Ser75 residue on human BAD [1]. Although NPB is also predicted to interact with BCL-2, NPB interacts directly with BAD in the absence of BCL-2 as predicted by Laplacian-modified naive Bayesian classifier algorithm analysis and as previously demonstrated using surface plasmon resonance (SPR) [2]. Based on *in silico* docking analyses, the dichlorophenyl moiety of NPB was predicted to occupy an additional hydrophobic side pocket within the human BCL-2/BAD interface formed by the side-chains of Leu97, Trp144, and Phe198 residues, in comparison to BCL-2 inhibitors which do not. Additionally, no effect of NPB on AKT kinase activity was observed after cellular treatment with NPB nor in *in vitro* kinase assays [2]. Thus, NPB has been proposed to inhibit phosphorylation of BADS99 by steric hindrance independent of the upstream AKT-kinase or other phosphorylation activities [2].

Response: Based on bioinformatic (Laplacian-modified naive Bayesian classifier algorithm) and physio-chemical (surface plasmon resonance) analysis previously described [3], NPB interacts directly with BAD protein in the absence of BCL-2. Also, NPB does not affect phosphorylation levels of other signaling molecules (including AKT kinase activity) nor the predominantly p44/42 MAP kinase phosphorylated Ser75 residue on human BAD, demonstrated using multiple *in vitro* biochemical assays [3]. In the absence of a co-crystal structure of BAD and BCL-2, or indeed even a crystal structure for BAD, we are not currently able to definitively delineate a mechanism for NPB, but merely postulate a mechanism based on modeling (docking analyses) and correlated to observed results. Given the tight word limit on the research manuscript, we have now briefly described in the text of the revised manuscript as suggested by the reviewer as below;

Line 81-88:

"Based on bioinformatic (Laplacian-modified naive Bayesian classifier algorithm) and physio-chemical (surface plasmon resonance) analysis previously described [3], NPB interacts directly with BAD protein in the absence of BCL-2. Also, NPB does not affect

phosphorylation levels of other signaling molecules (including AKT kinase activity) nor the predominantly p44/42 MAP kinase phosphorylated Ser75 residue on human BAD, demonstrated using multiple *in vitro* biochemical assays [3]. Thus, NPB has been proposed to hinder phosphorylation of BADS99 independent of the upstream AKT-kinase or other phosphorylation activities [3].”

Reviewer’s comment 5: “Does NPB increase the BAD-specific apoptotic pathway? One would expect increased BAD: Bcl-2-like interactions.” The authors did not address this comment. They should show BAD immunoprecipitation and probe for Bcl-2/XL/w interaction.

Response: As per the reviewer’s request, we have now performed co-immunoprecipitation assays to examine BAD interaction with BCL-2, BCL-XL or BCL-W after treatment of CAOV-2 cells with NPB. We have now incorporated this new data in the results section of the revised manuscript (SI 5D), as described below;

Line 207-210:

“It has been reported that dephosphorylated BAD sequesters BCL-2, BCL-XL, and BCL-W, which results in BAK/BAX activation and apoptosis [4]. Treatment of CAOV-2 cells with NPB enhanced the interaction of BAD with BCL-2, BCL-XL or BCL-W as observed by co-immunoprecipitation (co-IP) assays (SI 5D)”.

D. Co-immunoprecipitation of BAD with BCL-2, BCL-XL or BCL-W in CAOV2 cells

SI 5D: Effect of NPB treatment on co-immunoprecipitation (co-IP) of BAD and BCL-2, BCL-XL or BCL-W in CAOV2 cells. Lysate lane: whole cell lysate of vehicle and NPB treated CAOV2 cells; Iso-IgG lane: co-IP using a Iso-IgG-conjugated beads; co-IP BAD lane: IP using BAD antibody conjugated beads. Heavy chain is denoted as HC.

Supplementary text (G): BAD co-immunoprecipitation (co-IP) assay was performed using BAD-antibody conjugated magnetic beads from Universal IP/Co-IP Toolkit (Abbkine) as per manufacturer's instructions (Cat #: KTD104-EN, Abbikine, China).

Reviewer's comment 5: "Similarly, the authors demonstrate that mutant BAD-S99A decreases cell survival. The authors should show that this tagged construct is appropriately located within the cell." The authors provide new data in SI 1H. However, there are numerous inconsistencies with both the data and the authors' conclusion.

Response: We have reperformed this section of data to address the reviewer's concerns and included new experimental data using wild type *flag-tagged* BAD as a comparator in all assays. It should be noted that the antibodies used in western blot and

immunofluorescence are different as indicated (Supplementary text G). Please see specific responses below.

1. *CAOV2 cells have endogenous BAD (strong signal with anti-BAD immunofluorescence in SI 1H upper panels). That endogenous BAD is phosphorylated (pBADS99 signal in WB SI 1G and SI 1H lower panel). Why then is there no pBADS99 signal in the immunofluorescence images when cells are transfected with hBADS99AFLAG (H lowest panel)? Clearly, there is pBADS99 signal in the corresponding samples analyzed by western blot shown in G. Why does transfection of hBADS99A-Flag eliminate endogenous pBADS99 in the immunofluorescence?*

Response: As mentioned above, we have reperformed the immunofluorescence experiments under controlled and consistent conditions, with a constant excitation period and intensity combined with higher resolution and generated new images. As shown in the new Figure SI 1H, forced expression of *hBAD-flag* in CAOV2 cells increased both BAD and pBADS99 immunofluorescence compared to Vector transfected cells. Forced expression of *hBADS99A-Flag* increased hBAD immunofluorescence compared to vector transfected cells and produced equivalently increased hBAD and Flag immunofluorescence, as was observed in *hBAD-flag* transfected cells compared to vector transfected cells. Forced expression of *hBADS99-Flag* did not appreciably decrease hBADS99 immunofluorescence compared to vector transfected cells but pBADS99 immunofluorescence was significantly less than that observed in *hBAD-flag* transfected cells. These results are consistent with the new western blot data now included below and indicates that the ratio of total pBADSer99 to total BAD is decreased with the hBADS99 mutation. Please see the improved figure below;

Supplementary information 1.

G. Transient transfection of *hBADS99A* in CAOV2 cells

H. Confocal microscopy of tagged *hBAD-Flag* and *hBADS99A-Flag* construct in CAOV2

SI 1G: Forced expression of *Flag-hBAD* and *Flag-hBADS99A* in CAOV2 was confirmed by western blot analysis. Forced expression of *Flag-hBADS99A* in CAOV2 cells reduced cell viability and induced apoptosis. Cell viability and caspase 3/7 activity were evaluated using the ApoTox-Glo Triplex Assay Kit in CAOV2 cells 72 hours after transfection (n=3).

SI 1H: Immunofluorescence staining was performed with a rabbit anti-BAD or pBADS99 and mouse monoclonal anti-Flag antibodies in Flag tagged wild type *hBAD* and *hBADS99A* transfected CAOV2 cells. Scale bar, 50 μm.

2. The localization of hBADS99A-Flag is not consistent (compare SI 1H Flag-tag signal in upper versus lower panels). The upper panel Flag-tag shows BAD dispersed in the cytosol. The lower shows BAD polarized to one side of the membrane. The upper and lower samples should be the same in that channel. What is the reason for this difference?

Response: As mentioned above in the response to point 2, we have reperformed the immunofluorescence experiments under controlled and consistent conditions, with a constant excitation period and intensity, combined with higher resolution and generated new images. The cellular localization of both *Flag-tagged* constructs appears equivalent and consistent with endogenous BAD.

3. The western blot (SI 1G) attempts to discriminate transfected BADS99A-Flag from endogenous BAD. In the Flag-BAD probed blot, why is endogenous BAD, (lower band) increased with transfection?

Response: We acknowledge the reviewer's concern. On re-examination of the data, endogenous BAD and transfected *Flag-BAD* could not be specifically discriminated in western blot as the Flag-tag is approximately 1 kDa. Indeed, when the blot quality was improved, transfected *BADS99A-Flag* and endogenous BAD fell within the same band (SI 1G). To confirm that *BADS99A-Flag* was expressed after transfection, we also performed the western blot analysis with a Flag-tag antibody as described below. We also performed densitometric analysis to show that the ratio of pBADS99/BAD decreased after forced expression of *pBADS99A*, as expected. We have now incorporated this new data in the supplementary information section of the revised manuscript (SI 1G), as described below:

Supplementary information 1.

G. Transient transfection of hBADS99A in CAOV2 cells

SI 1G: Forced expression of Flag-hBAD and Flag-hBADS99A in CAOV2 was confirmed by western blot analysis. Forced expression of Flag-hBADS99A in CAOV2 cells reduced cell viability and induced apoptosis. Cell viability and caspase 3/7 activity were evaluated using the ApoTox-Glo Triplex Assay Kit in CAOV2 cells 72 hours after transfection (n=3).

Reviewer's original comment 6: "Controls are needed for some of the immunohistochemistry data. Importantly, the specificities of the anti-P-BAD-S99 and anti-BAD antibodies are not clear."

(a) The authors provide new data in SI 8C however there are some questions regarding this data. In SI 8C, the control scrambled oligo samples appear to be stained differently than siRNA-BAD. There is a clear difference in hematoxylin (blue) staining between the upper and lower panels. Additionally, the siRNA-BAD alters the cell size and shape. The cells have a different morphology than the control scrambled-oligo treatment. Clarification is needed.

Response: To address the reviewer's concern regarding the use of *siRNA* to BAD, we have now also generated stable pooled A2780-Vector and A2780-BAD-knock out (KO) cells using *CRISPR-CAS9* system mediated deletion of the *BAD* gene in A2780 cells. Next, to confirm the specificities of the anti-pBADS99 and anti-BAD antibodies, we performed western blot, immunocytochemistry (ICC, as used for histology samples), and immunofluorescence analysis of A2780-Vector and A2780-BAD-KO cells. The impact of BAD on cell morphology is not the focus of this manuscript and these experiments have been performed to simply demonstrate antibody specificity. We have now incorporated new data in the results section of the revised manuscript (SI 8C), as described below:

SI 8C: Representative images of BAD and pBADS99 immunoreactivities (western blot,

ICC, and immunofluorescence) in A2780 cells with CRISPR-CAS9 mediated deletion of BAD. Top right: immunocytochemistry, 40x magnification, Scale bar, 50µm and 200x magnification, Scale bar, 20µm. Below: immunofluorescence, 1000x magnification, Scale bar, 20µm.

(b) The authors add: “To further exemplify the specificity of the BAD and pBADS99 antibody, we also transfected CAOV2 cells with the flag-tagged wild type hBAD and hBADS99A as part of Figure (SI 1H). There was no p-BADS99 immunoreactive signal observed in cells transfected with the phosphorylation deficient BADS99A mutant whereas control and hBAD transfected cells exhibited pBADSer99 immunoreactivity”. However, again, this observation is a concern. Why does the pBADS99 antibody not react with the endogenous pBAD upon BADS99A-Flag transfection? The antibody clearly reacts with endogenous BAD in the vector control samples. The authors should explain this.

Response: We have addressed this concern of the reviewer in the response to Reviewer’s comment 5. We have also modified the text of the manuscript to reflect the new data as suggested by the reviewer as below;

Line 125-140:

“Furthermore, the functional contribution of BADS99 in regulating EOC cell survival was assessed by transient-transfection of human BAD and a S99 mutated human BAD (hBADS99A) with a flag-tag (construct described in Supplementary text) into CAOV2 cells. As shown in SI 1G, forced expression of hBAD-flag in CAOV2 cells increased BAD expression and pBADS99 levels compared to Vector transfected cells. Forced expression of hBADS99A-Flag increased hBAD compared to vector transfected cells and produced equivalently increased hBAD (and Flag-tag) as was observed in hBAD-flag transfected cells, compared to vector transfected cells. Forced expression of hBADS99-Flag did not appreciably decrease hBADS99 compared to vector transfected cells but pBADS99 was significantly less than that observed in hBAD-flag transfected cells. Hence, densitometric analysis showed that the ratio of pBADS99/BAD decreased after forced expression of pBADS99A. Functionally, forced expression of hBADS99A in CAOV2 cells resulted in decreased cell viability ($p=0.002$) and increased caspase 3/7 activity ($p=0.0003$) compared to control vector transfected cells (SI 1G). To confirm the Flag-hBADS99A construct was appropriately located within the cell, immunofluorescence (IF) was performed with BAD, pBADS99 and flag antibodies in flag-tagged wild type hBAD and hBADS99A transfected CAOV2 cells (SI 1H). The results are consistent with the western blot. Note that similar to wild-type BAD protein, flag-tagged constructs localized within the cytoplasm of the cells (SI 1H).”

Supplementary information 1.

G. Transient transfection of *hBADS99A* in CAOV2 cells

H. Confocal microscopy of tagged *hBAD-Flag* and *hBADS99A-Flag* construct in CAOV2

SI 1G: Forced expression of *Flag-hBAD* and *Flag-hBADS99A* in CAOV2 was confirmed by western blot analysis. Forced expression of *Flag-hBADS99A* in CAOV2 cells reduced cell viability and induced apoptosis. Cell viability and caspase 3/7 activity were evaluated using the ApoTox-Glo Triplex Assay Kit in CAOV2 cells 72 hours after transfection (n=3).

SI 1H: Immunofluorescence staining was performed with a rabbit anti-BAD or pBADS99 and mouse monoclonal anti-Flag antibodies in Flag tagged wild type *hBAD* and *hBADS99A* transfected CAOV2 cells. Scale bar, 50 μm.

Reviewer's comment 10: In Fig. 4C, the authors conclude that caspase 7 cleavage is elevated in response to N/O vs N or O alone. This is not convincing on the representative western blot." The authors incorporated a new cleaved-CASPASE7 blot in Figure 4C of the revised manuscript. This new blot must be accompanied with a loading control from that same membrane. The b-actin loading control in the original figure is not relevant for a different membrane.

Figure 4C of the revised manuscript. This new blot must be accompanied with a loading control from that same membrane.

Response: All blots presented in the manuscript are normalized to input (endogenous) β -ACTIN blot for analysis purposes. To address the reviewer's concern, we have now incorporated the new cleaved-CASP7 blot with its specific loading control (shown below):

Figure 4C: Western blot analysis of ascitic fluid cancer (AFC) cell derived organoids (also designated as PDO-1) in 3D Matrigel after treatment with NPB (N) alone or in combination with Olaparib (O), Rucaparib (R) or Talazoparib (T). Organoids were harvested after depolymerization of Matrigel using Cultrex® organoid harvesting solution (R&D systems, US) and organoid-pellets were resuspended in RIPA buffer plus protease inhibitors. Extracts were run on an SDS-PAGE and immunoblotted as described in materials and methods. β -ACTIN (ACTB) was used as input control for cell lysate. The sizes of detected protein bands in kDa are shown on the left side.

Reviewer's original comment 12: "In Figure 5A, xenograft experiment, how was the statistical analysis done?" The authors respond, "The statistical differences between the treatment groups were compared using an unpaired two-tailed Student's t test based on endpoint tumor volume."

Statistical analysis using Student's t test does not account for error of multiple comparisons. An appropriate multiple comparison test should be used. Possibly a one-way ANOVA followed by a Tukey's multiple comparison test is appropriate, but the authors must first test the distribution of their data.

Response: In accordance with the reviewer's suggestion, we have now performed a one-way ANOVA followed by a Tukey's multiple comparison test on endpoint tumor volume and incorporated the revised data in Figure 5A and in the materials and methods section of the revised manuscript.

Figure 5.

A. Xenograft

Figure 5A: A2780 or A2780 cisplatin resistant (5×10^6) cells were injected subcutaneously into the flank of 5-week-old BALB/c athymic mice, respectively. When the xenograft reached 100-150m³ in each cell line group, the mice were randomized into the 4 indicated treatment groups (n = 6). Mice were treated daily with vehicle (V), 20 mg/kg NPB (N), 50 mg/kg Olaparib (O) or combined NPB-Olaparib by i.p. injection. Xenograft growth was monitored daily by measurement of volume. The statistical change between treatment groups were analyzed using a one-way ANOVA followed by a Tukey's multiple comparison test on endpoint xenograft volume. Below: mean animal weights of each treatment group are indicated. Animal weight was monitored daily. Yellow triangle points the start of the treatment. Brown triangle points the end of the treatment. Results represent the mean \pm SEM of six animals. **(B)** Mean xenograft weight of each treatment group after sacrifice on the 8th day. Results represent the mean \pm SEM of six animals.

Line 480-482:

“The statistical differences between the treatment groups were compared using a one-way ANOVA followed by a Tukey's multiple comparison test.”

Reference

1. Stark, J.M., et al., *Genetic steps of mammalian homologous repair with distinct mutagenic consequences*. Mol Cell Biol, 2004. **24**(21): p. 9305-16.
2. Chen, C.C., et al., *ATM loss leads to synthetic lethality in BRCA1 BRCT mutant mice associated with exacerbated defects in homology-directed repair*. Proc Natl Acad Sci U S A, 2017. **114**(29): p. 7665-7670.
3. Pandey, V., et al., *Discovery of a small-molecule inhibitor of specific serine residue BAD phosphorylation*. Proc Natl Acad Sci U S A, 2018. **115**(44): p. E10505-E10514.
4. Bui, N.L., et al., *Bad phosphorylation as a target of inhibition in oncology*. Cancer Lett, 2018. **415**: p. 177-186.

REVIEWERS' COMMENTS:

Reviewer #1 (Remarks to the Author):

Authors have sufficiently addressed the concerns raised in my previous review.

Reviewer #2 (Remarks to the Author):

The authors have responded adequately.